# MTCH2 controls energy demand and expenditure to fuel anabolism during adipogenesis

Sabita Chourasia [1✉], Christopher Petucci[2], Clarissa Shoffler [2], Dina Abbasian[2], Hu Wang[3], Xianlin Han [3], Ehud Sivan [4], Alexander Brandis [4], Tevie Mehlman[4], Sergey Malitsky [4], Maxim Itkin [4], Ayala Sharp[4], Ron Rotkopf [4], Bareket Dassa[4], Limor Regev [1], Yehudit Zaltsman[1] & Atan Gross [1✉]

## Abstract

**Mitochondrial carrier homolog 2 (MTCH2) is a regulator of apoptosis, mitochondrial dynamics, and metabolism. Loss of MTCH2 results in mitochondrial fragmentation, an increase in whole-body energy utilization, and protection against diet-induced obesity. In this study, we used temporal metabolomics on HeLa cells to show that MTCH2 deletion results in a high ATP demand, an oxidized cellular environment, and elevated utilization of lipids, amino acids, and carbohydrates, accompanied by a decrease in several metabolites. Lipidomics analysis revealed a strategic adaptive reduction in membrane lipids and an increase in storage lipids in MTCH2 knockout cells. Importantly, MTCH2 knockout cells showed an increase in mitochondrial oxidative function, which may explain the higher energy demand. Interestingly, this imbalance in energy metabolism and reductive potential triggered by MTCH2-deletion prevents NIH3T3L1 preadipocytes from differentiating into mature adipocytes, an energy consuming reductive biosynthetic process. In summary, the loss of MTCH2 leads to increased mitochondrial oxidative activity and energy demand, creating a catabolic and oxidative environment that fails to fuel the anabolic processes required for lipid accumulation and adipocyte differentiation.**

**Keywords** MTCH2; Energy Expenditure and Demand; Mitochondrial Oxidative Function; Adipogenesis
**Subject Categories** Development; Metabolism; Molecular Biology of Disease

## Introduction

Energy homeostasis is a vital physiological process essential for the survival and well-being of organisms. It is finely regulated through dynamic processes that balance energy intake with expenditure (Löffler et al, 2021; Galgani and Ravussin, 2008). AMP-activated protein kinase (AMPK), sirtuins (like SIRT1), and mTOR function as cellular energy sensors, regulating metabolic pathways and promoting energy conservation during periods of low energy availability (Abu Shelbayeh et al, 2023; Steinberg and Carling, 2019; Crosas-Molist et al, 2023; Saxton and Sabatini, 2017).

At the center of this complex system are the mitochondria, which play a crucial role in energy production, sensing, adaptation, and response to cellular energy demands. Mitochondria are the primary sites of cellular respiration, where oxidative phosphorylation (OXPHOS) converts the chemical energy stored in nutrients into adenosine triphosphate (ATP) (Picard and Shirihai, 2022; Han et al, 2023; Monzel et al, 2023; Berry et al, 2023). Redox cofactors, such as the oxidized ($NAD^+$) and reduced (NADH) forms of nicotinamide, serve as "fuel" for mitochondria (Xie et al, 2020; Yang and Sauve, 2016; Formentini et al, 2009). Mitochondrial NADH acts as a substrate for oxidative phosphorylation (OXPHOS), the process responsible for 90% of cellular ATP production (Formentini et al, 2009), additionally, $NAD^+$ stimulates the Krebs cycle. The relationship between mitochondrial redox cofactors and fuel governs the metabolic state and defines the overall blueprint for cellular bioenergetics and behavior (Hu et al, 2021).

Mitochondria are dynamic structures, constantly undergoing fission and fusion processes known as mitochondrial dynamics. This behavior is crucial for maintaining a healthy mitochondrial population, responding to energy demands, and enabling adaptability in response to changing metabolic conditions (Youle and Van Der Bliek, 2012; Liesa and Shirihai, 2013; Han et al, 2023; Monzel et al, 2023). In addition to its dynamic behavior, mitochondrial biogenesis is closely tied to energy homeostasis. Peroxisome proliferator-activated receptor gamma coactivator 1-alpha (PGC-1α) serves as a master regulator, orchestrating mitochondrial biogenesis in response to energy demands, environmental signals, and cellular stress (Scarpulla, 2010; Handschin and Spiegelman, 2006). Mitochondria preserve their health through a delicate balance between the biogenesis of new mitochondria and the clearance of old or dysfunctional ones via mitophagy (Liu et al, 2023).

[1]Department of Immunology and Regenerative Biology, Weizmann Institute of Science, 76100 Rehovot, Israel. [2]Metabolomics Core, Penn Cardiovascular Institute, Perelman School of Medicine, University of Pennsylvania, Philadelphia, PA 19104, USA. [3]Barshop Institute for Longevity and Aging Studies, and Department of Medicine, University of Texas Health Science Center at San Antonio, San Antonio, TX 78229, USA. [4]Department of Life Sciences Core Facilities, Weizmann Institute of Science, 76100 Rehovot, Israel. ✉E-mail: sabita.chourasia@weizmann.ac.il; atan.gross@weizmann.ac.il

Metabolism plays a vital role in cell differentiation and adaptation during periods of cellular energy stress. The dynamic transition from glycolysis to oxidative phosphorylation (OXPHOS) during differentiation highlights the adaptability and plasticity of cellular metabolism. Glucose metabolism is essential for meeting the energy needs of undifferentiated and proliferating cells, a phenomenon known as the Warburg effect (Vander Heiden et al, 2009). The switch from glycolysis to OXPHOS lies at the intersection of metabolism and cell fate, highlighting the close relationship between energy metabolism and cellular differentiation (Folmes et al, 2012). Adipocyte differentiation, or adipogenesis, requires a finely balanced regulation of energy utilization, involving dynamic shifts in energy metabolism to facilitate lipid accumulation and storage (Rosen and MacDougald, 2006). Transcriptional regulators, such as PPARγ, C/EBPs, and AMPK, serve as key modulators of the balance between energy storage and expenditure during adipocyte differentiation. Their complex interaction shapes the metabolic landscape of adipose tissue (Vernochet et al, 2014). NAD$^+$ serves as a crucial signaling molecule for the activation of PPARγ and C/EBPs during adipocyte differentiation (Barile et al, 2020). During adipocyte differentiation, the cytoplasmic NAD$^+$ pool helps meet cellular metabolic demands by regulating glucose metabolism, while the nuclear NAD$^+$ pool is involved in gene regulation (Ryu et al, 2018; Osawa et al, 2021).

Mitochondrial carrier homologue 2 (MTCH2), also known as MIMP or SLC25A50, is a distinct member of the SLC25A mitochondrial carrier protein family, located at the outer mitochondrial membrane (OMM) (Robinson et al, 2012). It was initially recognized for its involvement in mediating apoptosis (Grinberg et al, 2005; Zaltsman et al, 2010). Subsequent studies, however, revealed its multifaceted involvement in regulating mitochondrial and whole-body metabolism, as well as influencing hematopoietic stem cell fate(Buzaglo-Azriel et al, 2016; Maryanovich et al, 2015). Additionally, multiple genome-wide association studies have linked the MTCH2 locus to metabolic disorders such as diabetes and obesity (Speliotes et al, 2010; van Vliet-Ostaptchouk et al, 2014; Willer et al, 2009). More recently, MTCH2 has been implicated in the regulation of mitochondrial dynamics, further expanding its functional roles in the cell (Bahat et al, 2018; Labbé et al, 2021). Studies using genetic models in organisms like C.elegans, zebrafish, and mice have revealed MTCH2's involvement in lipid metabolism (Kulyté et al, 2011; Landgraf et al, 2016; Latorre-Muro et al, 2021; Rottiers et al, 2017). MTCH2 deletion has been associated with reduced lipid synthesis and storage, emphasizing its crucial role in lipid homeostasis (Buzaglo-Azriel et al, 2016; Landgraf et al, 2016; Rottiers et al, 2017). On the other hand, increased MTCH2 expression has been linked to enhanced lipid storage, highlighting its dynamic regulatory function in lipid metabolism (Kulyté et al, 2011).

A recent study has highlighted MTCH2's role in mitochondrial fusion, linking it to the pro-mitochondrial fusion lipid lysophosphatidic acid (LPA) (Labbé et al, 2021). This study suggests that MTCH2 is crucial for the biogenesis and transfer of lipids from the ER to the mitochondria, a process that is essential for mitochondrial fusion. Furthermore, MTCH2 has recently been shown to function as a protein insertase (Guna et al, 2022), and potentially as a scramblase (Li et al, 2024; Bartoš et al, 2024).

In this current study, we demonstrate that the loss of MTCH2 leads to increased energy demand and heightened metabolism across various pathways, most likely in an effort to meet this increased demand.

## Results

### A high ATP demand and an oxidizing environment in MTCH2 knockout cells

To evaluate MTCH2's metabolic function, we performed an untargeted global metabolomic analysis on HeLa cells. The analyses were performed on stable cell lines, of three genotypes: wild type (WT) with empty vector, MTCH2 knockout (MKO), and MKO reconstituted with MTCH2 (MKO-R) (Fig. EV1A,B). All stable lines were cultured in complete media (CM), and cells were collected for global metabolomics. We identified a total of 194 metabolites, of which 107 showed a decrease in MKO cells, and most of them were rescued to WT levels in the MKO-R cells (Fig. EV1C). Pathway enrichment analysis revealed alterations in several metabolic pathways and cycles, including glyoxylate and dicarboxylate metabolism, the TCA cycle, the pentose-phosphate pathway (PPP), amino acid metabolism, nucleotide (purine and pyrimidine) metabolism, and nitrogen metabolism (Fig. EV1D,E). The decreased metabolite levels in these pathways could reflect a persistently lower metabolic state in MTCH2 knockout cells (MKO), or may result from changes in nutrient utilization kinetics specific to MKO cells.

To differentiate between these hypotheses, we performed temporal targeted metabolomics, focusing on the pathways mentioned above. Samples were collected at 1, 6, 12, and 18 h-post media change. Hierarchical clustering and principal component analysis (PCA) based on the differential metabolite levels between the three groups revealed that WT and MKO-R cells clustered together, whereas MKO cells formed a distinct subpopulation (Fig. 1A). These results indicate that MTCH2 knockout leads to significant metabolic alterations, which are largely restored to the WT metabolic state upon reconstitution of MTCH2.

A detailed analysis of the results showed that MKO cells showed an increase in ADP levels at 12 h (Fig. 1B; top row, left panel), an increase in the ADP/ATP ratio at 12 and 18 h (top row, right panel), an increase in the NAD$^+$/NADH ratio (middle row, middle panel), an increase in NADP$^+$ levels (middle row, right panel), and an increase in the nicotinamide precursors, such as nicotinamide adenine dinucleotide (NAAD) and nicotinamide (NAM) at most time points (bottom row).The increase in the ADP/ATP ratio led us to check the phospho-AMPK levels, and indeed the MKO cells showed higher levels after 18 h-post media change (Fig. 1C). Overall, MTCH2 deletion led to an increase in both the ADP/ATP ratio and nicotinamide metabolism.

Importantly, the majority of the metabolic changes measured in the MKO cells were rescued to WT levels in the MKO-R cells (Fig. 1B), suggesting that these changes were directly caused by MTCH2 knockout.The increase in the ADP/ATP ratio stimulates oxidative phosphorylation (OXPHOS), while the elevated NAD +/NADH ratio and NADP+ levels indicate an oxidized environment, which in turn stimulates glycolysis (Hopp et al, 2019; Hikosaka et al, 2014). Collectively, these results suggest that

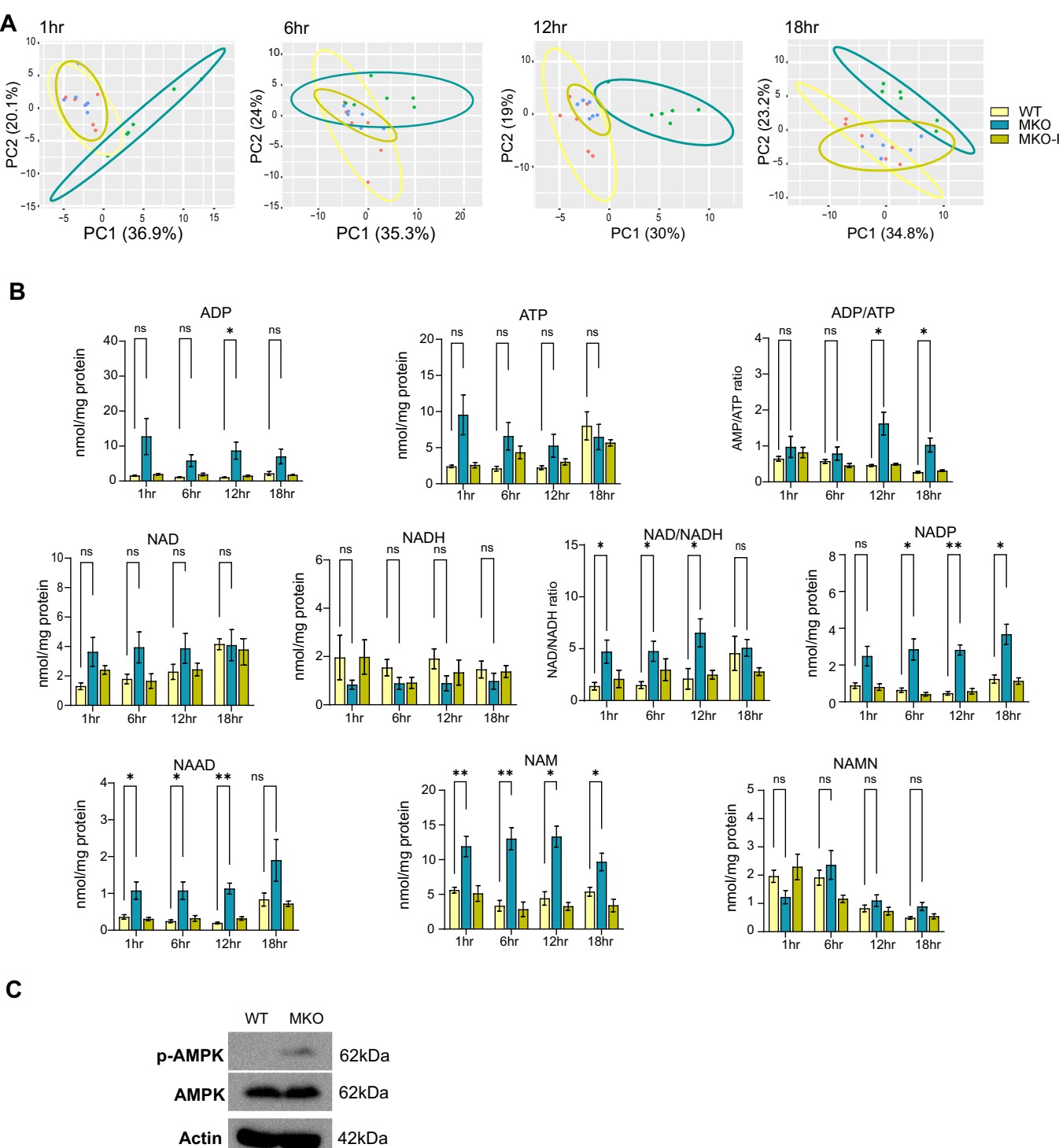

**Figure 1. A high ATP demand and an oxidizing environment in MTCH2 knockout cells.**

(A) PCA plots of WT, MKO, and MKO-R cell lines. Ellipses describe 95% confidence intervals. (B) Average levels of ADP, ATP, and ADP/ATP, NAD+, NADH, NAD+/NADH, NADP+, NAAD, NAM, and NAMN in all 3 cell lines at all four-time points. Results in all graphs in (B) are presented as mean ± SEM (*$P < 0.05$, **$P < 0.001$; two-way ANOVA with Dunnett multiple comparison test; $n = 6$ biological replicates). (C) MKO cells show an increase in AMPK phosphorylation. Western blot analysis of WT and MKO cells using anti-p-AMPK and anti-AMPK antibodies. β-Actin is used as a loading control. Source data are available online for this figure.

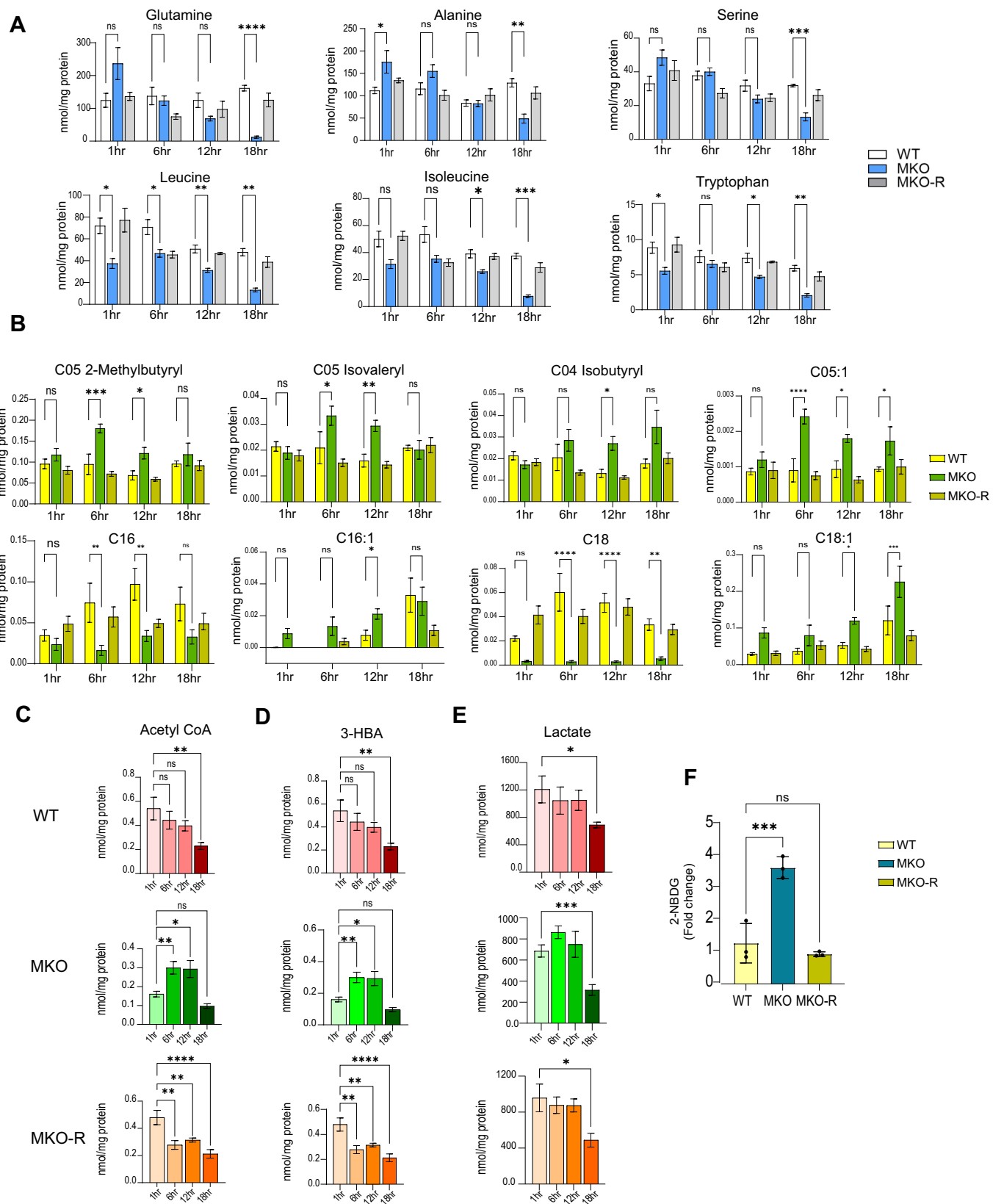

**Figure 2. An increase in amino acid/lipid/carbohydrate utilization in MKO cells.**

(A) Average levels of a set of amino acids in all 3 cell lines at all four time points. (B) Average levels of acylcarnitines in all 3 cell lines at all four time points. (C–E) Average levels of acetyl CoA (C), 3-HBA (D) and lactate (E) in all 3 cell lines at all four time points. Results in all graphs in (A–E) are presented as mean ± SEM (ns, non-significant, *$P<0.05$, **$P<0.001$, ***$P<0.0003$, ****$P<0.0007$; two-way ANOVA with Dunnett multiple comparison test ($n = 6$ independent biological replicates)). (F) Glucose uptake from media of WT and MKO cells after two-hour incubation in low-glucose medium. Results are presented as mean ± SEM (***$P<0.0003$; one-way ANOVA with Dunnett multiple comparison test ($N = 3$ independent experiment)). Source data are available online for this figure.

MTCH2 knockout results in the stimulation of oxidative metabolism and ATP production likely to meet the increased cellular energy demand.

## An increase in amino acid/lipid/carbohydrate utilization in MKO cells

The metabolomics analyses revealed additional significant changes in various nutrient substrates, including a decrease in most amino acids that enter the TCA cycle at different points (Sherry et al, 2015) (Figs. 2A and EV2A). Notably, a distinct pattern of change was observed in glutamine, a major amino acid-nutrient source(Yang et al, 2014), along with alanine and serine. These amino acids initially showed an increase in MKO cells, followed by a substantial decrease, which reached statistical significance only 18 h after media change (Fig. 2A, top panel). A similar profile of decreased levels at later time points was observed for several other amino acids (Figs. 2A, bottom panel and EV2A). A decrease in amino acids levels typically indicates increased TCA cycle activity (Shiratori et al, 2019), and indeed, we observed a decrease in TCA cycle intermediates 18 h post-media change in MKO cells (Fig. EV2B).

Acylcarnitines serve as a "ready-made" form of fatty acids, capable of entering the mitochondria passively for breakdown. In our targeted metabolomic analyses, we detected 22 species of acylcarnitines, including three branched-chain amino acid (BCAA) acylcarnitines (C05-2-Methylbutyryl, C05-Isovaleryl, and C04-Isobutyryl), five unsaturated acylcarnitines (C05:1, C16:1, C18:1, C14:2, C22:5), and eight saturated acylcarnitines (Figs. 2B and EV2C). MKO cells showed a sequential increase in BCAAs and unsaturated short-chain acylcarnitines (C05:1) at 6 and 12 h post-media change, with a further increase in some unsaturated acylcarnitines at later time points (Fig. 2B). Most of these changes were rescued to WT levels in the MKO-R cells, suggesting that the observed metabolic alterations were due to MTCH2 knockout. The increased acylcarnitine levels in MKO cells are consistent with enhanced transport and breakdown of fatty acids in mitochondria to meet the elevated cellular ATP demand.

On the other hand, MKO cells shows a decrease in some of the short and long-chain saturated acylcarnitines, which were restored to WT levels in the MKO-R cells (Fig. EV2C). An interesting comparison can be made between two of the saturated acylcarnitines, C16 and C18, which both showed decreased levels, while their corresponding unsaturated forms, C16:1 and C18:1, were increased (Fig. 2B). Thus, our data shows there is sequential increase in BCAA and unsaturated acylcarnitines over time in MKO cells.

Interestingly, the levels of acetyl-CoA, 3-hydroxybutyrate (3-HBA), and lactate were consistently lower in MKO cells compared to WT cells at all time points (Fig. 2C–E, respectively). Additionally, all three metabolites showed greater fluctuations in

MKO cells than in WT or MKO-R cells, with MKO cells showing an initial low level, followed by an increase, and then a decrease. Acetyl-CoA's primary role is to deliver the acetyl group to the TCA cycle for oxidation and energy production (Owen et al, 2002; Martínez-Reyes and Chandel, 2020). One hour-post media change, acetyl-CoA levels were nearly three times lower in MKO cells compared to WT and MKO-R cells (Fig. 2C). However, at 6 and 12 h, acetyl-CoA levels in MKO cells increased almost two-fold, whereas levels in WT and MKO-R cells gradually decreased (Fig. 2C). At 18 h post media change, the trend reversed again, with MKO cells showing lower acetyl-CoA levels compared to WT and MKO-R cells (Fig. 2C).

Similar dynamics were observed in the levels of the ketone body 3-hydroxybutyrate (3-HBA; Fig. 2D), which is an alternative product of fatty acid oxidation and serves as an energy source when glucose is insufficient or when there is a preference for a lipogenic diet (Nishitani et al, 2018, 2022). A similar trend was seen in lactate levels (Fig. 2E), although the differences in lactate levels were less pronounced and did not reach statistical significance. Additionally, we assessed glucose uptake, and found that MKO cells showed more than a two-fold increase in uptake compared to WT and MKO-R cells (Fig. 2F).

In summary, MTCH2 deletion results in increased glucose uptake, a decrease in various internal metabolites, and an increase in BCAAs and unsaturated acylcarnitines. This pattern likely reflects the increased utilization of a wide range of nutrients driven by elevated energy demands.

## Increased mitochondrial oxidative function in MKO cells

Based on the findings outlined above, we examined mitochondrial respiration and observed a significant increase in both basal and maximal oxygen consumption in MKO cells compared to WT cells (Fig. 3A, left and right panels). The increased basal respiration further supports the heightened energy demand, which likely accounts for the increased utilization of amino acids, lipids, and carbohydrates in MKO cells. In addition, the increase in maximal respiration reflects the additional capacity of the MKO cells to produce energy under stress conditions, aiding their survival.

To further investigate mitochondrial oxidative function, we measured NADH autofluorescence and calculated the NADH redox index. As NADH acts as the electron donor for complex I, its levels are inversely correlated with respiratory chain activity. To assess the redox index, we first added FCCP to maximize respiration, which reduced the mitochondrial NADH pool. We then used KCN to inhibit mitochondrial respiration, thereby maximizing the NADH pool, as described previously (Plun-Favreau et al, 2012; Maryanovich et al, 2015). Our results showed a decrease in the redox level in MKO cells (Fig. 3B), indicating that the increased respiration rate in these cells is due to accelerated electron

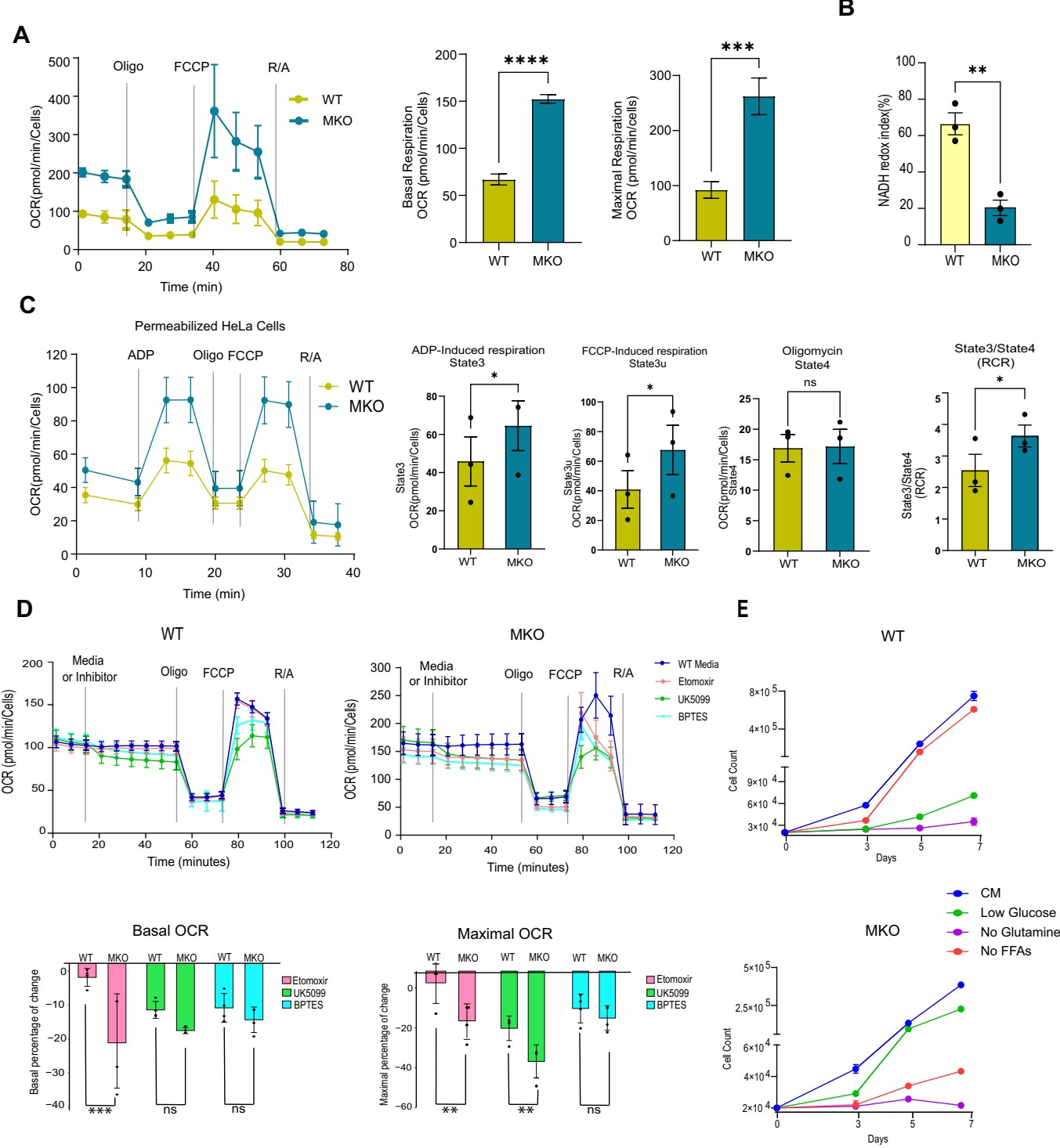

1012    *The EMBO Journal* Volume 44 | Issue 4 | February 2025 | 1007–1038

consumption. Additionally, the total mitochondrial NADH pool (after KCN treatment) in MKO cells was significantly higher than in WT cells, indicating increased substrate availability (Fig. EV3B). This rise in the NADH pool after KCN treatment supports the notion that MKO cells exhibit enhanced TCA cycle efficiency, contributing to the observed increase in TCA cycle activity and the reduction in its intermediates over time (Fig. EV2B).

To explore this effect further, we used permeabilized WT and MKO cells to measure ADP phosphorylation respiration. We found that MTCH2 deficiency led to increased oxygen consumption during state 3 respiration (ADP-induced respiration; Fig. 3C), but no difference in state 4 respiration rates (after oligomycin treatment) between WT and MKO cells (Fig. 3C). The respiratory control ratio (RCR), which reflects the degree of coupling between

**Figure 3.   Increased mitochondrial Oxidative function in MKO cells.**

(A) Representative traces of real-time OCR measurements in WT and MKO cells display basal respiration and OCR changes in response to sequential inhibitor injections (Left panel). The middle and right panels show basal and maximal respiration. Results are presented as mean ± SEM, with statistical significance determined by an unpaired *t* test (***$P<0.0003$, ****$P<0.0007$). Data represents one of three independent experiments, each with 6–8 technical replicates. (B) Reduced NADH redox in MKO cells. NADH redox index was calculated as described in "Methods". Results are presented as mean percent ± SEM (Unpaired *t* test, **$P<0.001$). $N = 3$ independent experiments. (C) Respiratory Control Ratio (RCR) analysis on permeabilized WT and MKO cells. Succinate and rotenone are used to measure ADP phosphorylation and maximal respiration. State 3, State 3u, State 4, and RCR were calculated as described in "Methods". Results are presented as mean percent ± SEM (Unpaired *t* test, *$P<0.05$; ns, non-significant). $N = 3$ independent experiments. (D) Representative traces of real-time OCR measurements of WT (top left panel) and MKO (top right panel) cells showing the basal respiration and OCR change in Basal and Maximal respiration Response to sequential injections of inhibitors. Percentage of change in basal OCR (bottom left panel) and Maximal OCR (bottom right panel) in response to UK5099, etomoxir, and BPTES in WT and MKO cells. Results are presented as mean percent ± SEM (two-way ANOVA, Post-hoc tests were done using estimated marginal means (R package 'emmeans'). **$P<0.001$, ***$P<0.0003$). $N = 4$ independent experiments. (E) Cell count of WT and MKO cells proliferating under the conditions of complete medium (25 mM glucose), and medium with no free fatty acids (FFAs), low glucose (12 mM), or no glutamine. $N = 3$ independent experiments. Source data are available online for this figure.

the mitochondrial respiratory chain and OXPHOS, was higher in MKO cells compared to WT cells (Fig. 3C). We also measured Cell Respiratory Control Ratio, as previously described (Brand and Nicholls, 2011), and the MKO cells have higher cellular RCR than WT cells (Fig. EV3A). Moreover, MKO cells exhibited increased mitochondrial membrane potential (ΔΨm; Fig. EV3C) but showed no change in mitochondrial reactive oxygen species (ROS) levels (Fig. EV3D).

Next, we explored whether the higher respiration in MKO cells leads to nutrient preferences. We examined OXPHOS activity by measuring changes in basal and maximal oxygen consumption rate (OCR) following treatment with inhibitors of pyruvate metabolism (UK5099), glutamine metabolism (BPTES), and fatty acid oxidation (etomoxir). Our results indicated that MKO cell respiration was affected by all three inhibitors, suggesting that MKO cells utilize pyruvate, glutamine, and fatty acid oxidation to support cellular respiration (Fig. 3D). In contrast, WT cells rely on pyruvate and glutamine but not on fatty acid oxidation (Fig. 3D).

Finally, we investigated whether MKO cells depend on different nutrient sources for their growth. As described previously(Han et al, 2023), we observed that restricting free fatty acids inhibited the growth of MKO cells but did not affect the growth of WT cells (Fig. 3E). Both cell types were dependent on glutamine, but MKO cells were less affected by low-glucose conditions. These findings suggest that MKO cells utilize a broader range of nutrients to support increased OXPHOS and growth, while WT cells rely on fewer nutrient sources.

## Membrane lipids decrease and storage lipids and lipid droplets increase in MKO cells

During nutrient depletion, cells adapt by breaking down membrane lipids, leading to the release of fatty acids that can be stored as energy reserves (e.g., triglycerides (TAG)) or used for immediate energy production (Hosios et al, 2022). Interestingly, under such nutrient stress, mitochondria also reorganize into interconnected networks, a structural adaptation that supports energy optimization and sustains cellular functions (Gomes et al, 2011). Targeted lipidomics analysis performed on cells harvested 20 h post media change revealed a marked decrease in membrane lipids and an increase in storage lipids in MKO cells (Fig. 4A). Importantly, these changes in lipids were largely restored in the MKO-R cells (Fig. 4A). MKO cells also showed a decrease in free fatty acids (FFA) or non-

esterified fatty acids (NEFA) (Figs. 4B, left panel and EV4A), while esterified fatty acids (TAG-fatty acid content) were elevated (Figs. 4B, right panel and EV4B). A volcano plot clearly illustrated the increase in TAGs and cholesterol esters (CEs), which are key storage lipids in MKO cells (Fig. 4C). Interestingly, a detailed analysis of the lipid data (Fig. 2B) revealed that TAGs were predominantly composed of C16:0, C16:1, and C18:1 fatty acids, and two of these—C16:1 and C18:1—also showed increased levels as acylcarnitines in MKO cells.

TAGs and CEs are major components of lipid droplets (LDs), which act as an "on-demand" energy source for cells. LDs can be mobilized in response to nutrient fluctuations (Nguyen et al, 2017). During extended periods of nutrient scarcity, cells shift from relying on glycolysis to utilizing fatty acid oxidation through mitochondrial β-oxidation for energy production. Consistent with the decrease in intracellular nutrients and membrane lipids, and the increase in TAGs and CEs, we observed an increase in LD numbers and size in MKO cells cultured in complete media 20 h after media change (Figs. 4D and EV4C). This effect was further amplified when the cells were grown in HBSS under nutrient-depletion conditions (Fig. 4E).

Notably, the accumulation of LDs in MKO cells was also accompanied by a redistribution of LDs from a dispersed to a highly clustered pattern (Fig. 4F), whereas WT cells showed only minimal changes in LD morphology (Fig. 4F). Notably, the RNA and/or protein expression of ASCL1 and SCD1, enzymes involved in TAG and CE metabolism, were upregulated in MKO cells (Fig. EV4D,E).

Previous studies, including our own, have shown that MTCH2 deletion leads to mitochondrial fragmentation (Bahat et al, 2018; Labbé et al, 2021). In this study, we found that prolonged growth of MKO cells in the same media resulted in mitochondrial elongation (Fig. EV4F), which became even more pronounced under HBSS nutrient-depletion conditions (Fig. EV4G).

In summary, the reduction in membrane lipids, the accumulation of storage lipids, the increase in LDs, and the enhanced mitochondrial elongation in MKO cells are likely adaptive responses that enable cells lacking MTCH2 to better cope with nutrient depletion and increased ATP demand.

## MTCH2 is critical for adipocyte differentiation

MTCH2 mRNA and protein levels increase in obese women and during adipocyte differentiation (Fischer et al, 2023), while MTCH2

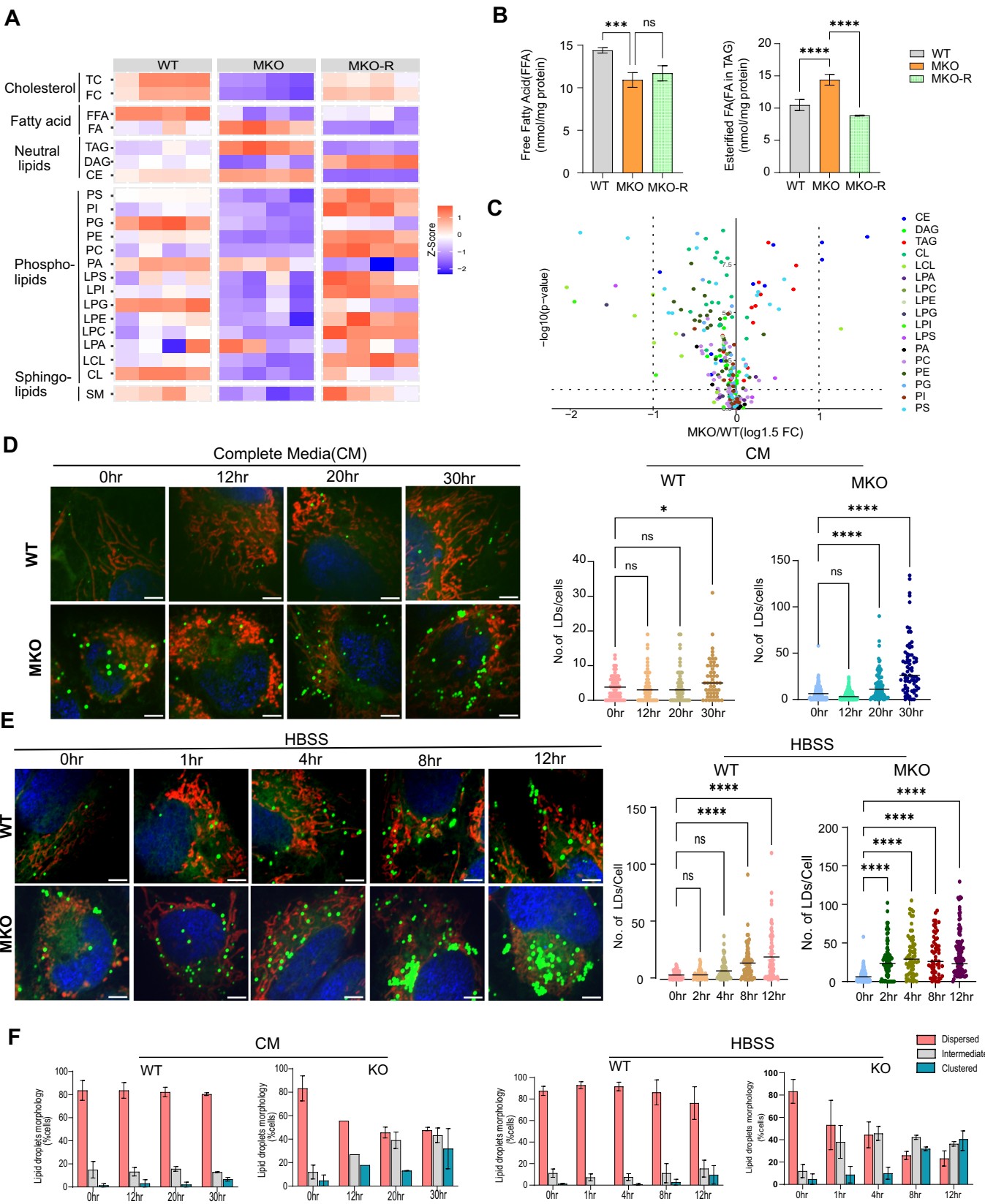

**Figure 4. Membrane lipids decrease and storage lipids and lipid droplets increase in MKO cells.**

(A) Heat map comparing the levels of lipids at 20 h post media change in all 3 cell lines. Membrane lipids: TC total cholesterol, FC free cholesterol, PC phosphatidylserine, PI phosphatidylinositol, PG phosphatidylglycerol, PE phosphatidylethanolamine, PC phosphatidylcholine, PA phosphatidic acid, LPS Lyso-phosphatidylserine, LPI Lyso-phosphatidylinositol, LPG Lyso-phosphatidylglycerol, LPE Lyso-phosphatidylethanolamine, LPC Lyso-phosphatidylcholine, LPA Lyso-phosphatidic acid, LCL Lyso-cardiolipin, CL cardiolipin, SM sphingomyelin. Neutral/Storage lipids: TAG triglycerides, CE cholesterol ester, DAG diglycerides. Fatty acids: FA esterified fatty acid, FFA free fatty acids. Cholesterol: TC total cholesterol, FC free cholesterol. We calculated the average value of all species per category per biological replicate. These values were compared between groups using ANOVA. Values in the heat map are scaled to z-scores per row (compound category). ANOVA data also appears in Table EV1. (B) Level of esterified FFA (left panel) and FA (right panel), in all 3 cell lines. Results in graphs are presented as mean ± SEM (ns, non-significant, ***$P<0.0003$, ****$P<0.0007$; One-way ANOVA, $n = 4$ biological replicates). (C) Volcano plot comparing the levels of membrane and storage lipids between the WT and MKO cell lines.one-way ANOVA followed by Tukey's post-hoc test, $n = 4$ biological replicates. (D) Left panel: WT and MKO cells were plated into complete media (CM), then media was refreshed (considered as time 0) and pictures were taken at 0, 12, 20, and 30 h-post media change. LDs were labeled using BODIPY 493/503, and mitochondria were labeled using MitoTracker deep red (MTDR) and nucleus by Hoechst. Right panel: Temporal quantification of the number of LDs in WT and MKO cells at the four time points. Results are presented as means ± SEM (ns, non-significant, *$P<0.05$, ****$P<0.0007$; $n = 3$ independent biological replicates). (E) Left panel: WT and MKO cells were plated into complete media, then transferred to HBSS and pictures were taken at 2, 4, 8, and 12 h-post media change. Right panel: Temporal quantification of the number of LDs in WT and MKO cells at five time points. Data are presented as means ± SEM (ns, non-significant, ****$P<0.0007$; $n = 3$ independent biological replicates). (F) Temporal quantification of the percentage of cells with dispersed, intermediate, or clustered LDs after incubation of cells for the indicated times in either CM (left panel; pictures of cells appear in D) or HBSS (right panel; pictures of cells appear in E). Results are presented as mean ± SEM. $n = 3$ independent biological replicates. Scale bar = 5 μm. Source data are available online for this figure.

deletion inhibits both adipogenesis and lipid accumulation in adipocyte cells (Rottiers et al, 2017; Jiang et al, 2019).

Adipose tissue is essential for regulating energy balance and glucose levels (Cristancho and Lazar, 2011), and functional adipocyte formation depends on the differentiation of preadipocytes into mature adipocytes.NAD+ biosynthesis also integrates cellular metabolism with the adipogenic transcriptional program (Ryu et al, 2018). This adipogenic process involves two key phases: an initial commitment phase, followed by the differentiation phase, which is crucial for achieving mature adipocytes (Audano et al, 2022). Therefore, adipocytes and the process of adipogenesis are likely to provide a relevant physiological model for studying MTCH2's role in metabolism.

To investigate MTCH2's role in adipocyte differentiation, we used the CRISPR-Cas9 system to generate MTCH2 knockout NIH3T3L1 cells, a standard model for adipogenesis studies. MTCH2 deletion was confirmed by examining MTCH2 mRNA and protein levels (Fig. EV5A), and as expected, MTCH2 knockout NIH3T3L1 preadipocytes exhibited mitochondrial fragmentation (Fig. EV5B). During differentiation, WT cells showed 80–90% differentiation by day 6, whereas MTCH2 knockout cells only reached 5–10% differentiation (Fig. 5A, left panel). MTCH2 knockout cells also showed a marked reduction in lipid droplets (LDs) 6 days post-differentiation (Fig. 5A, right panel). This decrease in LD expansion led us to question whether MTCH2 knockout cells are impaired in TAG accumulation within LDs or if the entire differentiation program is compromised.

To address this, we assessed mRNA levels of key genes involved in adipocyte differentiation, comparing levels between day 0 and day 6 post-differentiation. WT cells exhibited substantial upregulation of adipogenic transcription factors and related effectors, which was absent in MTCH2 knockout cells (Figs. 5B and EV5C). Targeted metabolic profiling of preadipocytes revealed increased levels of NAD+, NADP+, NADH, AMP, and ATP in MTCH2 knockout cells compared to WT cells (Fig. 5C). High levels of ATP and reducing equivalents are crucial for anabolic processes like lipid synthesis (Chen et al, 2019). The elevated NAD+ and NADP+ levels in MTCH2 knockout preadipocytes indicate a more oxidizing intracellular environment, which inhibits anabolism and thereby lipid biosynthesis (Chen et al, 2019). Despite higher ATP

levels in MTCH2 knockout preadipocytes than in WT, AMP levels were even more elevated, leading to a more than 5-fold increase in the AMP/ATP ratio compared to WT preadipocytes (Fig. 5C).

Together, the targeted metabolic profiling of MTCH2 knockout preadipocytes shows distinct profiles in both energy (AMP, ATP) and redox (NAD+, NADH, NADP+) metabolism compared to WT cells, which is similar to the metabolic profile observed in MKO HeLa cells. Moreover, targeted metabolomics analysis comparing day 0 to day 6 post-differentiation showed that while WT cells had expected increases in NAD+, NADP+, and ATP to support anabolism during differentiation, these metabolites were decreased in MTCH2 knockout cells (Fig. 5D). Thus, by examining the metabolic profiles at day 0 and day 6 in both WT and MTCH2 knockout cells, we identified dysregulated timing in the increase and decrease of metabolites critical for biosynthesis, preventing MTCH2 knockout preadipocytes from fully differentiating into adipocytes.

In summary, MTCH2 deletion disrupts energy metabolism and redox cofactor pathways, resulting in inadequate expression of transcription factors essential for adipocyte differentiation.

# Discussion

In this study, we aimed to understand the role of MTCH2 in metabolism. While MTCH2 is well-known for its role in apoptosis as the mitochondrial receptor for the pro-apoptotic protein BID (Zaltsman et al, 2010), its functions in mitochondrial dynamics and metabolic regulation are less explored. Interestingly, conditional knockout of MTCH2 in mouse skeletal muscle has been shown to protect against high-fat diet-induced obesity, likely due to increased whole-body energy utilization (Buzaglo-Azriel et al, 2016).

How does the loss of MTCH2 lead to increased energy utilization? Our findings suggest that MTCH2 plays a critical role in balancing energy flow across multiple metabolic pathways based on cellular demand and resource utilization. MTCH2 appears to regulate the activity of multiple metabolic pathways, ensuring the proper timing and sustainable use of metabolic intermediates to support cellular growth and proliferation. Under normal conditions, MTCH2, located on the mitochondrial surface, likely

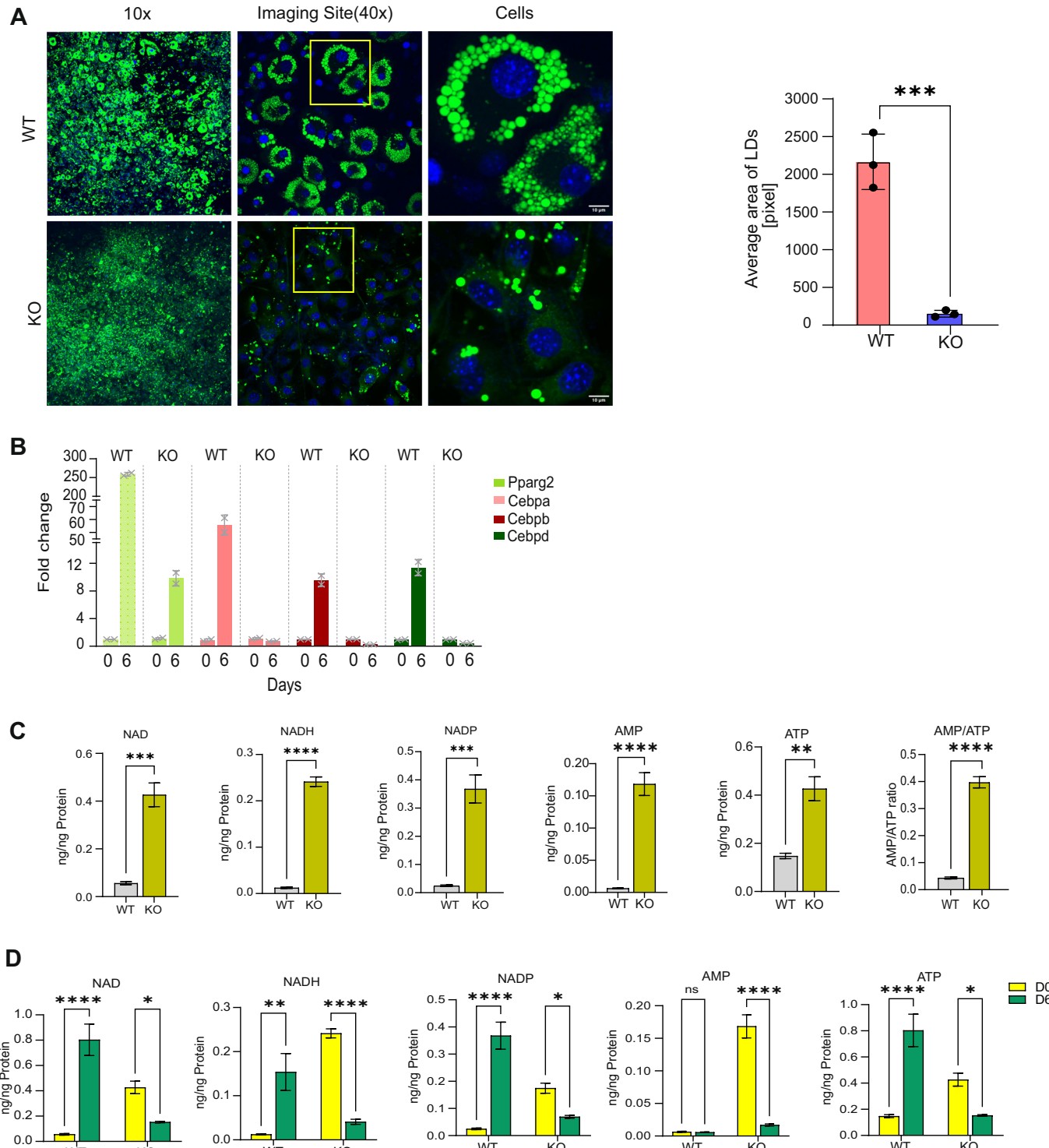

participates in the mitochondrial information processing system (MIPS) (Picard and Shirihai, 2022). Since mitochondria are central to cellular metabolism, housing a range of metabolic inputs and outputs, MTCH2 may facilitate the transfer of metabolic signals between the cytoplasm and mitochondria (Fig. 6, left panel).

The metabolomics analysis revealed that MTCH2 knockout results in an imbalance in several metabolic parameters. Imbalance

in the redox cofactors, $NAD^+$, NADH and $NADP^+$, leading to an oxidative environment. $NAD^+$ plays an indispensable role in OXPHOS by acting as a proton acceptor (Pehar et al, 2018). Along with its role in OXPHOS, it also acts as a signaling molecule in many cellular pathways like cell growth (Covarrubias et al, 2021), sirtuin activity (Anderson et al, 2017), and cell differentiation (Barile et al, 2020; Ryu et al, 2018; Sánchez-Ramírez et al, 2022). There

**Figure 5. MTCH2 is critical for adipocyte differentiation.**

(A) NIH3T3L1 cells were differentiated into adipocytes for 6 days in 4-well glass bottom plates. LDs were stained with Bodipy green and nuclei with Hoechst. The well overview is taken at ×10 magnification. One region (marked by a yellow box) was magnified. Right panel: measure of differentiation by quantification of the number of LDs. Results are presented as mean ± SEM (***$P<0.0003$; one-way ANOVA, $n=3$ biological replicates). (B) mRNA level of WT and MTCH2 knockout (KO) cells at day 0 and day 6-post differentiation. Components of the adipogenic program Pparg, Cebpa, Cebpb, and Cebpd. Results are presented as mean ± SEM of one representative out of three independent experiments. Normalization was done by taking geometric mean of three housekeeping genes, Importin, Tubulin and AcTH. (C) Levels of $NAD^+$, $NADH^+$, $NADP^+$, AMP, ATP (and AMP/ATP ratio) in undifferentiated WT and MTCH2 KO preadipocyte NIH3T3L1. Results are presented as mean ± SEM (**$P<0.001$, ***$P<0.0003$, ****$P<0.0007$, unpaired $t$ test, $n=4$ biological replicates). (D) Levels of $NAD^+$, $NADH^+$, $NADP^+$, AMP, ATP in WT and MTCH2 KO preadipocyte at day 0 and day 6-post differentiation. Results are presented as mean ± SEM (ns, non-significant, *$P<0.05$, **$P<0.001$, ****$P<0.0007$, Two-way ANOVA with sidak's multiple comparison test, $n=4$ biological replicates). Source data are available online for this figure.

is an intimate connection between energy metabolism and redox cofactors (Yang and Sauve, 2016), and they appear on the front defense line in incidents of mitochondrial insult (Stein and Imai, 2012), cellular stress (Xiao et al, 2018), nutrient depletion/starvation (Yang and Sauve, 2016), and high energy demand like during exercise (Cantó et al, 2015). There was also an imbalance in adenine nucleotides in the MKO cells, which resulted in an increase in the ADP/ATP ratio, representing an increase in energy demand.

Additionally, MKO cells showed a reduction in numerous metabolites, including carbohydrates, lipids, proteins, and amino acids, along with an increase in glucose uptake, suggesting enhanced metabolic and catabolic activity to meet higher energy demands. There was also an imbalance in metabolites associated with the urea cycle, such as citrulline and ornithine. Normally, the urea cycle removes excess nitrogen by producing urea; however, under nutrient-scarce conditions, cells may downregulate the urea cycle to conserve nitrogen, leading to reduced levels of these metabolites.

It's important to note that urea is produced only in the liver, so HeLa cells—originating from cervical cancer—likely lack a fully functional urea cycle. Therefore, the decrease in citrulline and ornithine may instead be due to increased utilization, which also contributes to the observed decrease in several amino acids. In summary, the absence of MTCH2, a critical metabolic regulator, appears to create a disconnection between cellular energy demand and utilization (Fig. 6, right panel).

Importantly, we found that MTCH2 knockout (MKO) cells display increased basal and maximal oxygen consumption, along with a higher cell respiratory control ratio (RCR). Respirometry on permeabilized HeLa cells further revealed increased state 3 (ADP-induced respiration), state 3u (FCCP-induced respiration), and RCR in MKO cells. This elevated RCR suggests two possibilities: an increase in mitochondrial number or enhanced respiratory capacity per mitochondrion in MKO cells. However, Mito Tracker deep Red (MTDR) staining for mitochondrial mass shows no significant change in MKO cells compared to WT, suggesting that the increased RCR is likely due to greater respiratory capacity per organelle. One possibility is that MTCH2 deletion, by limiting mitophagy, induces the accumulation of some dysfunctional mitochondria that consume ATP, while the other mitochondria inside the same cell compensate ATP wasting by increasing ATP production. Another possibility is that MTCH2 deletion activates some signaling pathways to stop proliferation and glycolysis, which activate catabolism to secondarily activate OXPHOS.

Additionally, MKO cells show a lower NADH redox index compared to WT cells, indicating an elevated respiration rate and accelerated electron consumption in the absence of MTCH2. MKO

cells also have higher levels of mitochondrial NADH, suggesting increased substrate availability, which supports a higher NADH production rate and serves as an indicator of TCA cycle efficiency. Furthermore, MKO cells display an elevated mitochondrial membrane potential ($\Delta\Psi$m), indicative of increased mitochondrial activity.

In HeLa cells, glycolysis and the conversion of pyruvate to lactate are key metabolic pathways for ATP and NADH production. Deletion of MTCH2, however, leads to an increase in mitochondrial oxidative function (OXPHOS) and even promotes fatty acid utilization, as indicated by the inhibition of basal respiration by etomoxir in MKO cells. The shift from a glycolytic to an OXPHOS phenotype in MKO cells may be due to an inhibition of glycolysis to lactate, which forces MKO cells to rely on OXPHOS and fatty acids to meet their energy demands. This is supported by the observed decrease in lactate levels in MKO cells (Fig. 2E).

Together, these findings support the idea that MTCH2 deletion enhances mitochondrial oxidative function, elevates ATP demand, and drives a catabolic program. Interestingly, MTCH2 deletion also reduces HeLa cell proliferation. Given these findings, it is likely that the observed metabolic phenotype—characterized by a progressive decrease in various metabolites over time—is primarily a result of increased mitochondrial oxidative function due to MTCH2 deletion, rather than a byproduct of reduced cell proliferation.

It is well established that preadipocytes require 85–95% confluency and growth arrest as prerequisites for entering the commitment phase for its differentiation (Gregoire et al, 1998). To determine whether the inability of MTCH2 KO preadipocytes to achieve the same confluency as WT cells on day 0 could be the reason for their failure to differentiate into mature adipocytes, we allowed the MTCH2 KO preadipocytes additional time to reach confluency before initiating differentiation. Despite reaching the required confluency, MTCH2 KO preadipocytes still failed to differentiate. Additionally, we observed that it took longer for the MTCH2 KO preadipocytes to enter the growth arrest phase after the introduction of differentiation media, suggesting that these cells might require more time than WT cells, which typically differentiate fully into mature adipocytes within 6 days. To explore this further, we extended the differentiation period of MTCH2 KO NIH3T3L1 cells to 12 days but still found no significant difference between day 6 and day 12. Therefore, we conclude that neither confluency nor extended time in culture facilitates the differentiation or lipid accumulation in MTCH2 KO preadipocytes.

Why does loss of MTCH2 result in an increase in mitochondrial oxidative function? It was recently reported that MTCH2 regulates mitochondrial fusion by modulating the pro-mitochondrial fusion lipid lysophosphatidic acid (LPA) (Labbé et al, 2021). Most

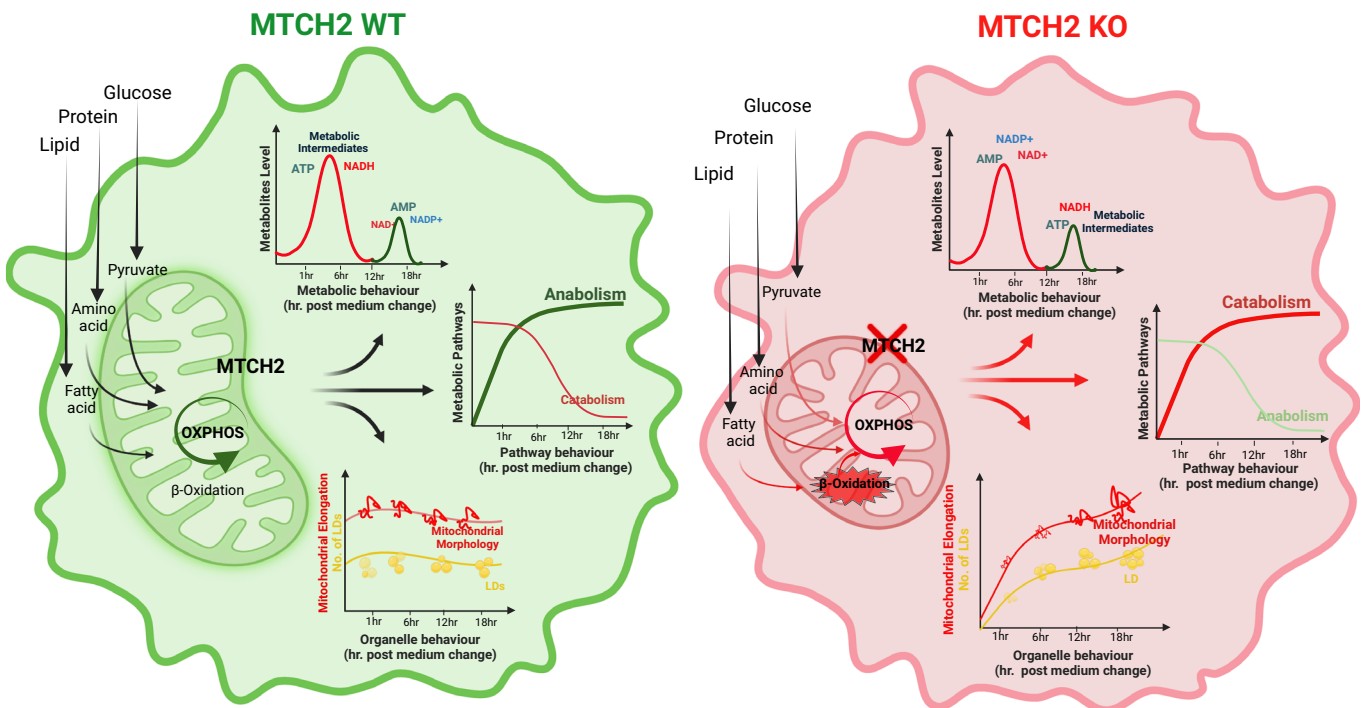

**Figure 6.  Schematic representation comparing the metabolic state of wild type and MTCH2 knockout cells.**

Left panel: In wild type cells, MTCH2 might act as a mitochondrial mediator/sensor by sensing and connecting between metabolic intermediates/pathways and dynamic changes in mitochondria morphology/energy production by receiving and sending signals. Right panel: MTCH2 knockout cells, missing a pivotal mediator/sensor, can lead to a disconnection between the cellular energy demand and the cellular energy utilization (created with BioRender.com).

recently, we reported that MTCH2 cooperates with ER-localized Mitofusin 2 (MFN2) and LPA synthesis at the ER to sustain mitochondrial fusion(Goldman et al, 2024).Thus, MTCH2 may play a role in phospholipid transfer from the ER to mitochondria and therefore regulate phospholipid composition of the mitochondrial membranes. This regulation may in turn affect many functional parameters of the mitochondria, including oxidative function.

Interestingly, MTCH2 was also demonstrated to act as an insertase, which aids tail-anchored mitochondrial proteins to integrate into the OMM (Guna et al, 2022).In addition, insertases can function as scramblases (including MTCH2), which flip phospholipids between the two leaflets of the membrane through their pore region (Li et al, 2024; Bartoš et al, 2024).This activity might be related to MTCH2's role in phospholipid transfer from the ER to mitochondria described above, and thus can further support a role for MTCH2 in regulating mitochondrial oxidative function via regulation of membrane lipid composition. Notably, this putative lipid-modifying activity might be related to MTCH2's apoptotic activity in regulating tBID-induced mitochondrial outer membrane permeabilization (MOMP).

Energy metabolism, redox potential, and efficient nutrient utilization are key determinants of cellular behavior (Mitchell et al, 2018; Covarrubias et al, 2021). Lack of coordination among these systems can lead to abnormal cell growth and development (Cantó et al, 2015), which may explain why MTCH2 knockout in mice results in embryonic lethality at E7.5 (Zaltsman et al, 2010). In

culture, MTCH2 knockout embryonic stem cells may survive at the cost of being smaller in size and growing slower (Bahat et al, 2018). The uncoordinated metabolic transition could also account for the delayed shift from naïve to primed states in MTCH2 knockout embryonic stem cells and the increased quiescence exit in MTCH2 knockout hematopoietic stem cells (Bahat et al, 2018; Maryanovich et al, 2015). In culture, MTCH2 knockout cells appear to adapt by reprogramming their metabolism. This adaptation includes an upregulation of mitochondrial oxidative function and rerouting lipids into storage forms to fuel cellular energy demands rather than producing membrane lipids essential for cell growth and proliferation.

Oxidative metabolism is not an innate cellular behavior but rather an adaptive response to specific stress conditions, such as starvation (Finn and Dice, 2006; Cantó et al, 2010). Given that oxidative metabolism is not the default pathway and that MTCH2 knockout leads to an oxidative environment, conditions become unfavorable for reductive biosynthetic pathways, such as lipid synthesis (Chen et al, 2019). Reductive biosynthesis requires a substantial supply of ATP and reducing cofactors, NADH and NADPH, to proceed efficiently (Chen et al, 2019; Xiao et al, 2018). NIH3T3L1 preadipocytes, a model for white adipocyte (WAT) differentiation, show that MTCH2 knockout preadipocytes fail to mature into adipocytes. This phenotype may be due to: (1) mitochondrial fragmentation, (2) an oxidative environment (elevated NAD+ and NADP+ levels), (3) Mitochondrial Oxidative Function and (4) a shortage of metabolic intermediates.

The dynamic shift of mitochondria from a fragmented to a tubular state, along with the metabolic transition from glycolysis to oxidative phosphorylation (OXPHOS), is essential to meet the increased energy demand for anabolic processes during differentiation (Barile et al, 2020; Sánchez-Ramírez et al, 2022). For a single molecule of palmitic acid, cells require approximately 14 molecules of NADPH, 7 molecules of ATP, and 16 carbons from 8 molecules of acetyl-CoA (Xu et al, 2021). Targeted metabolomics in MTCH2 knockout NIH3T3L1 preadipocytes revealed elevated levels of NAD+, NADP+, and a higher AMP/ATP ratio, indicating an oxidative, low-energy environment unfavorable for differentiation. Additionally, NAD+ in the nucleus serves as a signaling molecule, while increased cytoplasmic NAD+ supports the metabolic intermediates of glucose metabolism during differentiation (Sánchez-Ramírez et al, 2022; Ryu et al, 2018). Our findings show that during differentiation (day 0 to day 6), WT cells increase NAD+ levels, whereas MTCH2 knockout cells show a decrease.

In MTCH2-deleted NIH3T3L1 cells, AMP levels increase, similar to what is observed in HeLa cells, likely due to: (1) alterations in nucleotide metabolism (AMP, ADP, ATP), a common phenotype associated with MTCH2 deletion, and (2) the drive to utilize lipids and other nutrients to support heightened mitochondrial oxidative function. The increased lipolysis (release of fatty acids from stored TAGs) and re-esterification (conversion of fatty acids to fatty acyl-CoA), a process with high ATP demand, may contribute to the observed rise in AMP levels in NIH3T3L1 cells (Miyoshi et al, 2008; Sharma et al, 2023).

In summary, our findings demonstrate that MTCH2 knockout induces a hypermetabolic state, leading to an imbalance in cellular energy flow and activating multiple metabolic pathways to meet the heightened energy utilization and demand. These results underscore MTCH2's role as a crucial regulator of cellular energy flow.

# Methods

### Reagents and tools table

| Reagent/resource | Reference or source | Identifier or catalog number |
|---|---|---|
| **Experimental models** | | |
| Mouse NIH3T3-L1 cells | ATCC | |
| Human HeLa cells | ATCC | |
| **Recombinant DNA** | | |
| pKLV-U6gRNA (BbsI)-PGKpuro2ABFP | Addgene | Cat #50946 |
| pCas9_GFP | Addgene | Cat#44719 |
| pBABE-puro | Addgene | Cat#1764 |
| gag, pol genes | Addgene | Cat#8449 |
| vsv-g gene | Addgene | Cat#8454 |
| pBaBe-MTCH2 | This paper | N/A |
| **Antibodies** | | |
| Anti-mouse MTCH2 | Grinberg et al, 2005 | |
| Anti-human MTCH2 | Abcam | ab227926 |
| Anti-citrate synthase | Cell Signaling | Cat#14309 |

| Reagent/resource | Reference or source | Identifier or catalog number |
|---|---|---|
| Anti-AMPK | Cell Signaling | Cat #2532 |
| Anti-phospho-AMPK (Thr172) | Cell Signaling | Cat #2535 |
| Anti-β-actin | Sigma-Aldrich | Cat #A5441 |
| Anti-ASCL1 | Cell Signaling | Cat #4047 |
| Anti-SCD1 | Cell Signaling | Cat #2794 |
| Amersham ECL HRP-Conjugated Antibodies | Cytiva | Cat #NA934 |
| Peroxidase AffiniPure Donkey Anti-Mouse IgG(H+L) | Jackson Immuno Research Laboratory (JIR) | Cat #715-035-151 |
| **Oligonucleotides and other sequence-based reagents** | | |
| All primers used in the paper are listed in Table EV2 | Sigma | |
| **Chemicals, enzymes and other reagents** | | |
| Fetal bovine serum (FBS) | Gibco | Cat. # 12657 |
| Fetal calf serum (FCS) | Biological Industries | Cat. # 04-102-1A |
| Sodium pyruvate | Biological Industries | Cat. # 03-042 |
| DMEM | Gibco | Cat# 41965 |
| DMEM with 25 mM glucose (4.5 g/L) | Gibco | Cat #41965 |
| DMEM 6.25 mM glucose (1 g/L) | Corning | Cat # 10-014-CM |
| DMEM without L-glutamine | Capricorn | Cat # DMEM-HPXA |
| Opti-MEM™ | Gibco | Cat#11058021 |
| HBSS, Hank's Balancing Salt Solution | Biological Industries | Cat # 02-015 |
| Phosphate-buffered saline solution (PBS) | Thermo-Fisher Scientific | Cat#D2650 |
| DMSO | Sigma-Aldrich | |
| Bovine Insulin | Sigma-Aldrich | Cat#I0516 |
| Dexamethasone | Sigma-Aldrich | Cat#D4902 |
| 3-Isobutyl-1-methylxanthine (IBMX) | Sigma-Aldrich | Cat#I5879 |
| Oligomycin | Sigma-Aldrich | Cat# O4876 |
| FCCP Carbonyl cyanide 4-(trifluoromethoxy) phenylhydrazone | Sigma-Aldrich | Cat# C2920 |
| Rotenone | Sigma-Aldrich | Cat# R8875 |
| Antimycin A | Sigma-Aldrich | Cat# A8674 |
| UK-5099 | Sigma-Aldrich | Cat# 5.04817 |
| BPTES | Sigma-Aldrich | Cat# SML0601 |
| Etomoxir | Sigma-Aldrich | Cat# 5.09455 |
| ADP | Sigma-Aldrich | Cat# A2754 |
| Succinate | Sigma-Aldrich | Cat# 398055 |
| XF Plasma Membrane Permeabilizer | Agilent | Cat#102504-100 |
| KCN | Sigma-Aldrich | Cat# 207810 |
| Puromycin | Gibco | Cat#A1113803 |

| Reagent/resource | Reference or source | Identifier or catalog number |
|---|---|---|
| Triton X-100 | Thermo-Fisher Scientific | Cat# 28314 |
| cyQUANT,CyQUANT™ Direct Cell Proliferation Assay | Invitrogen | Cat# C35011 |
| Hoechst | Invitrogen | Cat# 33342 |
| BODIPY 493/503 | Thermo Scientific | Cat# D3922 |
| 2-NBDG | Invitrogen | Cat # N13195 |
| **Chemicals, enzymes and other reagents** | | |
| FCCP | Sigma-Aldrich | Cat # C2920 |
| MitoSOX (Molecular Probes, Life Technologies) | Invitrogen | Cat # M36008 |
| Tetramethylrhodamine, Methyl Ester, Perchlorate (TMRM) | Invitrogen | Cat# T668 |
| MitoTracker™ Deep Red FM | Invitrogen | Cat# M22426 |
| **Software** | | |
| GraphPad Prism 10.2.3 | | |
| Fiji ImageJ | Schindelin et al, 2012 | |
| Ilastik | Berg et al, 2019 | |
| StarDist | Schmidt et al, 2018 | |
| CellPose | Stringer et al, 2021 | |
| BioRender.com | Bio Render | |
| FlowJo software | BD Biosciences | |
| Agilent Seahorse XF Wave | Agilent | |
| **Other** | | |
| RadianceQ Chemiluminescent substrate Quantitative Western | Azure Biosystems | Cat# AC2101 |
| Bio-Rad Protein Assay Dye Reagent Concentrate | BIO-RAD | Cat# 5000006 |
| DNeasy Blood & Tissue Kit | QIAGEN | Cat# 69505 |
| NucleoSpin RNA II Mini Spin Kit | Macherey-Nagel | Cat #740955.50 |
| High-Capacity cDNA Reverse Transcription Kit | Applied Biosystems | Cat. # 4368814 |
| Fast SYBR Green Master Mix | Applied Biosystems | Cat. # 4385614 |
| Pierce 660nm Protein Assay Reagent | Thermo-Fisher Scientific | Cat. # 22660 |
| Seahorse Bioscience XF96 platform (Agilent Technologies, USA) | Agilent | |
| Infinite M200PRO (TECAN) plate reader | TECAN | |
| Medical X-ray Processor | Carestream | |
| StepOnePlus_Real-Time PCR System | Applied Biosystems | |
| ZE5 Cell Analyzer | BIO-RAD | |
| Nikon ECLIPSE Ti2-E inverted microscope with a CSW-1 spinning disc system (Yokogawa) | Nikon | |

## Experimental model and subject details

### Cell lines

**HeLa cells**: HeLa cells were routinely cultured in Complete Media (CM), composed of Dulbecco's Modified Eagle's Medium (DMEM) with 4.5 g/L glucose and L-glutamine (Gibco, cat. #41965), supplemented with 1 mM sodium pyruvate (Biological Industries, cat. #03-042) and 10% fetal bovine serum (FBS; Gibco, cat. #12657), maintained at 37 °C in a 5% $CO_2$ atmosphere. For nutrient depletion experiments, cells were grown in Hank's Balanced Salt Solution (HBSS; Biological Industries, cat. #02-015).

**NIH3T3L1 cells:** NIH3T3L1 preadipocytes (ATCC) were cultured in Dulbecco's Modified Eagle's Medium (DMEM) with 4.5 g/L glucose and L-glutamine (Gibco, cat. #41965), supplemented with 1 mM sodium pyruvate (Biological Industries, cat. #03-042) and 10% fetal calf serum (FCS; Biological Industries, cat. #04-102-1A), maintained at 37 °C in a 5% $CO_2$ atmosphere. Cells were grown in 10 cm tissue culture dishes, with the medium changed every alternative days. Preadipocytes were maintained at densities below 70% confluence, except during differentiation (details provided below).

## Preparation of adipocyte differentiation medium

The differentiation medium for NIH3T3L1 preadipocytes was prepared using Dulbecco's Modified Eagle's Medium (DMEM; Gibco, Cat. #41965) supplemented with 10% fetal bovine serum (FBS; Gibco, Cat. #12657). To induce differentiation, specific reagents were added: 500 μM 3-Isobutyl-1-methylxanthine (IBMX; Sigma-Aldrich, Cat. #I5879), 1 μM dexamethasone (Sigma-Aldrich, Cat. #D4902), and 1 μg/mL bovine insulin (Sigma-Aldrich, Cat. #I0516). For each experiment, IBMX was freshly prepared as a 50 mM stock solution in DMSO (Sigma-Aldrich, Cat. #D2650) at a 100× concentration. Bovine insulin was prepared as a 10,000× stock solution at a concentration of 10 mg/mL, following the manufacturer's specifications, and dexamethasone was prepared as a 10 mM stock in phosphate-buffered saline (PBS; Thermo-Fisher) at a 10,000× concentration. Prior to use, these stock solutions were diluted in DMEM with 10% FBS to achieve final concentrations: IBMX at 1:100, insulin at 1:10,000, and dexamethasone at 1:1000. All solutions were then sterile-filtered to maintain aseptic conditions throughout the experiments.

## Adipocyte differentiation in 4-well glass bottom imaging plate

To initiate adipocyte differentiation, NIH3T3L1 pre-adipocytes were seeded at a density of 10,000 cells per well in a 4-well glass bottom imaging plate. The cells were cultured for 2 days until they reached full confluence, with the medium replaced with fresh DMEM containing 10% fetal calf serum (FCS). After 48 h, differentiation was induced using the differentiation medium as previously described. On the 6th day of differentiation, the cells were stained with BODIPY 493/503 (Thermo Scientific, Cat. #D3922) for lipid detection and MitoTracker Deep Red (MTDR; Invitrogen, Cat. #M22426) for mitochondrial visualization, then prepared for live imaging.

## Adipocyte differentiation in 6 well plate

NIH3T3L1 pre-adipocytes were seeded at a density of 60,000 cells per well in 6-well plates and cultured for 2 days until they achieved complete confluence. The culture medium was refreshed with fresh DMEM supplemented with 10% FCS. After 48 h, the cells underwent differentiation using the same protocol employed for 10 cm tissue culture dishes, as described below.

## Adipocyte differentiation in 10 cm tissue culture plate

The cells were maintained following the above procedure. To initiate differentiation, cells were cultured until they reached complete confluence. Once the cells achieved 90–95% confluence, the medium was replaced with the differentiation medium, prepared as previously described, marking the start of differentiation (day 0). After 2 days, this medium was switched to DMEM with 10% FBS supplemented with 1 μg/mL insulin. On day 4, the medium was changed again to DMEM with 10% FBS. Finally, on day 6 post-differentiation initiation, cells were collected for various analyses.

## Generation of MTCH2 knockout stable cell lines

***Generation of MTCH2 knockout cells using the CRISPR-Cas9 system***
The MTCH2 CRISPR knockout (KO) cell line was generated in both HeLa and NIH3T3L1 cells. Guide RNAs (gRNAs) targeting MTCH2 were designed using the CHOP web application (https://chopchop.cbu.uib.no/), with the following Guide RNA sequences:F-CACCAGCACTTTCACGTACATGAGGT and R-TAAAACCT CATGTACGTGAAAGTGCT. These gRNAs were cloned into the pKLV-U6gRNA (BbsI)-PGKpuro2ABFP vector (Plasmid #50946). To create the CRISPR-Cas9 knockout cell lines, HeLa and NIH3T3L1 cells were co-transfected with two plasmids: (1) the gRNA-containing plasmid and (2) GFP-Cas9 (pCas9_GFP, Plasmid #44719). As a control, a CRISPR-Cas9 control cell line was also generated using the same CRISPR-Cas9 construct but without the gRNA. Transfected cells were selected by culturing in DMEM with 10% FBS and 2 μg/mL puromycin for 2 weeks to ensure stable integration.

## Plasmid transfection of HeLa and NIH3T3L1 cells

For transfection, HeLa and NIH3T3L1 cells were seeded in 6-well plates with DMEM and 10% FBS, and cultured to 80% confluence. Transfection was carried out using the Lipofectamine 3000 reagent (Thermo-Fisher) according to the manufacturer's instructions. Both the transfection reagent and DNA were pre-diluted in Opti-MEM medium (Cat# 11058021, Opti-MEM™; Gibco). The DNA-transfection reagent complex was allowed to form for 15 min at room temperature, then added dropwise to the culture medium. The cells were incubated with the transfection mix for 6 h at 37 °C in DMEM/10% FBS. Following this, the medium was replaced with fresh DMEM/10% FBS. The next day, the medium was changed to DMEM/10% FBS supplemented with puromycin (2 μg/ml) for 2 weeks of selection.

## Generation of MKO-R cells by expressing MTCH2 in MKO cells using a retroviral system

To generate HeLa cells with stable MTCH2 expression, we utilized a pBaBe-based retroviral construct for gene transfer. The MTCH2

gene was subcloned into the ENTRY vector (pBaBe; Addgene) using TOPO cloning. Stable cell pools were established by selecting positively transfected cells with 2 μg/mL puromycin for 2 weeks.

## Production of retroviruses for gene transfer

The retrovirus for gene expression was generated in HEK293T cells, which were maintained in DMEM with 10% FBS and supplemented with sodium pyruvate. The cells were cultured in 6 cm dishes until they reached 70% confluence. Transfection of HEK293T cells was carried out using PolyJet™ (PolyJet™ In Vitro DNA Transfection Reagent; Cat #SL100688). Cells were co-transfected with three plasmids: 1 μg of a plasmid containing the expression construct (pBaBe-MTCH2), 1 μg of a plasmid containing the gag and pol genes (Addgene plasmid no. 8449), and 1 μg of the vsv-g gene plasmid, which produces the envelope for the expressing plasmid (Addgene plasmid no. 8454). The plasmids were mixed in double-distilled water (ddH2O) to a final volume of 200 μL. After incubating for 30 min at room temperature, the transfection mix was added to the cells. The medium was changed after 12 h and replaced with 4 mL of DMEM with 10% FBS. The cells were then incubated for 24 h. The media was collected at 24 and 48 h post-transfection. Each fraction was filtered through a 0.45 μm sterile filter and stored at 4 °C until use. To concentrate the virus, the collected fractions were combined and centrifuged using an ultrafiltration centricon (Amicon "Ultra – 15 centrifugal filter" with a 100 kDa MW cutoff) for 45 min at 4 °C. The filtrate was then resuspended in a total volume of 1 mL (DMEM, 10% FBS), which was used to infect the HeLa MTCH2 KO (MKO) cells.

## Quantification of the adipocyte differentiation efficiency

Image processing and analysis were performed using FIJI (https://fiji.sc/). Lipid droplet (LD) size and number were quantified using the ImageJ "Analyze Particles" function on thresholded images. The settings for size (square pixels) were set from 0.1 to 100, and circularity was set from 0 to 1. The average area of LDs was analyzed. Data were presented as means ± SEM. Statistical analysis among groups was conducted using a Student's *t* test.

## Quantitative real-time PCR

Samples were collected from HeLa cells and on Day 0 and Day 7 of differentiation from NIH3T3L1 cells. RNA was extracted using the NucleoSpin RNA kit (Macherey-Nagel #740955) according to the manufacturer's instructions. A 1 μg RNA sample from each group was used for cDNA synthesis with the High-Capacity cDNA Reverse Transcription Kit (Cat. # 4368814, Applied Biosystems). Quantitative PCR (qPCR) was performed using 0.4 ng/μl of cDNA and 0.5 μM of each primer. The primer sequences are listed in Table EV2.

## Fluorescence microscopy

***Live imaging***
For live cell imaging experiments, HeLa cells were seeded the day before the experiment. The media was replaced with either fresh Complete Media (CM) or Hank's Balanced Salt Solution (HBSS) at time=0. The cells were pre-incubated with 100 nM Mitotracker

Deep Red (MTDR; cat. # M22426) for mitochondrial staining, 1 nM Hoechst (Cat# 33342; Invitrogen) for nuclear staining, and 1 µg BODIPY 493/503 (Cat# D3922; Thermo Scientific) for lipid droplet (LD) staining for 30 min. Since the goal was to observe changes over time, the same media was maintained until imaging was complete. After staining, cells were stabilized for an additional 30 min at 37 °C and 5% $CO_2$. Cells were then imaged using a Nikon ECLIPSE Ti2-E inverted microscope with a CSW-1 spinning disc system (Yokogawa) and a ×100 CFI Plan Apo100× oil objective (na 1.45, wd 0.13 mm), equipped with temperature and CO2 control. The cells were incubated at 37 °C in a 5% $CO_2$ humidified chamber, and images were captured at 0, 1, 12, 20, and 30 h for CM conditions, and at 0, 2, 4, 8, and 12 h for HBSS conditions.

## Automated image analysis of live imaging experiments with HeLa cells

All images were analyzed with open-source software we used Fiji (Schindelin et al, 2012), StarDist (Schmidt et al, 2018), Ilastik (Berg et al, 2019) and Cellpose (Stringer et al, 2021). Below we describe the main steps taken.

### Single cell segmentation by cellpose
To identify individual cells in the image we trained a Cellpose model using both the mitochondria and Nucleus channel. The training was done on representative images of different conditions, from both WT and MKO groups. We then dilated the identified cells as to include the cell's membrane.

### LD segmentation and clustering by StarDist
To identify LDs, we used an out-of-the-box StarDist model and then filtered LDs having low mean intensity (the same threshold was used for WT and MKO cells). For quantitating LD clustering, we used Fiji's SSIDC cluster indicator plugin.

### Pixel-based mitochondria segmentation by Ilastik
To segment mitochondria, we trained an Ilastik model using representative images from both WT and MKO cell groups from different conditions.

### Calculating mitochondria aspect ratio
We define the Aspect ratio (AR) of the mitochondria of each cell to be the ratio between the mitochondria mean length and the mitochondria mean width. The mean width was taken to be the mean local-thickness of the mitochondria skeleton (using Fiji's "Local Thickness" and "Skeletonize" plugins, respectively), the mean length was taken to be the mitochondria mean area divided by the mean width (mean area was calculated to be the mitochondria total area divided by the number of mitochondria fragments—small fragments were ignored). Finaly, we multiplied the result by π/4 so that circular mitochondria would get AR of around 1.

### Cell categorization
For each cell identified in each image, we exported the associated lipid droplet (LD) and mitochondrial data to an Excel spreadsheet. This spreadsheet was then used to plot aspect ratio (AR) for mitochondrial length, number of LDs per cell, LD size and the degree of LD clustering.

## Representative images

Representative images were processed using FIJI (https://fiji.sc/). The merged images from all three channels—nuclei (Hoechst), mitochondria (MTDR), and lipid droplets (BODIPY)—were cropped to focus on specific details. The regions of interest (ROIs) were standardized in size, and the same setting parameters were uniformly applied across all images for each channel to highlight the fluorescent structures without altering the raw pixel values. The images were then converted to RGB format and imported into Adobe Illustrator for further processing.

## Glucose uptake assay

Glucose uptake was measured using the fluorescent probe 2-NBDG (Cat # N13195, Invitrogen). HeLa WT, MTCH2 KO (MKO), and MTCH2 KO-Rescue (MKO-R) stable cells were seeded at 25,000 cells in a 6-well plate, a day before the experiment. HeLa cells were incubated with Opti-MEM (Cat#11058021, Opti-MEM™; Gibco) to mimic a low-glucose medium containing 0.5% Serum for 2 h. Cells were treated with 100 µM fluorescent 2-NBDG for 30 min. Next, the culture medium was removed, and the cells were washed with PBS. The fluorescence intensity was measured using the ZE5 flow cytometer, and data analysis was done using the FlowJo software.

## NADH autofluorescence

Mitochondrial NADH levels were estimated as previously described (Plun-Favreau et al, 2012). Briefly, mitochondrial NADH was calculated by measuring the difference in arbitrary units between the maximum NADH autofluorescence, achieved by adding 1 mM potassium cyanide (KCN; Sigma-Andrich), a potent complex IV inhibitor that blocks mitochondrial respiration and leads to maximum NADH accumulation, and the minimum NADH autofluorescence, achieved by adding 0.5 mM FCCP (Sigma-Andrich), an OXPHOS uncoupler that induces maximum respiration and depletes the mitochondrial NADH pool. The NADH redox index was estimated by normalizing the initial NADH autofluorescence to 0% and the maximum NADH autofluorescence to 100%. NADH autofluorescence was measured using a flow cytometer, with excitation by a UV laser and a main emission peak at 470 nm.

## Mitochondrial ROS measurement

ROS levels, specifically superoxide anion, were assessed by incubating HeLa cells with 5 µM MitoSOX (Molecular Probes, Life Technologies) for 20 min at 37 °C to measure mitochondrial ROS. After incubation, the cells were washed once with PBS, and MitoSOX fluorescence was measured using a flow cytometer.

## Mitochondrial membrane potential

HeLa cells were incubated with 50nM Tetramethylrhodamine, Methyl Ester, Perchlorate (TMRM) (Cat# T668; Invitrogen) for 20 min at 37 °C, followed by a single wash with PBS. TMRM fluorescence was then measured using a flow cytometer. To estimate mitochondrial mass, 100 nM MTDR (Cat# M22426,

MitoTracker™ Deep Red FM) was added alongside TMRM during the incubation.

## Respirometry assays

### Oxygen consumption of intact cells

Oxygen consumption of intact HeLa cells was assessed using the Seahorse Bioscience XF96 platform (Agilent Technologies, USA), as previously described (Ferrick et al, 2008; Taddeo et al, 2018). HeLa cells were seeded at a density of 20,000 cells per well in a Seahorse microplate one day before the experiment. On the day of the experiment, the cells were washed twice and incubated with freshly prepared XF assay medium (Seahorse XF Base Medium + 2 mM L-glutamine + 1 mM pyruvate + 10 mM glucose) for 1 h. Respiration was measured under basal conditions, followed by the addition of oligomycin (ATP synthase inhibitor, 1.5 µM; Cat# O4876, Sigma-Aldrich) and FCCP (Carbonyl cyanide 4-(trifluoromethoxy) phenylhydrazone, 0.5 µM; Cat# C2920, Sigma-Aldrich), which stimulates the electron transport chain to reveal the maximal oxygen consumption rate (OCR). Respiration was then halted by the addition of electron transport chain inhibitors rotenone (0.5 µM; Cat# R8875, Sigma-Aldrich) and antimycin A (0.5 µM; Cat# A8674, Sigma-Aldrich).

For the Substrate Oxidation Test, OCR was measured in intact cells following the same pre-incubation with XF assay medium. The following compounds were injected in sequence: oligomycin (1.5 µM), FCCP (0.5 µM), and rotenone (0.5 µM) + antimycin A (0.5 µM). For nutrient-dependent respirometry analysis, we applied UK5099 (5 µM), BPTES (3 µM), and etomoxir (4 µM). OCR values were normalized to the cell count per well. Cell count was determined by staining the nuclei with cyQUANT (10 µg/mL; Cat# C35011, CyQUANT™ Direct Cell Proliferation Assay) and quantifying them using the Infinite M200PRO (TECAN) plate reader. Fluorescence intensity was measured with the Infinite M200PRO (TECAN) after excitation at 480 nm and emission at 520 nm.

### Oxygen consumption of permeabilized cells

#### Respiratory control ratio (RCR) measurement

Cells were seeded into a Seahorse XF96 microplate a day before the assay and maintained in a tissue culture incubator (37 °C in 5% CO$_2$). The respirometry assay was performed in MAS buffer, consisting of 210 mM sucrose, 660 mM mannitol, 30 mM KH$_2$PO$_4$, 15 mM MgCl$_2$, 6 mM HEPES, 3 mM EGTA, and 0.6% (w/v) fatty acid-free BSA. To permeabilize the cells, 1-2 nM of XF plasma membrane permeabilizer (Cat# 102504-100; XF Plasma Membrane Permeabilizer; Agilent) was used. Succinate (5 mM) and rotenone (4 µM), a complex II substrate and complex I inhibitor, were added to measure ADP-phosphorylation respiration. Following state 2 (Initial OCR measurement) measurements, injections included the following - port A: ADP (4 mM); port B: oligomycin (3 µM); port C: FCCP (3 µM); port D: rotenone (3 µM) + antimycin (3 µM). We calculated State 3, State 3u, and State 4 and mitochondrial Respiratory Control Ratio (RCR). The oxygen consumption rate (OCR) was normalized to the cell count per well, which was determined by staining the nuclei with cyQUANT (10 µg/mL; Cat# C35011, CyQUANT™ Direct Cell Proliferation Assay) and quantifying fluorescence intensity using an Infinite M200PRO (TECAN) plate reader. Fluorescence was measured spectrometrically after excitation at 480 nm and with a primary emission peak at 520 nm.

Mitochondrial Respiratory Control Ratio (RCR) was measured by the following equation:

State 3 Respiration = OCR following ADP injection.
State 4 Respiration = OCR following Oligomycin injection.
State 3u Respiration = OCR following FCCP injection.
RCR = (State3)/(State4) or (State3u/State4).

## Growth curve

Growth curve pattern analysis was adapted from methods previously described (Han et al, 2023). Briefly, cells were cultured in complete media (CM) as well as under four additional conditions. The specific media conditions were as follows:

1. Complete media (CM): DMEM with 25 mM glucose (4.5 g/L) (cat #41965, Gibco) +10% FBS (CM refers to the media routinely used for maintaining the cells).
2. Low glucose medium: DMEM with 12.5 mM glucose + 10% FBS. We mixed DMEM 25 mM glucose (4.5 g/L) (cat #41965, Gibco) with DMEM 6.25 mM glucose (1g/L) (cat # 10-014-CM, Corning) with the ratio of 1:2 to reach the glucose concentration of 12.5 mM.
3. No glutamine medium: DMEM without L-glutamine (cat # DMEM-HPXA, Capricorn) +10% FBS.
4. No free fatty acids medium: DMEM with 25 mM glucose (4.5 g/L) (cat #41965, Gibco) + 10% Delipidated FBS. Delipidation of serum was done according to a previously reported protocol (Brovkovych et al, 2019).

## Western blot analysis

Whole Cell lysates were prepared in RIPA buffer and then diluted in Laemmli sample buffer (100 mM Tris–HCl, 2% SDS, 10% glycerol, 0.1% bromophenol blue) containing 5% β-mercaptoethanol. After heating at 95 °C for 5 min, proteins were separated by SDS–PAGE and transferred onto PVDF membranes. Membranes were blocked with 5% non-fat milk or 5% BSA and detection of individual proteins was carried out by blotting with the following specific primary antibodies: anti-mouse MTCH2 (Grinberg et al, 2005), (1:1500); anti-human MTCH2 (ab227926, Abcam,1:1000); anti-citrate synthase (CS; #14309, Cell Signaling, 1:1000); anti-AMPK (#2532, Cell Signaling, 1:1000); anti-phospho-AMPK (Thr172; #2535, Cell Signaling, 1:1000); anti-β-actin (A5441, Sigma-Aldrich, 1:10,000); anti-ASCL1 (#4047, Cell Signaling, 1:1000); anti-SCD1(#2794, Cell Signaling, 1:1000). Proteins of interest were detected by chemiluminescence using a secondary peroxidase-linked anti-rabbit (1:10,000) or anti-mouse (1:10,000) antibody.

## LC-MS based targeted metabolomics

Frozen cell lysates were aliquoted and extracted in organic extraction solvents for targeted LC/MS metabolomics (acylcarnitines, amino acids, organic acids, nucleotides, and malonyl and acetyl CoA) according to validated, optimized protocols in our previously published studies (Lanfear et al, 2017; Hahn et al, 2023). These protocols use cold conditions and solvents to arrest cellular metabolism and maximize the stability and extraction recovery of metabolites. Each class of metabolites was separated with a unique HPLC method to optimize their chromatographic resolution and

sensitivity. Quantitation of metabolites in each assay module was achieved using multiple reaction monitoring of calibration solutions and study samples on an Agilent 1290 Infinity UHPLC/6495 triple quadrupole mass spectrometer (Lanfear et al, 2017; Hahn et al, 2023). Raw data was processed using Mass Hunter quantitative analysis software (Agilent). Calibration curves ($R^2 = 0.99$ or greater) are either fitted with a linear or a quadratic curve with a 1/X or 1/X2 weighting.

## Non-targeted (Global) metabolomics

### Metabolite extraction

Extraction and analysis of polar metabolites were performed as previously described (Zheng et al, 2015; Malitsky et al, 2016) with a few modifications: Samples were lyophilized and extracted with 1ml of a pre-cooled ($-20$ °C) homogenous methanol: methyl-tert-butyl-ether (MTBE) (1:3, v/v) mixture. The tubes were vortexed and then sonicated for 30 min in an ice-cold sonication bath (taken for a brief vortex every 10 min). Then, DDW: methanol (3:1, v/v) solution (0.5 ml), containing internal following standards: C13 and N15 labeled amino acids standard mix (Sigma, 767964) (1:500), was added to the tubes followed by vortex and centrifugation. The upper organic phase was removed, and the lower polar phase was re-extracted as described above, with 0.5ml of MTBE, moved to a new Eppendorf tube, dried in speed vac, and stored at $-80$ °C until analysis. For analysis, the polar dry samples were re-suspended in 150 µl methanol: DDW (50:50) and centrifuged twice to remove the debris. 125 µl were transferred to the HPLC vials for injection.

## LC-MS polar metabolite analysis

Metabolic profiling of the polar phase was done as described (Zheng et al, 2015), with minor modifications. Briefly, analysis was performed using Acquity I class UPLC System combined with mass spectrometer Q Exactive Plus Orbitrap™ (Thermo Fisher Scientific) operated in a negative ionization mode. The LC separation was done using the SeQuant Zic-pHilic (150 mm × 2.1 mm) with the SeQuant guard column (20 mm × 2.1 mm) (Merck). The Mobile Phase B: acetonitrile and Mobile Phase A: 20 mM ammonium carbonate with 0.1% ammonia hydroxide in water: acetonitrile (80:20, v/v). The flow rate was kept at 200 µl* min$^{-1}$, and the gradient was as follows: 0–2 min 75% of B, 14 min 25% of B, 18 min 25% of B, 19 min 75% of B, for 4 min, 23 min 75% of B.

## Polar metabolites data analysis

The data was processed using Progenesis QI (Waters) when detected compounds were identified by accurate mass, retention time, isotope pattern, and fragments and verified using an in-house-generated mass spectra library.

## Shotgun lipidomics

Lipid species were analyzed using multidimensional mass spectrometry-based shotgun lipidomic analysis (Wang and Han, 2014). In brief, each cell sample homogenate containing 0.5 mg of protein, which was determined with a Pierce BCA assay was accurately transferred to a disposable glass culture test tube. A premixture of lipid internal standards (IS) was added prior to conducting lipid extraction for quantification of the targeted lipid species. Lipid extraction was performed using a modified Bligh and Dyer procedure (Wang and Han, 2014), and each lipid extract was reconstituted in chloroform:methanol (1:1, v:v) at a volume of 400 µl/mg protein. Phosphoethanolamine (PE), cholesterol (CHL), free fatty acid (FFA) and diacylglycerol (DAG) were derivatized as described previously (Han et al, 2005; Wang et al, 2013, 2014, 2015) before lipidomic analysis. Lysophospholipids (LPA, LPG, LPI and LPS) in water phase were enriched using HybridSPE cartridge, after washing with methanol, the lysophospholipids were eluted with methanol/ammonia hydroxide (9:1 and 8:2), dried and reconstituted in methanol for lipidomic analysis (Wang et al, 2015).

For shotgun lipidomics, lipid extract was further diluted to a final concentration of ~500 fmol total lipids per µl. Mass spectrometric analysis was performed on a triple quadrupole mass spectrometer (TSQ Altis, Thermo Fisher Scientific, San Jose, CA) and a Q Exactive mass spectrometer (Thermo Scientific, San Jose, CA), both of which were equipped with an automated nanospray device (TriVersa NanoMate, Advion Bioscience Ltd., Ithaca, NY) as described (Han et al, 2008). Identification and quantification of lipid species were performed using an automated software program (Wang et al, 2016; Yang et al, 2009).Data processing (e.g., ion peak selection, baseline correction, data transfer, peak intensity comparison and quantitation) was performed as described (Yang et al, 2009). The results were normalized to the protein content (nmol lipid/mg protein).

## Targeted metabolomics for ATP, AMP, NAD$^+$, NADH, NADP$^+$ in NIH3T3L1 cells

### Materials

Adenosine 5'-triphosphate (ATP), adenosine 5'-diphosphate (ADP), adenosine 5'-monophosphate (AMP), β-Nicotinamide adenine dinucleotide (NAD$^+$), β-Nicotinamide adenine dinucleotide reduced (NADH), β-Nicotinamide adenine dinucleotide phosphate (NADP), $^{13}$C$_{10}$-adenosine 5'-triphosphate ($^{13}$C$_{10}$-ATP), $^{15}$N$_5$-adenosine 5'-monophosphate ($^{15}$N$_5$-AMP), and amino acid internal standard mix - all were purchased from Merck.

## Sample preparation

Dried pellet of 20 million cells was extracted with 400 µl of 10mM ammonium acetate and 5 mM ammonium bicarbonate buffer, pH 7.7 and 600 µl methanol in bead beater (10 Hz, 1 min; Retsch MM400) and next in shaker (1200 rpm, 30 min; Thermomixer C Eppendorf). Then the extract was centrifuged (19,000×g, 10 min), the supernatant collected and evaporated under reduced pressure. The obtained residue was re-dissolved in 100 µl of 50%-aqueous acetonitrile and placed in LC-MS filter vial (0.2-µm PES, Thomson).

## Liquid chromatography–tandem mass spectrometry (LC-MS/MS)

LC-MS/MS analysis was performed using an instrument consisted of Acquity I-class UPLC system (Waters) and Xevo TQ-S triple quadrupole mass spectrometer (Waters).

### LC

Metabolites were separated on an Atlantis Premier Z-HILIC column (2.1 × 150 mm, 1.7 μm particle size; Waters). Mobile phase consisted of (A) 20% acetonitrile in 20mM ammonium carbonate buffer, pH 9.25 and (B) acetonitrile. Gradient conditions were: 0 to 0.8 min = 80% B; then to 5.6 min gradient (curve 3) to 25% B; 5.6 to 6 min = hold at 25% B; 6 to 6.4 min = back to 80% B. Total run time 9 min. Injection volume was 3 μl, and flow rate was 0.3ml/min.

### MS/MS

Desolvation temperature 400 °C, desolvation gas flow 800 L/h, cone gas flow 150 L/h nebulizer pressure 4 Bar, capillary voltage 2.49 kV, collision gas (argon) flow 0.25 mL/min, source temperature 150 °C. The MRM transitions used were: ATP: 507.9>136 *m/z* at collision 35V and cone voltage 14 V; ADP: 428>136.05 and 428>348.1 *m/z* at collision 25 and 14 V, respectively, and cone voltage 25 V; AMP: 348>96.8, 348>119, and 348>135.7 *m/z* at collision 28, 55, and 20V, respectively, and cone voltage 10V; $NAD^+$: 664>136, 664>428 *m/z* at collision 37 and 27V, respectively, and cone voltage 10 V; NADH: 666>514, 664>649 *m/z* at collision 30 and 20 V, respectively, and cone voltage 10 V; $NADP^+$: 744>508, 664>604 *m/z* at collision 28 and 20 V, respectively, and cone voltage 10 V; GSH: 308>179 *m/z* at collision 17V and cone voltage 10 V; GSSG: 613>355 *m/z* at collision 32 V and cone voltage 10 V. Internal standards: $^{13}C_{10}$-ATP: 518.12>141.07, 518.12>420.1 *m/z* at collision 35 and 18V, respectively, and cone voltage 14 V; $^{15}N_5$-AMP: 353.1>141.1 m/z at collision 20V and cone voltage 17 V. MassLynx and TargetLynx software (v. 4.2 Waters) were applied for quantitative analysis using standard curve in 0.001–10 μg/ml concentration range for each metabolite. $^{13}C_{10}$-ATP and $^{15}N_5$-AMP were added to standards and samples as internal standards to get 0.1 and 0.5 μM, respectively.

## Quantification and statistical analysis

### Metabolomics and lipidomics

Comparisons of compounds or categories between the 3 groups were done using one-way ANOVA on log2-transformed values, followed by Tukey's post-hoc test. All statistics were done in R, v. 4.3.1. Graphs were made in GraphPad Prism 10.2.3. Heat maps and Volcano plot were generated using ggplot 2, v. 3.4.4. Metabolomics and Lipidomics analyses corrected p-value excel file appear in supplementary Dataset EV1 and Table EV2, respectively. All instances where n replicates are reported had n biological replicates.

### Fluorescence microscopy

Fluorescence microscopy images were quantified using Fiji-ImageJ software (Schindelin et al, 2012).NIH3T3L1 Images were thresholded, the area of BODIPY 493/503 stained LDs were quantified from three biological replicates (average of 10–18 cells per replicate), and the mean ± SEM was determined. Statistical significance was evaluated using the Student *t* test with a *P* value <0.05. HeLa cell mitochondrial morphology (aspect ratio (AR)) and number of LDs data were presented as a scattered plot generated in GraphPad Prism Software. Statistical significance was evaluated using the Student one-way ANOVA with a *P* value <0.05.

### RT-qPCR

The mean ± SEM was determined. Statistical significance was evaluated using the Student *t* test and one-way ANOVA with a *P* value < 0.05.

## Data availability

The Fiji macro used to run the analysis, and the Excel template used to categorize the individual cells, together with the trained Ilastik and Cellpose models are deposited and available for download on GitHub. Fiji can be downloaded from https://imagej.net/software/fiji/downloads. It includes ready-to-use Star-Dist and can be configured to run the BioVoxxel plugin that implements the SSCID clustering algorithm; Cellpose can be downloaded from https://github.com/mouseland/cellpose; and Ilastik can be downloaded from https://www.ilastik.org/download. (1) Source Data: https://zenodo.org/uploads/14219451. (2) Code for Image Analysis: https://github.com/WIS-MICC-CellObservatory/Mitochondria-structure-and-Lipid-droplets-clustering.

The source data of this paper are collected in the following database record: biostudies:S-SCDT-10_1038-S44318-024-00335-7.

## Peer review information

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

## Acknowledgements

We thank Dr. Elena Ainbinder from the Stem Cell Core and Advanced Cell Technologies Unit, LCSF, Weizmann Institute of Science, for providing the Seahorse facility to measure cellular respiration (Seahorse XFe96, Agilent Technologies, USA). We also thank the Penn Metabolomics Core (RRID: SCR_022381) in the Cardiovascular Institute at the University of Pennsylvania for metabolomics analyses. We also want to thank Dr. Inna Grosheva for insightful discussions and valuable suggestions. We are grateful to all the members of the Gross lab for their support, insightful discussions, and comments on the manuscript. Also, thank you to Dr. Nishanth Belugali Nataraj for his technical support during certain experiments and thank Dr. Reeba Jacob for her valuable input on image analysis discussions. The studies described in this paper were supported by the Israel Science Foundation (ISF grant # 1562/20 to Atan Gross).

## Author contributions

**Sabita Chourasia**: Conceptualization; Data curation; Software; Formal analysis; Validation; Investigation; Visualization; Methodology; Writing—original draft; Project administration; Writing—review and editing. **Christopher Petucci**: Data curation; Formal analysis; Investigation. **Clarissa Shoffler**: Data curation; Formal analysis; Investigation. **Dina Abbasian**: Data curation; Formal analysis; Investigation. **Hu Wang**: Data curation; Formal analysis; Investigation. **Xianlin Han**: Data curation; Formal analysis; Investigation. **Ehud Sivan**: Data curation; Software; Formal analysis; Investigation. **Alexander Brandis**: Data curation; Formal analysis; Investigation. **Tevie Mehlman**: Data curation; Software; Formal analysis; Investigation. **Sergey Malitsky**: Data curation; Formal analysis; Methodology. **Maxim Itkin**: Data curation; Formal analysis; Investigation. **Ayala Sharp**: Data curation; Formal analysis; Investigation; Methodology. **Ron Rotkopf**: Data curation; Formal analysis; Investigation. **Bareket Dassa**: Data curation; Formal analysis; Investigation. **Limor Regev**: Investigation. **Yehudit Zaltsman**: Investigation; Methodology. **Atan Gross**: Conceptualization; Resources; Supervision; Funding acquisition; Validation; Investigation; Visualization; Writing—original draft; Project administration; Writing—review and editing.

Source data underlying figure panels in this paper may have individual authorship assigned. Where available, figure panel/source data authorship is listed in the following database record: biostudies:S-SCDT-10_1038-S44318-024-00335-7.

## Disclosure and competing interests statement

# Expanded View Figures

**Figure EV1.  Global untargeted metabolomics analysis of WT, MKO, and MKO-R cells.**

(**A**) MTCH2 mRNA expression checked by RT-PCR in all 6 clones of WT and MKO and WT cells. Results are presented as mean ± SEM (****$P<0.0007$, ordinary one-way ANOVA). $N=$Two independent experiments. (**B**) A representative immunoblot of MTCH2 protein level by for MTCH2 expression in all 6 clones of MKO and WT cells. (**C**) A heat map comparing the levels of the top 70 metabolites (out of 107 differential metabolites detected in global metabolomics, all 107 differential metabolites appear in Dataset EV1) in the WT, MKO, and MKO-R cell lines. Metabolite concentration values (Relative abundance) were log 1.5-transformed for statistics. The groups were compared by ANOVA. Values are scaled to Z-scores per row (metabolites); $n=4$ independent biological replicates. (**D**) Pathway enrichment analysis of 107 differential metabolites detected in global metabolomics. Enrichment was detected using a Hypergeometric Test using a Relative-betweenness Certrality topology against the Homo sapiens (KEGG) database, using MetaboAnalyst server ($n=4$ independent biological replicates). (**E**) Number of metabolites detected in significantly enriched pathways with an FDR cutoff<0.12 ($n=4$ independent biological replicates). Source data are available online for this figure.

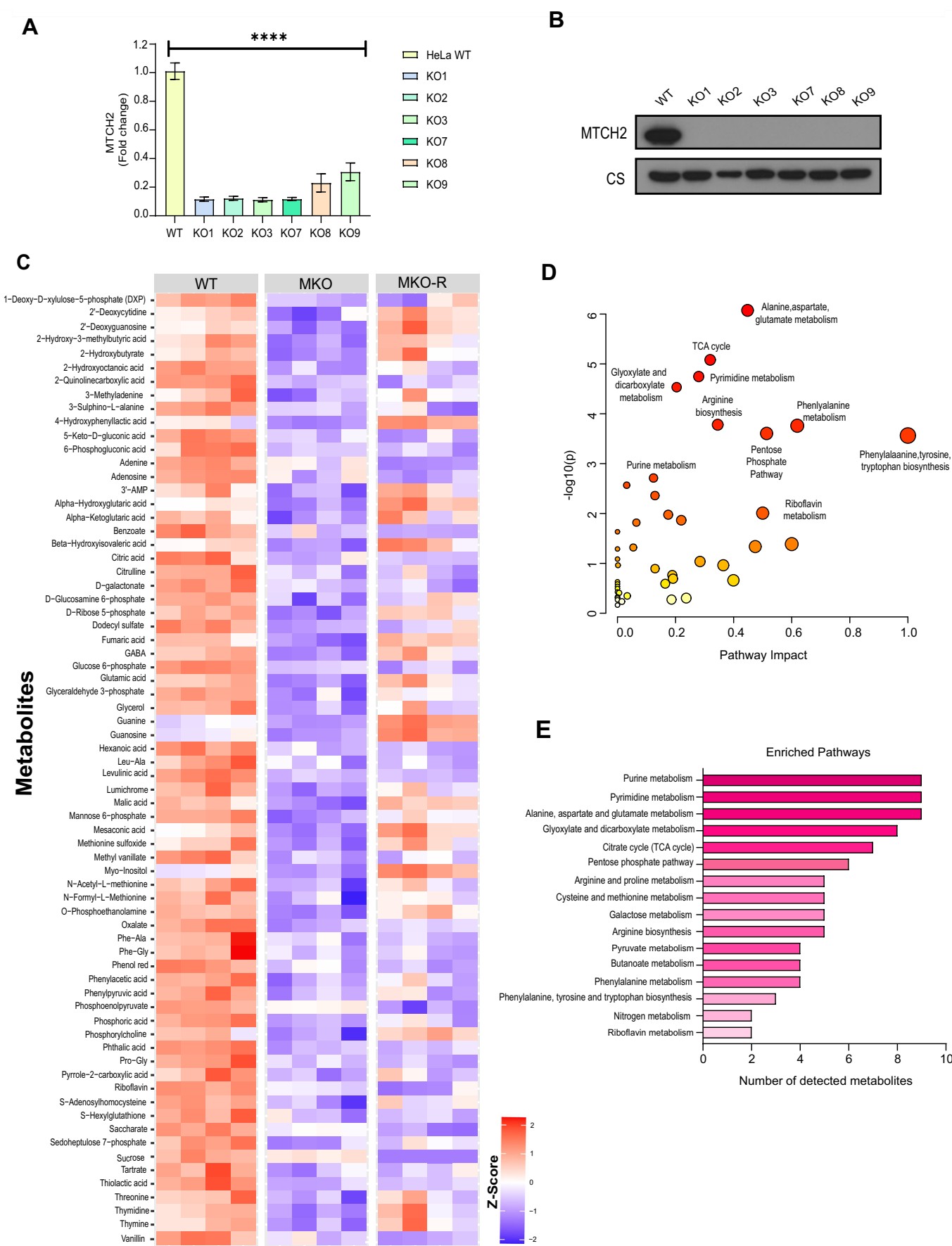

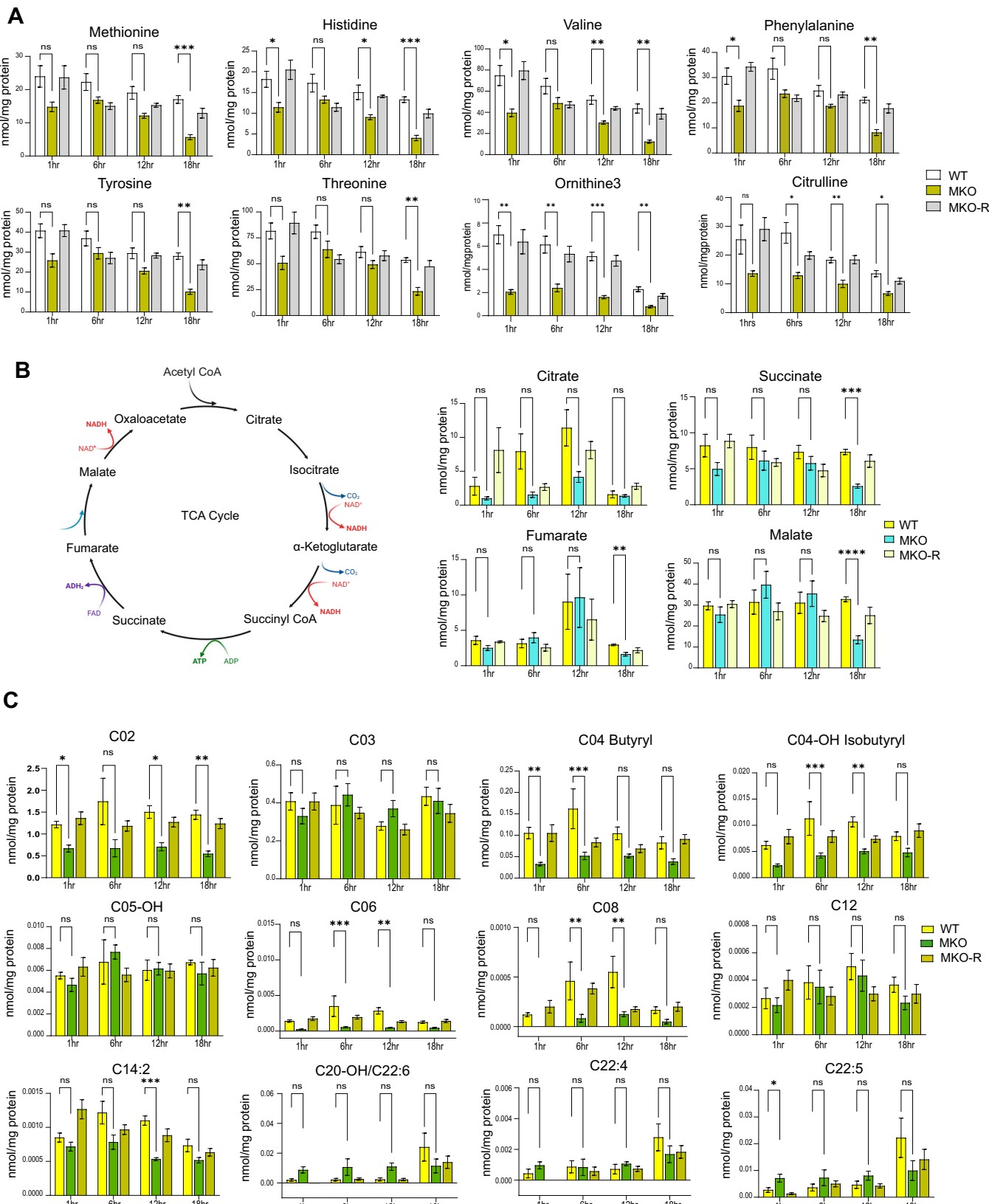

◄ **Figure EV2. An increase in amino acid/TCA cycle/lipid utilization in MKO cells.**

(A) Average levels of a set of amino acids in all 3 cell lines at all four time points. (B) Left panel: Schematic representation of the TCA cycle. Right panels: Average levels of a set of TCA cycle intermediates in all 3 cell lines at all four time points. (C) Average levels of acylcarnitines in all 3 cell lines at all four time points. Results in all graphs in (A–C) are presented as mean ± SEM (ns, non-significant, *$P<0.05$, **$P<0.001$, ***$P<0.0003$, ****$P<0.0007$; two-way ANOVA with Dunnett multiple comparison test; $n=6$ biological replicates). Source data are available online for this figure.

**A**

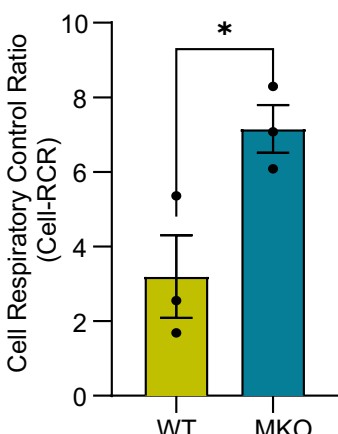

**B**

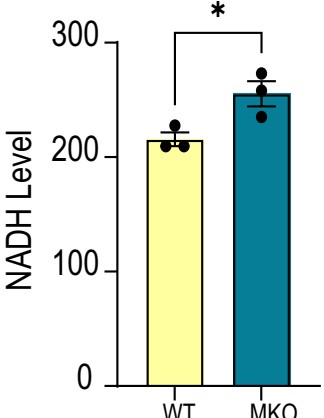

**C**

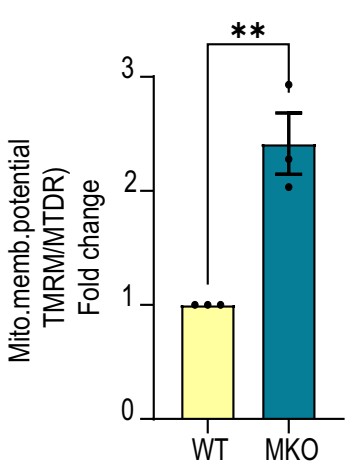

**D**

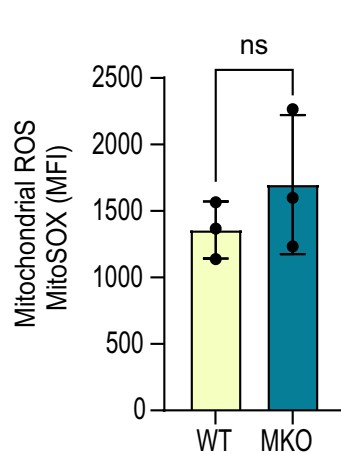

◀  **Figure EV3.  Increased mitochondrial oxidative function in MKO cells.**

(A) Cell Respiratory Control Ratio of WT and MKO cells. Results are presented as mean ± SEM (Unpaired $t$ test, *$P<0.05$. $N=3$ independent experiments with 6–8 technical replicates in each experiment). (B) Increased total mitochondrial NADH content in MKO cells. Mitochondrial NADH levels were calculated as described in the Methods. Results are presented as mean ± SEM (Unpaired $t$ test, *$P<0.05$, $N=3$ independent experiments). Mitochondrial NADH content was calculated as the difference in NADH mean autofluorescence intensity (MFI). Maximal NADH autofluorescence was determined in response to KCN and minimal NADH autofluorescence was determined in response to FCCP as described in the Methods. (C) Increased mitochondrial membrane potential in MKO cells. Results are presented as means ± SEM (Unpaired $t$ test, **$P<0.001$, $N=3$ independent experiments). (D) Mitochondrial ROS levels in WT and MKO cells. Levels of mitochondrial ROS (measured using mitoSOX) are presented. Results are presented as mean ± SEM (Unpaired $t$ test, ns-nonsignificant, $N=3$ independent experiments). Source data are available online for this figure.

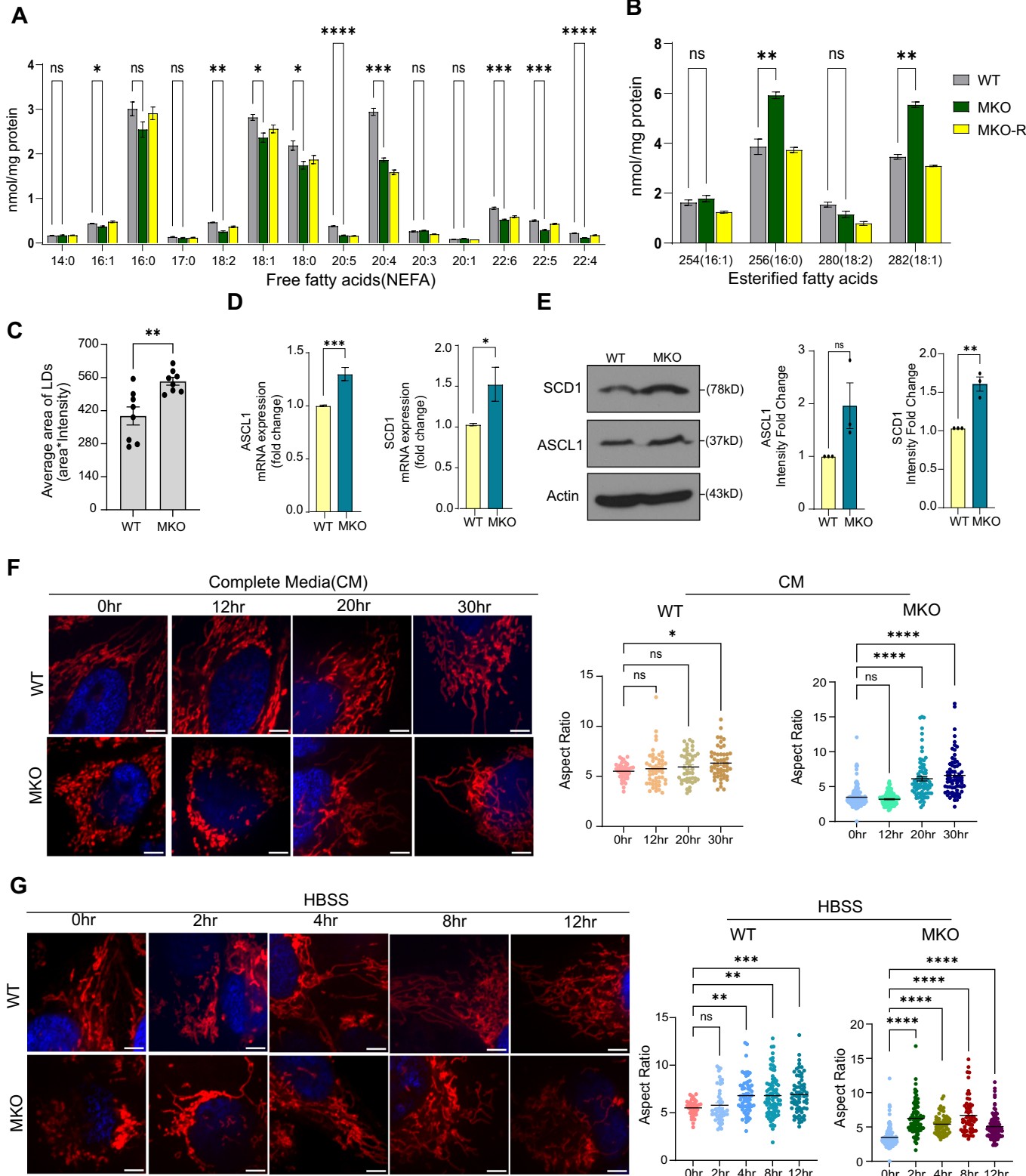

◀ **Figure EV4. MKO cells show accelerated mitochondria elongation under nutrient depletion conditions.**

(**A, B**) The levels of Free fatty acids (NEFA)(A) and Esterified fatty acids (**B**) in all 3 cell lines. Results are presented as mean ± SEM (ns, non-significant, *P<0.05, **P<0.001, ***P<0.0003, ****P<0.0007; one-way ANOVA, $n = 4$ independent biological replicates). (**C**) Quantification of LD average size in WT and MKO cells. The average LD size from different time points was combined and plotted as a single group for both WT and KO cells. Results are presented as mean ± SEM (**P<0.001; Unpaired *t* test, $n=3$ independent biological replicates). (**D**) mRNA levels of ACSL1 and SCD1in WT and MKO cells. Results are presented as mean ± SEM (*P<0.05, ***P<0.0003; Unpaired *t* test, $N = 3$ independent experiments). (**E**) Left panel: Western blot for ACSL1 and SCD1 proteins in lysates from WT and MKO cells 20 h-post media change. Right panels: Quantification of relative density of ACSL1 and SCD1 normalized to Actin (loading control). Results are presented as mean ± SEM (ns, non-significant, **P<0.001, Unpaired *t* test). $N = 3$ independent experiments (**F**). Analyses of mitochondria morphology. Left panel: WT and MKO cells were plated into complete media (CM), then media was refreshed (considered as time 0) and pictures were taken at 0, 12, 20, and 30 h-post media change. Mitochondria were labeled using Mito Tracker Deep red (MTDR). Scale bar=5 μm. Right panels: Quantification of mitochondria morphology of WT and MKO cells. Results are presented as mean ± SEM (ns, non-significant, *P<0.05, ****P<0.0007; Unpaired t-test, $N=3$ independent experiments). (**G**) Left panel: Analyses of mitochondria morphology in WT and MKO cells incubated in HBSS, and pictures were taken at 2, 4, 8 and 12-h post incubation and labeled as in (**F**). Scale bar = 5 μm. Right panels: Quantification of mitochondria morphology of WT and MKO cells. Results are presented as means ± SEM (ns, non-significant, **P<0.001, ***P<0.0003, ****P<0.0007; $n = 3$ Independent biological replicates). Source data are available online for this figure.

**A**

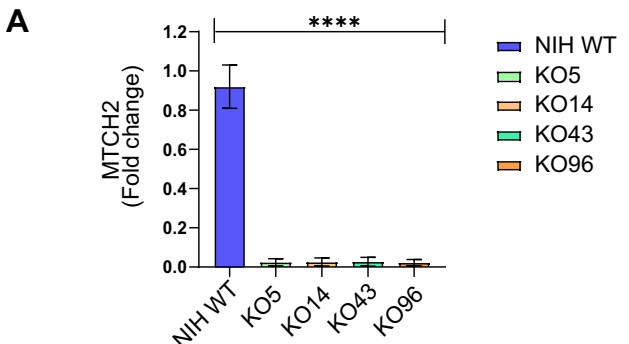

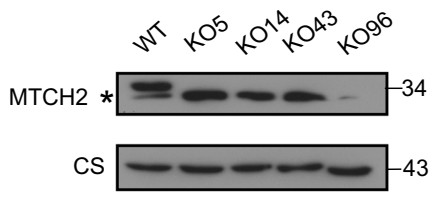

**B**

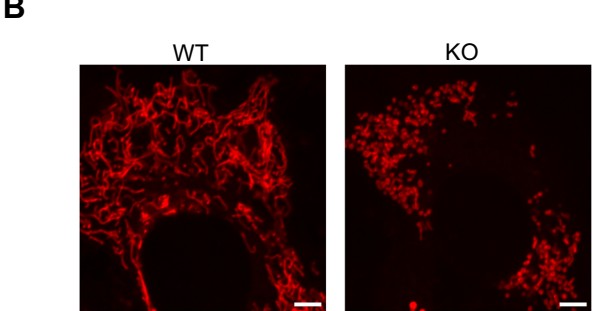

**C**

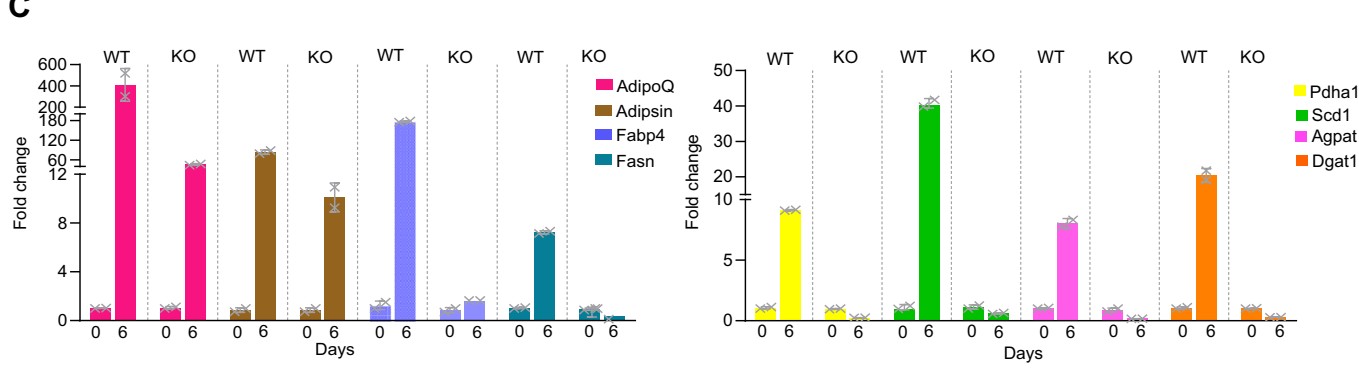

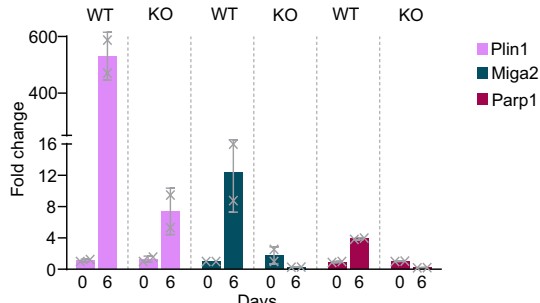

◀ **Figure EV5.   MTCH2 is critical for adipocyte differentiation.**

(**A**) MTCH2 mRNA expression was checked by RT-PCR (left panel) and protein level were checked by Immunoblot (right panel) in 4 different MTCH2 knockout (KO) and WT clones. *, nonspecific band. Results are presented as mean ± SEM of one representative out of three independent experiments (****$P<0.0007$, ordinary one-way ANOVA). (**B**) Mitochondrial morphology in NIH3T3L1 Preadipocytes. MTCH2 knockout (KO) leads to mitochondrial fragmentation. Mitochondria were labeled using MitoTracker Deep red (MTDR). Scale bar = 5 μm. (**C**) RT-PCR of WT and MTCH2 knockout (KO) cells at day 0 and day 6-post differentiation. Components of the adipogenic effector genes were analyzed: adiponectin (AdipoQ), Adipsin, fatty-acid-binding protein 4 (Fabp4), fatty acid synthase (FASN), pyruvate dehydrogenase (Pdha1), stearyl-CoA desaturase (Scd1), 1-acyl-sn-glycerol-3-phosphate (Agpat), diacylglycerolacyltransferase (Dgat1), perilipin (Plin1), mitoguardin 2 (Miga2), and poly(ADP-ribose) polymerase1 (Parp1). Results are presented as mean ± SD of one representative out of three independent experiments. Normalization was done by taking geometric mean of three housekeeping genes, Importin, Tubulin and AcTH. Source data are available online for this figure.

    