## [Peer Review File · The EMBO Journal]

MTCH2 controls energy demand and expenditure to fuel anabolism during adipogenesis

Sabita Chourasia, Christopher Petucci, Clarissa Shoffler, Dina Abbasian, Hu Wang, Xianlin Han, Ehud Sivan, Alexander Brandis, Tevie Mehlman, Sergey Malitsky, Maxim Itkin, Ayala Sharp, Ron Rotkopf, Bareket Dassa, Limor Regev, Yehudit Zaltsman, and Atan Gross

Corresponding authors: Atan Gross (atan.gross@weizmann.ac.il) , Sabita Chourasia (sabita.chourasia@weizmann.ac.il)

Review Timeline:

Transfer from Review Commons:	12th Mar 24
Editorial Decision:	28th Mar 24
Revision Received:	8th Sep 24
Editorial Decision:	23rd Oct 24
Revision Received:	11th Nov 24
Accepted:	19th Nov 24

Editors: Kelly Anderson / Daniel Klimmeck

Transaction Report:

This manuscript was transferred to The EMBO Journal following peer review at Review Commons.

Review #1**1. Evidence, reproducibility and clarity:****Evidence, reproducibility and clarity (Required)**

In the present study, Chourasia et al. describe the effects of MTCH2 deficiency on various metabolic parameters. Using temporal metabolomics in HeLa cells, they show an increase in ATP demand in cells lacking MTCH2. They also show altered lipid metabolism in NIH3T3L1 preadipocytes lacking MTCH2, associated with impaired maturation.

This study is mainly descriptive, containing large amount of information that would be of interest to the understanding of the role of MTCH2 in cell metabolism. The manuscript would benefit from thorough editing to make it more focused and accurate. This is challenging since there is a very large amount of data. Below are some suggestions, along with a couple of experiments that I suggest to add in order to clarify the mechanism through which MTCH2 acts.

****Major Comments:****

1. The ratio ADP/ATP as well as AMP/ATP as well as the decrease in TCA metabolites are indications of ATP demand. Nevertheless, these are not fluxes. It would therefore be worthy to complement these results with respirometry.
2. The dramatic increase in AMP/ATP ratio suggests an increase in AMPK activity. Testing it (by measuring AMPK phosphorylation or ACC phosphorylation) could further strengthen the results but it is not a must-have.
3. The cause of the increase in AMP in the NIH3T3L1 is not addressed. The involvement of acyl-CoA synthetase is worth discussing or investigating.
4. Fig. 2F shows MKO and WT, but it doesn't show MKO-R. This is an important control since lactate doesn't seem to be affected in the MKO-R.

****Minor comments:****

In the intro there are a few instances that could benefit from some more accuracy:

1. In the abstract there is no mention of the cells that are used.
2. Line 56: "...(OXPHOS) converts nutrients into adenosine triphosphate (ATP)". It would be more accurate to write that OXPHOS converts the chemical energy that is stored in

nutrients into ATP.

3. Line 58: "The mitochondrial NAD⁺/NADH pool are substrates for OXPHOS". It would be more accurate to write that NADH is the substrate. (NAD⁺ is after all the product of oxidation)

4. Line 59-60: "Along with ADP, NAD⁺ also plays an important role in the regulation of the Krebs cycle". Instead of "regulation" I think it is more specific to write "stimulates" (otherwise add ATP and NADH which are also involved in regulation. 5. Line 65: "...changing metabolic states. In addition, mitochondria..." A link between the two sentences seems to be missing.

5. Figure 1 is a heavy figure. Some results are significant and some show only a tendency. The description (Lines 119-125) is too general. It addresses only the "trends". A bit more specificity as to the metabolites or ratios of metabolites and the time points that are significant would be in place.

9. Lines 136-139 "The metabolomics analyses revealed additional important changes in many more nutrient substrates, which included a decrease in most amino acids (Fig. 2A and Fig. S2A). Notably, the most significant change was seen in glutamine (Fig. 2A, left top graph), one of the major amino acid-nutrient sources" Glutamine is indeed an important amino acid and the effect is strong, but in this case it's increased. From the first sentence one would think it's decreased. Again, be more specific in your sentencings.

10. Lines 157-159: "Thus, the acyl carnitine profile suggests that 1- to 12-hrs post media change the MKO cells use BCAAs as a nutrient source, and later shift to unsaturated acyl carnitines, specifically to the C16:1 and C18:1 forms". This conclusion does not derive from the description by the result - address the difference between the different carnitines that occurs at different times.

11. Lines 167-168: "These results suggest that there is higher metabolism of acetyl CoA in the MKO cells leading to a bell-shape dynamics (low-high-low levels)." The interpretation is unclear. I understand that you mean that there is fluctuation within the group, however the acetyl-CoA levels remain lower MKO than in the control at all time points. This further suggests a decrease in TCA cycle.

12. Lines 172-173 addresses the bell shape of lactate, yet the most prominent result is the 3-fold increase in lactate levels compared to WT which is not mentioned. Unfortunately, the MKO-R shows a similar increase. Still, this should be addressed in the text.

13. 182-184: "Taken together, the results presented above are consistent with the idea that the increased amino acid/lipid/carbohydrate metabolism and substantial decrease of many metabolites in MKO cells is most likely due to their increased utilization to meet them increased cellular energy demand." I'm not sure how this conclusion is reached. From metabolite levels alone, it's difficult to conclude about fluxes. Also, in a previous conclusion the authors wrote that the TCA cycle is probably reduced; this contradicts the

above conclusion. If the authors mean that the FFAs and amino acids are used for anabolism and glucose for meeting energy demand, they should state so more clearly.

14. Figure 3A can be split into 2 or 3 subfigures. I think it would make it more comprehensible.

15. Line 210-211: "Notably, we also found that MTCH2 knockout cells showed accelerated mitochondria elongation (Fig. S3D, top panels), which was further pronounced when cells were grown in HBSS". The increase in mitochondrial elongation comes after fragmentation in MTCH2 KO. Although this is a known phenotype, it is good to address it shortly in the text.

16. Line 208-209: "LDs from dispersed to a highly clustered distribution that was often observed in close proximity to mitochondria..." The proximity of mitochondria to LD suggests the possibility that there is an increase in peridroplet-mitochondria, which have been shown to be involved in biogenesis LD. It might be interesting to investigate this path as an explanation to the observed phenotype.

17. Lines 244-245: "These results suggest that the MTCH2 knockout preadipocytes face a cellular energy crisis that is similar to the one seen in the MTCH2 knockout HeLa cells presented earlier" It's true that NAD⁺ and AMP (as well as AMP/ATP ratio) are increased but in view of the high ATP and NADH, it's difficult reach the conclusion that there's an energy crisis.

18. In the discussion- Lines 267-269: "Thus, MTCH2 might act like a "relay station" by sensing and connecting between metabolic intermediates/pathways and dynamic changes in mitochondria morphology/energy production by receiving and sending Wi-Fi signals." It's difficult to raise such specific hypothesis from the results. Use milder terms.

2. Significance:

Significance (Required)

Great study

3. How much time do you estimate the authors will need to complete the suggested revisions:

Estimated time to Complete Revisions (Required)

(Decision Recommendation)

Between 1 and 3 months

4. Review Commons values the work of reviewers and encourages them to get credit for their work. Select 'Yes' below to register your reviewing activity at Web of Science

Reviewer Recognition Service (formerly Publons); note that the content of your review will not be visible on Web of Science.

Yes

Review #2

1. Evidence, reproducibility and clarity:

Evidence, reproducibility and clarity (Required)

Mitochondrial carrier homolog 2 (MTCH2, SLC25A50) loss induces alterations in mitochondrial dynamics and energy utilization. However, the molecular mechanisms underlying these changes are still unknown. The study employs temporal metabolomic and lipidomic analyses, uncovering heightened catabolism, increased lipid storage, and disrupted adipogenesis in MTCH2 KO cells. The manuscript provides a comprehensive metabolic profile, revealing ATP demand increase, oxidized cellular environment, and adaptive changes in MTCH2 KO cells. Notably, in line with the fundamental role of fatty acid biosynthesis and anabolism in adipogenesis, the authors demonstrate that MTCH2 loss inhibits adipocyte differentiation. This work offers novel insights into the broader metabolic consequences of MTCH2 depletion.

****Major comments:****

- The key conclusions of the paper align with the conducted experiments, but a few additional experiments are necessary to state some claims and provide more robust conclusions.
- The paper could benefit from the inclusion of specific experiments, particularly those that address the following aspects:

1. Validate MTCH2 ablation in HeLa and NIH3T3L1 through sequencing of clonal lines, Western blot analysis to confirm the absence of the protein, and real-time PCR to assess whether the mechanism involves mRNA decay.
2. Provide a more detailed rationale for their temporal metabolomics approach, elucidating the choice of the media and the timepoints of cell collection. The method involves an initial culture of the cells in DMEM medium, followed by a switch to complete medium (CM) for overnight cell growth, and subsequent refreshment with CM for different timepoints before the metabolomics analyses. Authors should articulate the reasoning for opting for CM.

Furthermore, authors should explicitly explain the rationale behind selecting specific timepoints for cell collection after the addition of the fresh medium.

3. In Figure 1, authors conclude that MTCH2 ablation stimulates oxidative metabolism and ATP production to fulfill increased cellular ATP demands. However, this conclusion is based only on metabolomic analyses of the ADP/ATP ratio. To comprehensively assess the impact on cellular respiration, the authors should monitor the Oxygen Consumption Rate (OCR) and report the Respiratory Control Ratio (RCR).

4. NAD⁺/NADH ratio: authors should measure NADH levels in both mitochondria and cytosol. This can be accomplished through NADH autofluorescence (recommended) or commercially available kits. This additional analysis would contribute to a more comprehensive interpretation of the observed changes in oxidative metabolism. They should also include measurement of mitochondrial membrane potential using TMRM. Suggested experiment: measuring NADH autofluorescence. The autofluorescence of mitochondrial NADH can be distinguished from cytosolic NADH by optimizing substrate consumption followed by the complete inhibition of electron feeding to the ETC. The redox state of NADH reflects the equilibrium between mitochondrial ETC activity and the rate of substrate supply. After acquiring basal autofluorescence levels through live imaging, max signal is obtained by stimulating maximal respiration (FCCP), and min signal is obtained by inhibiting respiration (NaCN or Rot+AA). Subsequently, "NADH redox indexes" are generated by expressing the basal NADH levels as a percentage of the difference between the oxidized and reduced signals. Furthermore, by examining the fluorescence signal increase after NaCN addition, the rate of NADH production can be monitored. This rate serves as a proxy of TCA efficiency.

5. Authors observe a reduction in the levels of various amino acids and TCA cycle intermediates, indicative of an increased flux through the TCA cycle. This proposition could be further supported by measuring the kinetics of NADH autofluorescence. Additionally, a decrease in metabolites associated with the urea cycle, such as citrulline and ornithine, is observed, yet this observation remains uncommented and warrants discussion.

Intriguingly, an elevation in Branched-Chain Amino Acids (BCAAs) and unsaturated acyl carnitines is noted, leading to the hypothesis of an increased transport and breakdown of fatty acids in the mitochondria to meet the heightened cellular demand for ATP in MTCH2 KO cells. To substantiate this, and to quantitatively measure mitochondrial fuel utilization in live cells, authors shall perform a Mitofuel Flex Test by measuring the Oxygen Consumption Rate (OCR) in cells treated with inhibitors of each mitochondrial oxidative pathway including etomoxir. This approach would enable the measurement of the dependency, capacity, and flexibility of cells concerning the pathway of interest in meeting ATP demand. It is also recommended to perform MitoStress test in cells supplemented with only one of the carbon sources (such as Glucose, Glutamine, Long chain and Short

Chain Fatty acids).

6. In Fig 3, a reduction in membrane lipids, free fatty acids, and non-esterified fatty acids is observed, while there is an increase in esterified fatty acids, storage lipids like Triacylglycerols (TAG) and Cholesterol Esters (CE), and lipid droplet number and size. Notably, these lipid droplets are positioned closer to mitochondria in MKO cells. The authors propose that MKO results in enhanced transfer and metabolism of lipid moieties at the mitochondria to generate ATP. To provide insights into the molecular mechanisms underlying the observed lipid changes in MTCH2 KO cells, the following experiments are recommended: Employ Western blot and real-time PCR to measure the levels of enzymes crucial in TAG and CE formation and accumulation (e.g., Long-chain acyl-CoA synthetase (Acsl), Stearoyl-CoA desaturase (SCD) or others). Evaluate the enzymatic activity of these identified enzymes to understand their functional role in lipid metabolism in MTCH2 KO cells.

- The suggested experiments are realistic in terms of time and resources, ensuring practical feasibility.
- The data and methods are presented in a clear and reproducible manner.
- The experiments appear adequately replicated, and the statistical analysis seems OK.

****Minor comments:****

- There are no specific experimental issues that require addressing.
- Prior studies are appropriately referenced
- In general, both the text and figures are clear and accurate. The significant alteration of metabolites found in their metabolomic dataset should be plotted using the online tool MetaboAnalyst to analyze metabolic pathways and generate better visualizations.
- Overall, the presentation is satisfactory with only minor language adjustments recommended. A minor suggestion for improvement involves refining the language used in the text. Instead of consistently using the term "produce energy," please use "conversion of energy".

2. Significance:

Significance (Required)

General Assessment: The study, through the integration of metabolomic and lipidomic data in MTCH2 KO cells, provide a comprehensive overview of the metabolic rewiring of these cells. This metabolic change is particularly interesting in the context of adipogenesis, offering valuable insights into the interconnectedness of a mitochondrial solute carrier,

cellular metabolism and adipogenesis.

Comparison and Advance: The current study significantly advances our understanding of the mitochondrial carrier homolog 2 (MTCH2) by uncovering its intricate roles in metabolism, and adipogenesis. While prior research identified MTCH2 as a regulator of apoptosis and mitochondrial dynamics, the present study expands our knowledge by elucidating its involvement in cellular metabolism and adipocyte differentiation. The major advance lies in the detailed exploration of MTCH2's impact on cellular metabolism through temporal metabolomic and lipidomic analyses. The study reveals that MTCH2 deletion leads to heightened ATP demand, an oxidized cellular environment, and alterations in lipid, amino acid, and carbohydrate metabolism. Additionally, the adaptive response in MTCH2 knockout cells involves a strategic decrease in membrane lipids and an increase in storage lipids. Furthermore, the study unveils a novel connection between the imbalance in energy metabolism triggered by MTCH2 deletion and the inhibition of adipocyte differentiation—a process that demands substantial energy and reductive biosynthetic activities. This mechanistic insight provides a conceptual advance, indicating how MTCH2, beyond its known role in apoptosis and mitochondrial dynamics, plays a pivotal role in orchestrating cellular metabolism and adipogenesis. Importantly, this work aligns with prior observations that hinted at MTCH2's involvement in fatty acid synthesis, storage, and use through his identified interactome. In summary, the study advances our knowledge of MTCH2 by providing a more comprehensive understanding of its roles in cellular metabolism and adipocyte differentiation, shedding new light on its multifaceted functions beyond its originally identified roles.

Audience: This research will appeal to a broad audience, ranging from specialists in cellular metabolism to those with a general interest in mitochondrial dynamics and biochemistry.

3. How much time do you estimate the authors will need to complete the suggested revisions:

Estimated time to Complete Revisions (Required)

(Decision Recommendation)

Between 1 and 3 months

4. Review Commons values the work of reviewers and encourages them to get credit for their work. Select 'Yes' below to register your reviewing activity at Web of Science

Reviewer Recognition Service (formerly Publons); note that the content of your review will not be visible on Web of Science.

No

Review #3

1. Evidence, reproducibility and clarity:

Evidence, reproducibility and clarity (Required)

This study by Chourasia et al determined the effects of MTCH2 deletion on the total metabolite content (polar and lipid metabolites) per total protein of HeLa cells, analyzed at different consecutive times after adding fresh and complete media (high glucose, 10%FBS). They also analyzed the effects of nutrient depletion (no FBS). In addition, authors assessed the effects of MTCH2 deletion on mitochondrial morphology, lipid droplet number and size in HeLa cells, as well as in the differentiation of mouse fibroblasts to adipocytes. From the metabolite snapshots under these different times and conditions, authors conclude that MTCH2 deletion increases mitochondrial oxidative function to induce a catabolic state, which impedes lipid synthesis and, as a result, adipocyte differentiation. The major concerns are that it is unclear whether the metabolic phenotype observed is a consequence of MTCH2 deletion inducing a decrease in proliferation of HeLa, as well as of fibroblasts that need to reach confluence to differentiate. In this regard, it is also unclear whether MTCH2 deletion increases ATP demand and/or promotes a catabolic program, as metabolic flux analyses are missing. A minor concern is the use of computer (processing system), antenna and wifi analogies to describe the role of MTCH2 in mitochondrial function, which is confusing.

2. Significance:

Significance (Required)

This study represents a thorough characterization of the metabolite content in proliferating HeLa cells in the absence of MTCH2 expression. The changes observed in polar and lipidic metabolites are novel, interesting and contribute to our understanding on the role of MTCH2 function in cellular metabolism. The main limitation of the study is that the levels of most metabolites are normalized by protein content, comparing conditions in which cell number and protein synthesis have changed. Thus, it is unclear whether some of the effects observed are a consequence of the reported role of MTCH2 supporting the

proliferation of different tumors and cell lines, or whether it is a direct effect of MTCH2 increasing ATP demand and/or being a direct activator of mitochondrial catabolism. Related to this point, it is unclear whether the defect in adipocyte differentiation induced by MTCH2 KO in NIH3T3 fibroblasts might be caused by an inability of MTCH2 KO to reach confluency at day 0, needed for differentiation. Finally, respirometry and mitochondrial ROS content analyses would be needed to confirm that the changes in the metabolite levels induced by MTCH2 are caused by an increase in mitochondrial oxidation leading to nutrient depletion, as authors conclude. For example, an increase in the ADP/ATP ratio could also be caused by an inhibition of mitochondrial ATP synthase in the mitochondria, concurrent to an increase in ROS production, which would decrease NADH and NADPH content.

3. How much time do you estimate the authors will need to complete the suggested revisions:

Estimated time to Complete Revisions (Required)

(Decision Recommendation)

Between 3 and 6 months

Yes

Revision Plan

Manuscript number: RC-2023-02310

Corresponding author(s): Sabita Chourasia & Atan Gross

1. General Statements [optional]

The main goal of this study was to determine the cellular metabolic changes resulting from knocking out mitochondrial MTCH2.

We thank the reviewers for their efforts and excellent comments.

2. Description of the planned revisions and those that have already been incorporated in the transferred manuscript (description of new data incorporated in the transferred MS is marked below in yellow; accordingly, we have also modified the text in the transferred MS, and those modifications were also marked in yellow)

Reviewer #1

Major Comments:

1) The ratio ADP/ATP as well as AMP/ATP as well as the decrease in TCA metabolites are indications of ATP demand. Nevertheless, these are not fluxes. It would therefore be worthy to complement these results with respirometry.

R: We have performed respirometry experiments using the seahorse instrument and we found that the MTCH2 KO (MKO) cells show both higher basal respiration and higher spare respiratory capacity (see new Fig 2F in the transferred MS). The increased respiration in the MKO cells is an indication of an increase in ATP demand.

2) The dramatic increase in AMP/ATP ratio suggests an increase in AMPK activity. Testing it (by measuring AMPK phosphorylation or ACC phosphorylation) could further strengthen the results but it is not a must-have.

R: We agree that the high AMP/ATP ratio suggests an increase in AMPK activity. As suggested by the reviewer, we have measured AMPK phosphorylation, and found an increase in in the MKO cells (see new Fig 1C in the transferred MS).

3) The cause of the increase in AMP in the NIH3T3L1 is not addressed. The involvement of acyl-CoA synthetase is worth discussing or investigating.

R: We agree and we will discuss the involvement of acyl-CoA synthetase in NIH3T3L1.

Revision Plan

4) Fig. 2F shows MKO and WT, but it doesn't show MKO-R. This is an important control since lactate doesn't seem to be affected in the MKO-R.

R: We will repeat the experiment with the MKO-R cells.

Minor comments:

As requested by the reviewer, we will rephrase all the sentences requested.

Reviewer #2

Major comments:

1. Validate MTCH2 ablation in HeLa and NIH3T3L1 through sequencing of clonal lines, Western blot analysis to confirm the absence of the protein, and real-time PCR to assess whether the mechanism involves mRNA decay.

R: We will perform the sequencing, Western blot, and PCR to confirm the absence of MTCH2 and to assess mRNA decay.

2. Provide a more detailed rationale for their temporal metabolomics approach, elucidating the choice of the media and the timepoints of cell collection. The method involves an initial culture of the cells in DMEM medium, followed by a switch to complete medium (CM) for overnight cell growth, and subsequent refreshment with CM for different timepoints before the metabolomics analyses. Authors should articulate the reasoning for opting for CM. Furthermore, authors should explicitly explain the rationale behind selecting specific timepoints for cell collection after the addition of the fresh medium.

R: We will provide a more detailed rationale for our temporal metabolomics approach, elucidating the choice of the media and the time points of cell collection.

3. In Figure 1, authors conclude that MTCH2 ablation stimulates oxidative metabolism and ATP production to fulfill increased cellular ATP demands. However, this conclusion is based only on metabolomic analyses of the ADP/ATP ratio. To comprehensively assess the impact on cellular respiration, the authors should monitor the Oxygen Consumption Rate (OCR) and report the Respiratory Control Ratio (RCR).

R: We have performed respirometry experiments using the seahorse instrument and we found that the MTCH2 KO (MKO) cells show both higher basal respiration and higher spare respiratory capacity (see new Fig 2F in the transferred MS). We still need to perform the experiment to determine the Respiratory Control Ratio (RCR).

4. NAD⁺/NADH ratio: authors should measure NADH levels in both mitochondria and cytosol. This can be accomplished through NADH autofluorescence (recommended) or commercially available kits. This additional analysis would contribute to a more comprehensive interpretation of the observed changes in oxidative metabolism. They should also include measurement of mitochondrial membrane potential using TMRM. Suggested experiment: measuring NADH autofluorescence. The autofluorescence of mitochondrial NADH can be distinguished from cytosolic NADH by optimizing substrate consumption followed by the complete inhibition of

electron feeding to the ETC. The redox state of NADH reflects the equilibrium between mitochondrial ETC activity and the rate of substrate supply. After acquiring basal autofluorescence levels through live imaging, max signal is obtained by stimulating maximal respiration (FCCP), and min signal is obtained by inhibiting respiration (NaCN or Rot+AA). Subsequently, "NADH redox indexes" are generated by expressing the basal NADH levels as a percentage of the difference between the oxidized and reduced signals. Furthermore, by examining the fluorescence signal increase after NaCN addition, the rate of NADH production can be monitored. This rate serves as a proxy of TCA efficiency.

R: We are in the process of measuring the autofluorescence of both mitochondrial and cytosolic NADH, to obtain the redox state of NADH, and the new results will be included in the revised MS.

5. Authors observe a reduction in the levels of various amino acids and TCA cycle intermediates, indicative of an increased flux through the TCA cycle. This proposition could be further supported by measuring the kinetics of NADH autofluorescence. Additionally, a decrease in metabolites associated with the urea cycle, such as citrulline and ornithine, is observed, yet this observation remains uncommented and warrants discussion. Intriguingly, an elevation in Branched-Chain Amino Acids (BCAAs) and unsaturated acyl carnitines is noted, leading to the hypothesis of an increased transport and breakdown of fatty acids in the mitochondria to meet the heightened cellular demand for ATP in MTCH2 KO cells. To substantiate this, and to quantitatively measure mitochondrial fuel utilization in live cells, authors shall perform a Mitofuel Flex Test by measuring the Oxygen Consumption Rate (OCR) in cells treated with inhibitors of each mitochondrial oxidative pathway including etomoxir. This approach would enable the measurement of the dependency, capacity, and flexibility of cells concerning the pathway of interest in meeting ATP demand. It is also recommended to perform MitoStress test in cells supplemented with only one of the carbon sources (such as Glucose, Glutamine, Long chain and Short Chain Fatty acids).

R: We are in the process of setting up the Mitofuel Flex Test, which will include measuring respiration of cells pretreated with inhibitors to different oxidative pathways. We will also perform the MitoStress Test by supplementing cells with only one of the carbon sources.

6. In Fig 3, a reduction in membrane lipids, free fatty acids, and non-esterified fatty acids is observed, while there is an increase in esterified fatty acids, storage lipids like Triacylglycerols (TAG) and Cholesterol Esters (CE), and lipid droplet number and size. Notably, these lipid droplets are positioned closer to mitochondria in MKO cells. The authors propose that MKO results in enhanced transfer and metabolism of lipid moieties at the mitochondria to generate ATP. To provide insights into the molecular mechanisms underlying the observed lipid changes in MTCH2 KO cells, the following experiments are recommended: Employ Western blot and real-time PCR to measure the levels of enzymes crucial in TAG and CE formation and accumulation (e.g., Long-chain acyl-CoA synthetase (Acsl), Stearoyl-CoA desaturase (SCD) or others). Evaluate the enzymatic activity of these identified enzymes to understand their functional role in lipid metabolism in MTCH2 KO cells.

R: We will employ Western blot and real-time PCR analyses to determine the expression levels of enzymes crucial in TAG and CE formation and accumulation (e.g., Long-chain acyl-CoA synthetase (Acsl) and Stearoyl-CoA desaturase (SCD)). However, evaluating the enzymatic activity of these enzymes will take a long time, and such additional data does not seem necessary given the scope of the study.

Revision Plan

Minor comments:

As requested by the reviewer, we will use the online tool MetaboAnalyst and refine the language used in the text.

Reviewer #3

This study by Chourasia et al determined the effects of MTCH2 deletion on the total metabolite content (polar and lipid metabolites) per total protein of HeLa cells, analyzed at different consecutive times after adding fresh and complete media (high glucose, 10%FBS). They also analyzed the effects of nutrient depletion (no FBS). In addition, authors assessed the effects of MTCH2 deletion on mitochondrial morphology, lipid droplet number and size in HeLa cells, as well as in the differentiation of mouse fibroblasts to adipocytes. From the metabolite snapshots under these different times and conditions, authors conclude that MTCH2 deletion increases mitochondrial oxidative function to induce a catabolic state, which impedes lipid synthesis and, as a result, adipocyte differentiation. The major concerns are that it is unclear whether the metabolic phenotype observed is a consequence of MTCH2 deletion inducing a decrease in proliferation of HeLa, as well as of fibroblasts that need to reach confluence to differentiate. In this regard, it is also unclear whether MTCH2 deletion increases ATP demand and/or promotes a catabolic program, as metabolic flux analyses are missing. A minor concern is the use of computer (processing system), antenna and wifi analogies to describe the role of MTCH2 in mitochondrial function, which is confusing.

This study represents a thorough characterization of the metabolite content in proliferating HeLa cells in the absence of MTCH2 expression. The changes observed in polar and lipidic metabolites are novel, interesting and contribute to our understanding on the role of MTCH2 function in cellular metabolism. The main limitation of the study is that the levels of most metabolites are normalized by protein content, comparing conditions in which cell number and protein synthesis have changed. Thus, it is unclear whether some of the effects observed are a consequence of the reported role of MTCH2 supporting the proliferation of different tumors and cell lines, or whether it is a direct effect of MTCH2 increasing ATP demand and/or being a direct activator of mitochondrial catabolism. Related to this point, it is unclear whether the defect in adipocyte differentiation induced by MTCH2 KO in NIH3T3 fibroblasts might be caused by an inability of MTCH2 KO to reach confluency at day 0, needed for differentiation. Finally, respirometry and mitochondrial ROS content analyses would be needed to confirm that the changes in the metabolite levels induced by MTCH2 are caused by an increase in mitochondrial oxidation leading to nutrient depletion, as authors conclude. For example, an increase in the ADP/ATP ratio could also be caused by an inhibition of mitochondrial ATP synthase in the mitochondria, concurrent to an increase in ROS production, which would decrease NADH and NADPH content.

R:

1) This reviewer has raised an important point, suggesting that some of the metabolic changes that appeared in the MTCH2 KO (MKO) cells are possibly due to their slower growth/proliferation. We agree with this point however it is difficult to critically assess it since it is equally possible that the slow growth/proliferation of the MKO cells is due to a metabolic defect resulting from the absence of MTCH2. Moreover, the results obtained in NIH3T3 actually support the “metabolic defect” theory, since the MKO cells were induced to differentiate only once they reached confluency (like in the case of the WT cells). This important point will be emphasized in the revised MS.

Revision Plan

2) We performed respirometry experiments using the seahorse instrument and we found that the MKO cells show both higher basal respiration and higher spare respiratory capacity (see new Fig 2F in the transferred MS). These new findings suggest that MTCH2 deletion increases ATP demand which promotes a catabolic program leading to nutrient depletion). We still need to determine the levels of mitochondrial ROS by FACS using mitoSOX.

3) We will correct the wording requested.

3. Description of analyses that authors prefer not to carry out

Major comment #6 of Rev #2:

Evaluating the enzymatic activity of Long-chain acyl-CoA synthetase (AcsI) and Stearoyl-CoA desaturase (SCD) will take a long time, and such additional data does not seem necessary given the scope of the study.

Dear Prof. Gross,

Thank you for transferring your manuscript with Review Commons referee reports and responses to The EMBO Journal.

Given the referees' positive recommendations, I would like to invite you to submit a revised version of the manuscript, addressing the comments of all three reviewers. I should add that it is EMBO Journal policy to allow only a single round of revision, and acceptance of your manuscript will therefore depend on the completeness of your responses in this revised version.

When preparing your letter of response to the referees' comments, please bear in mind that this will form part of the Review Process File, and will therefore be available online to the community. For more details on our Transparent Editorial Process, please visit our website.

Thank you for the opportunity to consider your work for publication. I look forward to your revision.

Yours sincerely,

Kelly M Anderson, PhD
Editor, The EMBO Journal
k.anderson@embojournal.org

We realize that it is difficult to revise to a specific deadline. In the interest of protecting the conceptual advance provided by the work, we recommend a revision within 3 months (26th Jun 2024). Please discuss the revision progress ahead of this time with the editor if you require more time to complete the revisions. Use the link below to submit your revision:

Link Not Available

Rev_Com_number: RC-2023-02310

New_manu_number: EMBOJ-2024-117259-T

Corr_author: Gross

Title: High-energy demand and nutrient exhaustion in MTCH2 knockout cells

Reviewer #1

In the present study, Chourasia et al. describe the effects of MTCH2 deficiency on various metabolic parameters. Using temporal metabolomics in HeLa cells, they show an increase in ATP demand in cells lacking MTCH2. They also show altered lipid metabolism in NIH3T3L1 preadipocytes lacking MTCH2, associated with impaired maturation. This study is mainly descriptive, containing a large amount of information that would be of interest to the understanding of the role of MTCH2 in cell metabolism.

The manuscript would benefit from thorough editing to make it more focused and accurate. This is challenging since there is a very large amount of data. Below are some suggestions, along with a couple of experiments that I suggest to add in order to clarify the mechanism through which MTCH2

Major Comments

1) The ratio ADP/ATP as well as AMP/ATP as well as the decrease in TCA metabolites are indications of ATP demand. Nevertheless, these are not fluxes. It would therefore be worthy to complement these results with respirometry.

We sincerely thank the reviewer and appreciate his/her positive evaluation of our work. To address this point, we performed respirometry experiments and found increased respiration in MKO cells. The new data is presented in **Figure R1.1** below and was added as **Figure 3A** to our revised manuscript.

Figure R1.1. A. Representative traces of real-time OCR measurements in WT and MKO cells display basal respiration and OCR changes in response to sequential inhibitor injections. **B.** Basal and Maximal respiration are shown in the left and right panels, respectively. Results are presented as mean \pm SEM, with statistical significance determined by an unpaired t-test (*** p <0.0003, **** p <0.0007). Data represents one of three independent experiments, each with 6-8 technical replicates.

2) The dramatic increase in AMP/ATP ratio suggests an increase in AMPK activity. Testing it (by measuring AMPK phosphorylation or ACC phosphorylation) could further strengthen the results but it is not a must-have.

We performed the suggested experiment and found an increase in phospho-AMPK in MKO cells. The new data is presented in **Figure R1.2** below and was added as **Figure 1C** to our revised manuscript.

Figure R1.2 MKO cells show an increase in AMPK phosphorylation. Western blot analysis of WT and MKO cells using anti-p-AMPK and anti-AMPK antibodies. β -Actin is used as a loading control. A representative of 3 independent experiments is shown.

3)The cause of the increase in AMP in the NIH3T3L1 is not addressed. The involvement of acyl-CoA synthetase is worth discussing or investigating.

We thank the reviewer for raising this important point. Our metabolomic data from HeLa cells, along with previous research on hematopoietic stem cells (Maryanovich et al., 2015) and muscle tissue (Buzaglo-Azriel et al., 2016), demonstrate that MTCH2 deletion induces significant changes in energy metabolism. This alteration is also observed in NIH3T3L1 preadipocytes, further supporting MTCH2's role in regulating energy metabolism.

Our lipidomics data from HeLa cells suggest that MTCH2 knockout (MKO) cells rely more on lipids as a nutrient source for mitochondrial respiration compared to WT cells. This reliance is accompanied by an adaptive increase in the storage lipids TAGs and CEs to sustain higher mitochondrial oxidative respiration, likely to meet the increased energy demands.

Considering these findings, it is plausible that the alteration in energy metabolism is a key phenomenon following MTCH2 deletion, which may explain the similar AMP profile observed in NIH3T3L1 preadipocytes. Another potential source of increased AMP in MTCH2 KO NIH3T3L1 cells could be the conversion of free fatty acids (FFAs) released during lipolysis into fatty acyl-CoA through re-esterification. Re-esterification is an energy-consuming process catalyzed by acyl-CoA synthetase (ACS), which involves ATP consumption and AMP generation (Miyoshi et al., 2008; Sharma et al., 2023), and during stimulated lipolysis serves as a protective mechanism against lipotoxicity (Sharma et al. 2023).

In summary, MTCH2 deletion in NIH3T3L1 cells leads to an increase in AMP levels can be for two primary reasons: (1) changes in nucleotide metabolism (AMP, ADP, ATP), which seems to be a common phenotype associated with MTCH2 deletion, and (2) MTCH2 deletion drives cells to utilize lipids, along with other nutrients, to support enhanced mitochondrial oxidation. The increase in lipolysis (release of fatty acids from stored TAGs) and re-esterification (conversion of fatty acids into

fatty acyl-CoA), an expensive process in terms of ATP, can contribute to the observed rise in AMP levels in NIH3T3L1 cells.

(lines 444-451 in our revised manuscript).

4) Fig. 2F shows MKO and WT, but it doesn't show MKO-R. This is an important control since lactate doesn't seem to be affected in the MKO-R.

We thank the reviewer for raising this important point. We performed this experiment and found that glucose uptake was increased in MKO cells but not in MKO-R cells. The new data is presented in **Figure R1.4** below and was added as **Figure 2F** to our revised manuscript.

Figure R1.4 Glucose uptake from media of WT, MKO, and MKO-R cells after 2 hrs of glucose starvation. Results are presented as mean \pm SEM ($***p < 0.0003$); one-way ANOVA with Dunnett multiple comparison test (N=3 independent experiment).

Minor comments:

In the intro there are a few instances that could benefit from some more accuracy:

4) In the abstract there is no mention of the cells that are used.

We included "HeLa cells" and "NIH3T3L1 cells" in the abstract.

5) Line 56: "...(OXPHOS) converts nutrients into adenosine triphosphate (ATP)". It would be more accurate to write that OXPHOS converts the chemical energy that is stored in nutrients into ATP.

We addressed this point and added that "Mitochondria are the primary sites of cellular respiration, where the process of oxidative phosphorylation (OXPHOS) converts the chemical energy that is stored in nutrients into adenosine triphosphate (ATP)" (lines 55-57 in our revised manuscript).

6) Line 58: "The mitochondrial NAD⁺/NADH pool are substrates for OXPHOS". It would be more accurate to write that NADH is the substrate. (NAD⁺ is after all the product of oxidation).

We revised the text to read: "The mitochondrial NADH is the substrate for OXPHOS, where 90% of cellular ATP production takes place" (lines 59-60 in our revised manuscript).

7) Line 59-60: "Along with ADP, NAD⁺ also plays an important role in the regulation of the Krebs

cycle". Instead of "regulation" I think it is more specific to write "stimulates"(otherwise add ATP and NADH which are also involved in regulation).

We revised the text to read: "Along with ADP, NAD⁺ also stimulates the Krebs cycle." (lines 60 in our revised manuscript).

8) Line 65: "...changing metabolic states. In addition, mitochondria..." A link between the two Sentences seems to be missing.

We added a link: This dynamic behavior is essential for maintaining a healthy mitochondrial population, responding to energy demands, and orchestrating adaptability in the face of changing metabolic states (Han, Eric A Bushong, et al. 2023; Liesa and Shirihai 2013; Monzel, Enríquez, and Picard 2023; Youle and Van Der Bliek 2012). In addition to its dynamics, mitochondrial biogenesis is intricately linked to energy homeostasis. (lines 64-67 in our revised manuscript).

8) Figure 1 is a heavy figure. Some results are significant and some show only a tendency. The description (Lines 119-125) is too general. It addresses only the "trends". A bit more specificity as to the metabolites or ratios of metabolites and the time points that are significant would be in place.

We thank the reviewer for suggesting this change and we rephrased the text accordingly: Detailed analysis of the results revealed that the MKO cells showed an increase in ADP levels at 12 hrs (**Figure 1B**, top row, left panel), an increase in the ADP/ATP ratio at 12 and 18 hrs (top row, right panel), an increase in the NAD⁺/NADH ratio (middle row, middle panel), an increase in NADP⁺ levels (middle row, right left), an increase in the nicotinamide precursors, i.e. nicotinamide adenine dinucleotide (NAAD) and nicotinamide (NAM) at most time points (bottom row). The increase in the ADP/ATP ratio led us to check the phospho-AMPK levels, and indeed the MKO cells showed higher levels 18 hrs-post media change (**Figure 1C**). Overall, MTCH2 deletion leads to an increase in both the ADP/ATP ratio and nicotinamide metabolism. (lines 129-136 in our revised manuscript).

9) Lines 136-139 "The metabolomics analyses revealed additional important changes in many more nutrient substrates, which included a decrease in most amino acids (Fig. 2A and Fig. S2A). Notably, the most significant change was seen in glutamine (Fig. 2A, left top graph), one of the major amino acid-nutrient sources" Glutamine is indeed an important amino acid and the effect is strong, but in this case, it's increased. From the first sentence, one would think it's decreased. Again, be more specific in your sentencing.

We thank the reviewer for noticing this and we rephrased the text accordingly: The metabolomics analyses revealed additional important changes in many more nutrient substrates, which included a decrease in most amino acids that enter the TCA cycle at different points (Sherry, Lee, and Choi 2015) (**Figure 2A** and **Figure EV2A**). Notably, an interesting pattern of significant change was seen in glutamine, one of the major amino acid-nutrient sources (Yang et al. 2014), along with alanine and serine. There was an initial trend of an increase in the level of these amino acids in the MKO cells followed by a strong decrease, which reached statistical significance only 18hr post-media change (**Figure 2A**, top panel). A similar profile of decreased levels at later time points was seen for many amino acids (**Figure 2A**, bottom panel, and **Figure EV2A**). A decrease in amino acids usually represents an increase in TCA cycle metabolism(Shiratori et al. 2019). Indeed, we found a decrease in TCA cycle intermediates 18 hrs-post media change in the MKO cells (**Figure EV2B**). (lines 146-155 in our revised manuscript).

10) Lines 157-159: "Thus, the acyl carnitine profile suggests that 1- to 12-hrs post media change the MKO cells use BCAAs as a nutrient source, and later shift to unsaturated acyl carnitines, specifically to the C16:1 and C18:1 forms ". This conclusion does not derive from the description by the result - address the difference between the different carnitines that occur at different times.

We thank the reviewer for noticing this and we addressed the difference between the different carnitines that occur at different times. (lines 161-173 in our revised manuscript).

11) Lines 167-168: "These results suggest that there is higher metabolism of acetyl CoA in the MKO cells leading to a bell-shape dynamics (low-high-low levels)." The interpretation is unclear. I understand that you mean that there is fluctuation within the group, however, the acetyl-CoA levels remain lower MKO than in the control at all time points. This further suggests a decrease in TCA cycle.

We appreciate the reviewer's insightful comments. We agree that acetyl-CoA levels are consistently lower in MKO cells compared to the control at all time points. Our primary aim was to highlight the significant fluctuations in acetyl-CoA levels within the MKO group over time, in contrast to the WT cells. We also concur with the reviewer that the persistent low levels of acetyl-CoA in MKO cells could suggest a decrease in TCA cycle activity. In our revised manuscript, we addressed this by measuring NADH autofluorescence following KCN treatment, a potent complex IV inhibitor that blocks mitochondrial respiration and leads to maximal NADH accumulation. Our findings indicate that MKO cells exhibit higher NADH autofluorescence compared to WT cells (**Figure EV3B** in our revised manuscript). This suggests that MKO cells have a larger mitochondrial NADH pool after KCN treatment, which may serve as a proxy for enhanced TCA cycle efficiency. (lines 204-215 in our revised manuscript).

12) Lines 172-173 address the bell shape of lactate, yet the most prominent result is the 3-fold increase in lactate levels compared to WT which is not mentioned. Unfortunately, the MKO-R shows a similar increase. Still, this should be addressed in the text.

We thank the reviewer for this comment, and we sincerely apologize for our mistake. It was a mistake while preparing the figure: in Figure 2E (WT, top panel) the value appeared as 200 instead of 1200 (actual value) by missing the initial "1". We corrected Figure 2E in our revised manuscript.

13) 182-184: "Taken together, the results presented above are consistent with the idea that the increased amino acid/lipid/carbohydrate metabolism and substantial decrease of many metabolites in MKO cells is most likely due to their increased utilization to meet the increased cellular energy demand." I'm not sure how this conclusion is reached. From metabolite levels alone, it's difficult to conclude about fluxes. Also, in a previous conclusion the authors wrote that the TCA cycle is probably reduced; this contradicts the above conclusion. If the authors mean that the FFAs and amino acids are used for anabolism and glucose for meeting energy demand, they should state so more clearly.

We appreciate the reviewer's comment and apologize for the confusion. We have revised our conclusion in the text as follows: In summary, MTCH2 deletion leads to increased uptake of glucose, to a decrease in various internal metabolites, and to an increase in BCAAs and unsaturated acylcarnitine. This phenomenon is likely due to the heightened utilization of a broad range of nutrients driven by increased energy demands. (lines 192-195 in our revised manuscript).

14) Figure 3A can be split into 2 or 3 subfigures. I think it would make it more comprehensible.

We thank the reviewer for suggesting this change, and we split the revised Figure 4 into 4A, 4B, and 4C.

15) Line 210-211: "Notably, we also found that MTCH2 knockout cells showed accelerated mitochondria elongation (Fig. S3D, top panels), which was further pronounced when cells were grown in HBSS". The increase in mitochondrial elongation comes after fragmentation in MTCH2 KO. Although this is a known phenotype, it is good to address it shortly in the text.

We thank the reviewer for noticing this and we rephrased the text accordingly. Consistent with previous studies from our lab (Bahat et al. 2018; Goldman et al. 2024) and others (Labbé et al. 2021), MTCH2 deletion leads to mitochondrial fragmentation in HeLa cells. Notably, we observed that MTCH2 knockout (MKO) cells, which in general display fragmented mitochondrial morphology (**Figure EV4F**, 0 hr), exhibit accelerated mitochondrial elongation over time (**Figure EV4F**). This elongation was even more pronounced when the cells were cultured in HBSS (**Figure EV4G**). (lines 273-276 in our revised manuscript).

16) Line 208-209: "LDs from dispersed to a highly clustered distribution that was often observed in close proximity to mitochondria..." The proximity of mitochondria to LD suggests the possibility that there is an increase in per droplet-mitochondria, which have been shown to be involved in biogenesis LD. It might be interesting to investigate this path as an explanation to the observed phenotype.

We appreciate the reviewer's suggestion to explore this area further. While MKO cells might develop peri-droplet mitochondria (PDM), the primary finding of our manuscript focuses on nutrient depletion over time, likely driven by increased nutrient utilization to meet heightened energy demands. Regarding lipid droplets (LDs): we observed an increase in LD numbers over time in complete media (CM), with a more pronounced effect within 2 hours in nutrient-depleted media (i.e., HBSS). Additionally, LD morphology shifted from a dispersed to a highly clustered distribution in CM, a pattern that became more evident within 2 hours in HBSS. This change in LD number and distribution is characteristic of nutrient-starved cells, as described by Nguyen et al. (2017), where amino acid deficiency triggers autophagy-dependent LD biogenesis. The increase in LD numbers and mitochondrial elongation are known indicators of nutrient-deprived cells (Nguyen et al., 2017; Rambold, Cohen, and Lippincott-Schwartz, 2015). Our data also show that along with many other metabolites, amino acid levels decrease over time in MKO cells, coinciding with the elongation of fragmented mitochondria. This suggests that MKO cells may be adapting to, or preparing for, nutrient depletion due to their increased nutrient utilization.

17) Lines 244-245: "These results suggest that the MTCH2 knockout preadipocytes face a cellular energy crisis that is similar to the one seen in the MTCH2 knockout HeLa cells presented earlier" It's true that NAD⁺ and AMP (as well as AMP/ATP ratio) are increased but in view of the high ATP and NADH, it's difficult reach the conclusion that there's an energy crisis.

We agree and thank the reviewer for this comment, removed this conclusion from the revised manuscript, and rewrote Lines 244-245 in the text of our revised manuscript: "Taken together, targeted metabolic profiling of preadipocytes showed different profiling for both energy (AMP, ATP) and reductive potential (NAD⁺, NADH, NADP⁺) metabolism for MTCH2 KO cells as compared to WT cells, which resembles the profile in MKO HeLa cells". (lines 318-321 in our revised manuscript).

18) In the discussion- Lines 267-269: "Thus, MTCH2 might act like a "relay station" by sensing and connecting between metabolic intermediates/pathways and dynamic changes in mitochondria morphology/energy production by receiving and sending Wi-Fi signals." It's difficult to raise such specific hypothesis from the results. Use milder terms.

We thank the reviewer and replaced these terms with milder terms: "Absence of MTCH2, a scenario of losing a pivotal metabolic mediator, can lead to a disconnection between the cellular energy demand and the cellular energy utilization". (lines 369-371 in our revised manuscript).

Reviewer#2

Mitochondrial carrier homolog 2 (MTCH2, SLC25A50) loss induces alterations in mitochondrial dynamics and energy utilization. However, the molecular mechanisms underlying these changes are still unknown. The study employs temporal metabolomic and lipidomic analyses, uncovering heightened catabolism, increased lipid storage, and disrupted adipogenesis in MTCH2 KO cells.

The manuscript provides a comprehensive metabolic profile, revealing ATP demand increase, oxidized cellular environment, and adaptive changes in MTCH2 KO cells. Notably, in line with the fundamental role of fatty acid biosynthesis and anabolism in adipogenesis, the authors demonstrate that MTCH2 loss inhibits adipocyte differentiation. This work offers novel insights into the broader metabolic consequences of MTCH2 depletion.

Major comments:

- The key conclusions of the paper align with the conducted experiments, but a few additional experiments are necessary to state some claims and provide more robust conclusions.
- The paper could benefit from the inclusion of specific experiments, particularly those that address the following aspects:

1. Validate MTCH2 ablation in HeLa and NIH3T3L1 through sequencing of clonal lines, Western blot analysis to confirm the absence of the protein, and real-time PCR to assess whether the mechanism involves mRNA decay.

We thank the reviewer for this comment. We validated MTCH2 deletion by using two different methods, real-time PCR and Western blot. In the revised manuscript, we show the mRNA and protein level of MTCH2 in all 6 Single-cell CRISPR knockout HeLa clones and all 4 Single-cell CRISPR knockout NIH3T3L1 clones (**Figure R2.1** below and **Figure EV1A and B and EV5A and B** in our revised manuscript). We found traces of mRNA in the MTCH2 KO clones and a complete deletion of protein in all 10 clones.

Figure R2.1. Validation of MTCH2 deletion by checking mRNA and Protein levels by RT-PCR and Western blot in HeLa and NIH3T3L1 CRISPR Clones. Asterix (*) mark the non-specific band of Mouse MTCH2. We used Commercial Abcam antibody to recognize Human MTCH2 for HeLa and Mouse MTCH2 for NIH3T3L1 cells) (Grinberg et al. 2005).

2. Provide a more detailed rationale for their temporal metabolomics approach, elucidating the choice of the media and the time points of cell collection. The method involves an initial culture of the cells in DMEM medium, followed by a switch to complete medium (CM) for overnight cell growth, and subsequent refreshment with CM for different timepoints before the metabolomics analyses. Authors should articulate the reasoning for opting for CM. Furthermore, authors should explicitly explain the rationale behind selecting specific time points for cell collection after the addition of the fresh medium.

We thank the reviewer for this comment and apologize for the confusion. Our method of cell growth did not involve an initial culture of the cells in only DMEM medium. We routinely grow cells in Complete Media (CM), which consists of standard DMEM with 4.5g/L glucose and L-glutamine, supplemented with sodium pyruvate and 10% fetal bovine serum (FBS). This is the regular medium used to maintain HeLa cells. The experimental pipeline involved seeding the cells in CM, allowing them to adapt and establish their morphology on the seeding plate. Once a monolayer is formed, the medium is refreshed to remove any cellular debris or unattached cells, marking this point as time 0. Cells are then collected at 1, 6, 12, and 18 hrs-post media change for metabolomics analysis.

The rationale for selecting these specific time points for temporal metabolomics stems from our initial global metabolomics analysis, conducted 30 hrs-post media change, where we observed significant differences between the three stable clones (**Figure EV1A**). To understand how early these changes begin and to track their dynamics, we selected four earlier time points arbitrarily. When we looked carefully for these differential metabolites, we found that they belong to many major metabolic pathways like glyoxylate and dicarboxylate metabolism, TCA cycle, pentose-phosphate pathway (PPP), amino acid metabolism, purine and pyrimidine metabolism, and nitrogen metabolism. In our revised manuscript, we have also included pathway enrichment analysis in **Figure EV1D** and the number of metabolites detected in enriched pathways in **Figure EV1E**.

3. In Figure 1, authors conclude that MTCH2 ablation stimulates oxidative metabolism and ATP production to fulfill increased cellular ATP demands. However, this conclusion is based only on metabolomic analyses of the ADP/ATP ratio. To comprehensively assess the impact on cellular respiration, the authors should monitor the Oxygen Consumption Rate (OCR) and report the Respiratory Control Ratio (RCR).

We thank the reviewer for this important comment. We initially monitored the Oxygen Consumption Rate (OCR) and found an increase in both basal and maximal oxygen consumption in MKO cells as compared to WT cells (**Figure R2.3** below, and new **Figure 3A** in revised manuscript) Thus, the increase in basal respiration is likely to result in a higher energy demand, which probably explains the increased amino acid/lipid/carbohydrate utilization in the MKO cells.

To comprehensively assess the cellular ATP demand we monitored OCR on both Intact cells and permeabilized HeLa cells and reported the Cellular-Respiratory Control Ratio (cell-RCR) in **Figure EV3A** and Mitochondrial Respiratory Control Ratio (RCR) in **Figure 3C**. The cell Respiratory Control Ratio is calculated as the ratio of the uncoupled rate (FCCP-induced respiration on the whole cell (analogous of mitochondrial state 3u) to the rate with oligomycin present (analogous to mitochondrial state 4) which is analogous to the respiratory control ratio of isolated mitochondria/permeabilized cells, previously described (Brand and Nicholls 2011) and we calculated the RCR on permeabilized cells according to the previously described (Brand and Nicholls 2011; Rogers et al. 2011). Both Intact (see **Figure R2.3** below and **Figure EV3A**) and permeabilized (**Figure 3C**, in our revised manuscript) MKO cells show higher RCR compared to WT cells.

Figure R2.3. A. Representative traces of real-time OCR measurements in WT and MKO cells showing the basal respiration and OCR change in response to sequential injections of inhibitors (Left panel). Basal and maximal respiration are shown in the middle and right panels. Results are presented as mean \pm SEM (Unpaired t-test, $***p < 0.0003$). N=3 independent experiments with 6-8 technical replicates in each experiment. **B.** Cell Respiratory Control Ratio of WT and MKO cells. Results are presented as mean \pm SEM (Unpaired t-test, $*p < 0.05$. N=3 independent experiments with 6-8 technical replicates in each experiment). **C.** Respiratory Control Ratio (RCR) analysis on permeabilized WT and MKO cells. Succinate and rotenone are used to measure ADP phosphorylation and maximal respiration. State 3, State 3u, State 4, and RCR were calculated as described in the Methods. Results are presented as mean percent \pm SEM (Unpaired t-test, $*p < 0.05$; ns, non-significant). N=3 independent experiments.

4. NAD⁺/NADH ratio: authors should measure NADH levels in both mitochondria and cytosol. This can be accomplished through NADH autofluorescence (recommended) or commercially available kits. This additional analysis would contribute to a more comprehensive interpretation of the observed changes in oxidative metabolism. They should also include measurement of mitochondrial membrane potential using TMRM.

Suggested experiment: Measuring NADH autofluorescence. The autofluorescence of mitochondrial NADH can be distinguished from cytosolic NADH by optimizing substrate consumption followed by the complete inhibition of electron feeding to the ETC. The redox state of NADH reflects the equilibrium between mitochondrial ETC activity and the rate of substrate supply. After acquiring basal autofluorescence levels through live imaging, max signal is obtained by stimulating maximal respiration (FCCP), and min signal is obtained by inhibiting respiration (NaCN or Rot+AA). Subsequently, "NADH redox indexes" are generated by expressing the basal NADH levels as a percentage of the difference between the oxidized and reduced signals. Furthermore, by examining the fluorescence signal increase after NaCN addition, the rate of NADH production can be monitored. This rate serves as a proxy of TCA efficiency.

We thank the reviewer for this comment and the suggested experiment. We addressed this comment by measuring NADH autofluorescence as recommended by the reviewer with some modifications instead of using commercially available kits. We measured the mitochondrial NADH pool in WT and MKO cells as described (Plun-Favreau et al. 2012, Maryanovich et al. 2015). Briefly, mitochondrial NADH level was calculated as the difference between the maximal NADH autofluorescence in response to KCN (KCN, a potent inhibitor of Complex IV which blocks the mitochondrial respiration completely) and minimum NADH autofluorescence in response to uncoupler, FCCP (FCCP which

stimulates maximal respiration) as suggested by the reviewer. NADH redox index was calculated by calculating the initial NADH autofluorescence when the minimum NADH autofluorescence was normalized to 0% and the maximum to 100%. Data shows that MKO cells have a lower NADH redox index than WT cells (see **Figure R2.4.A** below and **new Figure 3B** in our revised manuscript), suggesting an increased respiration rate and accelerated electron consumption in MKO cells.

We also found that there is an increase in autofluorescence signal after KCN addition in MKO cells as compared to WT cells (see **Figure R2.4B** below and **Figure EV3B** in the revised manuscript). This means that the total mitochondrial pool of NADH in MKO cells is higher than in WT cells, suggesting increased substrate availability in these cells resulting in a higher rate of NADH production and a proxy of TCA cycle efficiency as suggested by the reviewer and previously described (Plun-Favreau et al. 2012). NADH autofluorescence was measured by flow cytometry after excitation with a UV laser with a main emission peak at 470 nm.

We also measured and included the data of mitochondrial membrane potential, showing that there is an increase in MKO cells as compared to WT cells (see **Figure R2.4C** below and **Figure EV3C** in the revised manuscript).

Figure R2.4. A. Reduced NADH redox in MKO Cells. NADH redox index was calculated as described in the Methods. Results are presented as mean percent \pm SEM. **B.** Increased total mitochondrial NADH content in MKO cells. Mitochondrial NADH content was calculated as the difference in NADH mean

autofluorescence intensity (MFI). Maximal NADH autofluorescence was determined in response to KCN and minimal NADH autofluorescence was determined in response to FCCP calculated as described in the Methods. **C.** Increased mitochondrial membrane potential in MKO cells. All the data are presented as mean \pm SEM of three independent experiments with at least 3 technical replicates.

5. Authors observe a reduction in the levels of various amino acids and TCA cycle intermediates, indicative of an increased flux through the TCA cycle. This proposition could be further supported by measuring the kinetics of NADH autofluorescence. Additionally, a decrease in metabolites associated with the urea cycle, such as citrulline and ornithine, is observed, yet this observation remains un-commented and warrants discussion. Intriguingly, an elevation in Branched-Chain Amino Acids (BCAAs) and unsaturated acyl carnitines is noted, leading to the hypothesis of an increased transport and breakdown of fatty acids in the mitochondria to meet the heightened cellular demand for ATP in MTCH2 KO cells. To substantiate this, and to quantitatively measure mitochondrial fuel utilization in live cells, authors shall perform a Mitofuel Flex Test by measuring the Oxygen Consumption Rate (OCR) in cells treated with inhibitors of each mitochondrial oxidative pathway including etomoxir. This approach would enable the measurement of the dependency, capacity, and flexibility of cells concerning the pathway of interest in meeting ATP demand. It is also recommended to perform Mito Stress test in cells supplemented with only one of the carbon sources (such as Glucose, Glutamine, Long chain and Short Chain Fatty acids).

We thank the reviewer for this comment. Our above data from **Figure R2.4.** of NADH autofluorescence already indicates there is an increased substrate availability (i.e. NADH) in MKO cells meaning a higher rate of NADH production and this can serve as a proxy of TCA efficiency and increased flux through the TCA cycle.

Regarding the decrease in the urea cycle metabolites in the MKO cells. The urea cycle removes excess nitrogen by producing urea, however, under scarce metabolic conditions cells may inhibit the urea cycle to preserve nitrogen, resulting in a decrease in these metabolites. On the other hand, it should be noted that urea is produced only in the liver and thus the HeLa line, derived from cervical cancer cells, most likely does not have an intact urea cycle. If this is the case then the decrease in citrullin and ornithine is probably due to increased cellular metabolism, which leads to the decrease we observed in many amino acids. This new text was added to the Discussion (lines 363-369 in our revised manuscript).

To strengthen our data of an elevation in Branched-Chain Amino Acids (BCAAs) and unsaturated acylcarnitine and the hypothesis that this increase is due to increased breakdown of fatty acids in the mitochondria to meet the increased cellular demand for ATP in MKO cells, we performed the Substrate Oxidation test by measuring the Oxygen Consumption Rate (OCR) in the presence of three different inhibitors targeting three major metabolic pathways: UK5099 (Pyruvate), BPTES (Glutamine) and Etomoxir (Fatty acids). The results suggest that the Basal and Maximal OCR of WT cells are affected by UK5099 (Pyruvate), and BPTES (Glutamine) but not by Etomoxir (Fatty acids), whereas the OCR of MKO cells is affected by all three inhibitors (see **Figure R2.5A and B** below and **Figure 3D** in our revised manuscript).

The reviewer also recommended conducting the Mito-Stress test on cells supplemented with only one carbon source (such as glucose, glutamine, long-chain, or short-chain fatty acids). However, when we

attempted to grow the cells in 96-well Seahorse plates with only one carbon source, the cells experienced excessive stress, preventing accurate OCR measurements. Therefore, we adopted an alternative approach to assess the growth pattern of MKO cells by depleting one carbon source at a time, as previously described by Han et al. 2023.

We investigated whether MKO cells rely on different nutrient sources compared to WT cells for their growth rate. We found that restricting free fatty acids inhibited the growth of MKO cells but not WT cells (Figure 3E). Both cell types were dependent on glutamine; however, MKO cells were less affected by low-glucose conditions. Our data suggest that MKO cells utilize a broader range of nutrients to support increased OXPHOS and growth, whereas WT cells are dependent on fewer nutrient sources.

Figure R2.5. **A.** Representative traces of real-time OCR measurements of WT (left panel) and MKO (right panel) cells showing the basal OCR and Maximal OCR change in response to sequential injections of inhibitors. **B.** Percentage of change in basal OCR (bottom left panel) and Maximal OCR (bottom right panel) in response to UK5099, etomoxir, and BPTES in WT and KO cells. Results are presented as mean percent \pm SEM (two-way ANOVA, Post-hoc tests were done using estimated marginal means (R package 'emmeans')). ** $p < 0.001$, *** $p < 0.0003$). N = 4 independent experiments. **C.** Cell count of WT and KO cells proliferating under the conditions of normal medium (25mM glucose), and medium with no free fatty acids (FFAs), low glucose (12mM), or no glutamine. N=3 independent experiments.

6. In Fig 3, a reduction in membrane lipids, free fatty acids, and non-esterified fatty acids is observed, while there is an increase in esterified fatty acids, storage lipids like Triacylglycerols (TAG) and

Cholesterol Esters (CE), and lipid droplet number and size. Notably, these lipid droplets are positioned closer to mitochondria in MKO cells. The authors propose that MKO results in enhanced transfer and metabolism of lipid moieties at the mitochondria to generate ATP. To provide insights into the molecular mechanisms underlying the observed lipid changes in MTCH2 KO cells, the following experiments are recommended: Employ Western blot and real-time PCR to measure the levels of enzymes crucial in TAG and CE formation and accumulation (e.g., Long-chain acyl-CoA synthetase (Acsl), Stearoyl-CoA desaturase (SCD) or others). Evaluate the enzymatic activity of these identified enzymes to understand their functional role in lipid metabolism in MTCH2 KO cells.

We appreciate the reviewer's comment and the proposed experiments. In response, we examined the levels of enzymes crucial for TAG and CE formation, ACSL1 and SCD1, by assessing both their mRNA and protein levels (see **Figure R2.6** below and **Figure EV4D and E**). The data indicate that mRNA levels of both ACSL1 and SCD1 are elevated in MKO cells. At the protein level, SCD1 follows a similar pattern to its mRNA expression; however, ACSL1 did not reach statistical significance.

Figure R2.6.A. An Increase in mRNA levels of ACSL1 and SCD1 in MKO cells. **B.** Western blot for ACSL1 and SCD1 proteins in lysates from WT and MKO cells 20hrs post-media change. Right panels: Quantification of relative density of ACSL1 and SCD1 normalized to Actin (loading control). All Results are presented as mean \pm SEM (* p <0.05, ** p <0.001, *** p <0.0003; Unpaired t-test, N=3 independent experiments).

- The suggested experiments are realistic in terms of time and resources, ensuring practical feasibility.
- The data and methods are presented in a clear and reproducible manner.
- The experiments appear adequately replicated, and the statistical analysis seems OK.

Minor comments:

- There are no specific experimental issues that require addressing.

- Prior studies are appropriately referenced

• In general, both the text and figures are clear and accurate. The significant alteration of metabolites found in their metabolomic dataset should be plotted using the online tool MetaboAnalyst to analyze metabolic pathways and generate better visualizations.

We thank the reviewer for this comment. We plotted our global metabolomics data by using MetaboAnalyst and we performed pathway enrichment analysis to determine the metabolic pathways which are significantly altered in MKO cells. The analyses appear in **Figure R2.7** below and in **Figure 1EVB and C** in the revised Manuscript.

Figure R2.7. **A.** Pathway enrichment analysis of 107 differential metabolites detected in the global metabolomics. Enrichment was detected using a Hypergeometric Test using a Relative-betweenness Centrality topology against the Homo sapiens (KEGG) database, using MetaboAnalyst server (n=4 independent biological replicates). **B.** Number of metabolites detected in significantly enriched pathways with an FDR cutoff < 0.12 (n=4 independent biological replicates).

• Overall, the presentation is satisfactory with only minor language adjustments recommended. A minor suggestion for improvement involves refining the language used in the text. Instead of consistently using the term "produce energy," please use "conversion of energy".

Thank you for this comment and we have modified the text accordingly.

Reviewer #3(Evidence, reproducibility and clarity(required))

This study by Chourasia et al determined the effects of MTCH2 deletion on the total metabolite content (polar and lipid metabolites) per total protein of HeLa cells, analyzed at different consecutive times after adding fresh and complete media (high glucose, 10%FBS). They also analyzed the effects of nutrient depletion (no FBS). In addition, authors assessed the effects of MTCH2 deletion on mitochondrial morphology, lipid droplet number and size in HeLa cells, as well as in the differentiation of mouse fibroblasts to adipocytes. From the metabolite snapshots under these different times and conditions, authors conclude that MTCH2 deletion increases mitochondrial oxidative function to induce a catabolic state, which impedes lipid synthesis and, as a result, adipocyte differentiation.

The major concerns are that it is unclear whether the metabolic phenotype observed is a consequence of MTCH2 deletion inducing a decrease in proliferation of HeLa, as well as of fibroblasts that need to reach confluence to differentiate. In this regard, it is also unclear whether MTCH2 deletion increases ATP demand and/or promotes a catabolic program, as metabolic flux analyses are missing.

We appreciate the reviewer's insightful/valuable comments. To address the issue regarding metabolic flux analysis, we monitored oxygen consumption using the Seahorse apparatus and found an increase in both basal and maximal oxygen consumption in MKO cells as compared to WT cells (see **Figure R2.3** below, and **new Figure 3A** in our revised manuscript). We also performed several additional experiments: 1) Measured respiratory control ratio (RCR) and found that it was higher in MKO cells as compared to WT cells (see **Figure R2.3** below and **new Figure 3C** in revised manuscript); 2) measured NADH autofluorescence and found that MKO cells have a lower NADH redox index as compared to WT cells (see **Figure R2.4.A** below and **new Figure 3B** in the revised manuscript), suggesting an increased respiration rate and accelerated electron consumption in MKO cells; 3) measuring NADH autofluorescence we also found that MKO cells have higher levels of mitochondrial NADH (**Figure R2.4.B** below and **Figure EV3B** in the revised manuscript), suggesting increased substrate availability in MKO cells resulting in a higher rate of NADH production and a proxy of TCA cycle efficiency; 4) measure mitochondrial membrane potential and found that MKO cells have a higher ΔY_m as compared to WT cells (lines 198-226 in our revised manuscript).

Taken together, these findings are consistent with the idea that MTCH2 deletion increases mitochondrial oxidative function, increases ATP demand and promotes a catabolic program. Consistent with the reviewer's comment, MTCH2 deletion indeed induces a decrease in proliferation of HeLa cells (see **Figure R2.5** above and **new Figure 3E** in our revised manuscript). Based on our new findings it is more likely that the metabolic phenotype observed is a consequence of MTCH2 deletion increasing mitochondrial oxidative function and not a consequence of decreased proliferation of HeLa cells. (lines 380-383 in our revised manuscript).

A minor concern is the use of computer (processing system), antenna, and wifi analogies to describe the role of MTCH2 in mitochondrial function, which is confusing.

We thank the reviewer for this comment and we have rephrased these analogies.

Reviewer #3 (Significance (Required)):

This study represents a thorough characterization of the metabolite content in proliferating HeLa cells in the absence of MTCH2 expression. The changes observed in polar and lipidic metabolites are novel, and interesting and contribute to our understanding on the role of MTCH2 function in cellular metabolism. The main limitation of the study is that the levels of most metabolites are normalized by protein content, comparing conditions in which cell number and protein synthesis have changed.

We appreciate the reviewer's insightful comment and agree that MTCH2 deletion leads to slower cell growth. In light of this, we ensured that the same number of cells was used for all metabolic and lipidomics measurements, and along with this we also normalized the data based on protein concentration, a standard methodology for interpreting metabolomics data. Additionally, in other experiments, we measured several housekeeping genes (e.g., GAPDH, β -Actin, Citrate Synthase, and mitochondrial ETC complexes II, III, and V) and found no significant differences in their levels between WT and MKO cells. Therefore, we do not expect any dramatic changes in overall protein expression in our system.

Thus, it is unclear whether some of the effects observed are a consequence of the reported role of MTCH2 supporting the proliferation of different tumors and cell lines, or whether it is a direct effect of MTCH2 increasing ATP demand and/or being a direct activator of mitochondrial catabolism. Related to this point, it is unclear whether the defect in adipocyte differentiation induced by MTCH2 KO in NIH3T3 fibroblasts might be caused by an inability of MTCH2 KO to reach confluency at day 0, needed for differentiation.

The first paragraph concerning proliferation versus ATP demand was addressed above.

We also acknowledge the reviewer's major concern regarding the confluency of fibroblasts (NIH3T3L1 preadipocytes) on differentiation Day 0. It is well established that preadipocytes require 85-95% confluency and growth arrest (Gregoire et al., 1998) as prerequisites for entering the commitment phase. To determine whether the inability of MTCH2 KO cells to achieve the same confluency as MTCH2 WT cells on Day 0 could be responsible for their failure to differentiate into mature adipocytes, we allowed the MTCH2 KO preadipocytes additional time to reach confluency before initiating differentiation. Despite reaching the required confluency, MTCH2 KO cells still failed to differentiate. Additionally, we observed that MTCH2 KO cells took longer to enter the growth arrest phase after the introduction of differentiation media, suggesting that these cells might require more time than MTCH2 WT cells, which typically differentiate fully into adipocytes within 6 days. To explore this further, we extended the differentiation period of NIH3T3L1 MTCH2 KO cells to 12 days but found no significant differences between Day 6 and Day 12. Therefore, we conclude that neither confluency nor extended time in culture facilitates the differentiation or lipid accumulation in MTCH2 KO cells.

Finally, respirometry and mitochondrial ROS content analyses would be needed to confirm that the changes in the metabolite levels induced by MTCH2 are caused by an increase in mitochondrial oxidation leading to nutrient depletion, as authors conclude. For example, an increase in the ADP/ATP ratio could also be caused by an inhibition of mitochondrial ATP synthase in the mitochondria, concurrent to an increase in ROS production, which would decrease NADH and NADPH content.

The first concern regarding respiration was addressed above.

We also measured mitochondrial ROS with MitoSOX and reported the median fluorescence Intensity (MFI) and did not find a significant difference between WT and MKO (see **Figure R3.1.** below and **Figure EV3D** in the revised manuscript).

Figure R3.1. Mitochondrial ROS level of WT and MKO cells. Mitochondrial ROS (measured using mitoSOX) are presented. Results are presented as mean \pm SEM. (Unpaired t-test, ns-non-significant). N=3 independent experiments).

References

- Bahat, Amir, Andres Goldman, Yehudit Zaltsman, Dilshad H. Khan, Coral Halperin, Emmanuel Amzallag, Vladislav Krupalnik, Michael Mullokov, Alon Silberman, Ayelet Erez, Aaron D. Schimmer, Jacob H. Hanna, and Atan Gross. 2018. 'MTCH2-Mediated Mitochondrial Fusion Drives Exit from Naïve Pluripotency in Embryonic Stem Cells'. *Nature Communications* 9(1). doi: 10.1038/s41467-018-07519-w.
- Brand, Martin D., and David G. Nicholls. 2011. 'Assessing Mitochondrial Dysfunction in Cells'. *Biochemical Journal* 435(2):297–312.
- Goldman, Andres, Michael Mullokov, Yehudit Zaltsman, Limor Regev, Smadar Levin-Zaidman, and Atan Gross. 2024. 'MTCH2 Cooperates with MFN2 and Lysophosphatidic Acid Synthesis to Sustain Mitochondrial Fusion'. *EMBO Reports* 25(1):45–67. doi: 10.1038/s44319-023-00009-1.
- Han, Mingqi, Eric A. Bushong, Mayuko Segawa, Alexandre Tiard, Alex Wong, Morgan R. Brady, Milica Momcilovic, Dane M. Wolf, Ralph Zhang, Anton Petcherski, Matthew Madany, Shili Xu, Jason T. Lee, Masha V. Poyurovsky, Kellen Olszewski, Travis Holloway, Adrian Gomez, Maie St John, Steven M. Dubinett, Carla M. Koehler, Orian S. Shirihai, Linsey Stiles, Aaron Lisberg, Stefano Soatto, Saman Sadeghi, Mark H. Ellisman, and David B. Shackelford. 2023. 'Spatial Mapping of Mitochondrial Networks and Bioenergetics in Lung Cancer'. *Nature* 615(7953):712–19. doi: 10.1038/s41586-023-05793-3.
- Han, Mingqi, Eric A Bushong, Mayuko Segawa, Alexandre Tiard, Alex Wong, Morgan R. Brady, Milica Momcilovic, Dane M. Wolf, Ralph Zhang, Anton Petcherski, Matthew Madany, Shili Xu, Jason T.

- Lee, Masha V Poyurovsky, Kellen Olszewski, Travis Holloway, Adrian Gomez, Maie St John, Steven M. Dubinett, Stefano Soatto, Saman Sadeghi, Mark H. Ellisman, and David B. Shackelford. 2023. 'Spatial Mapping of Mitochondrial Networks and Bioenergetics in Lung Cancer'. *Nature* 615(11):17. doi: 10.1038/s41586-023-05793-3.
- Labbé, Katherine, Shona Mookerjee, Maxence Le Vasseur, Eddy Gibbs, Chad Lerner, and Jodi Nunnari. 2021. 'The Modified Mitochondrial Outer Membrane Carrier MTCH2 Links Mitochondrial Fusion to Lipogenesis'. *Journal of Cell Biology* 220(11). doi: 10.1083/jcb.202103122.
- Liesa, Marc, and Orian S. Shirihai. 2013. 'Mitochondrial Dynamics in the Regulation of Nutrient Utilization and Energy Expenditure'. *Cell Metabolism* 17(4):491–506.
- Maryanovich, Maria, Yehudit Zaltsman, Antonella Ruggiero, Andres Goldman, Liat Shachnai, Smadar Levin Zaidman, Ziv Porat, Karin Golan, Tsvee Lapidot, and Atan Gross. 2015. 'An MTCH2 Pathway Repressing Mitochondria Metabolism Regulates Haematopoietic Stem Cell Fate'. *Nature Communications* 6. doi: 10.1038/ncomms8901.
- Monzel, Anna S., José Antonio Enríquez, and Martin Picard. 2023. 'Multifaceted Mitochondria: Moving Mitochondrial Science beyond Function and Dysfunction'. *Nature Metabolism* 5(4):546–62. doi: 10.1038/s42255-023-00783-1.
- Plun-Favreau, H., V. S. Burchell, K. M. Holmström, Z. Yao, E. Deas, K. Cain, V. Fedele, N. Moiso, M. Campanella, L. Miguel Martins, N. W. Wood, A. V. Gourine, and A. Y. Abramov. 2012. 'HtrA2 Deficiency Causes Mitochondrial Uncoupling through the F₁F₀-ATP Synthase and Consequent ATP Depletion'. *Cell Death and Disease* 3(6). doi: 10.1038/cddis.2012.77.
- Rogers, George W., Martin D. Brand, Susanna Petrosyan, Deepthi Ashok, Alvaro A. Elorza, David A. Ferrick, and Anne N. Murphy. 2011. 'High Throughput Microplate Respiratory Measurements Using Minimal Quantities of Isolated Mitochondria'. *PLoS ONE* 6(7). doi: 10.1371/journal.pone.0021746.
- Sharma, Anand Kumar, Tongtong Wang, Alaa Othman, Radhika Khandelwal, Miroslav Balaz, Salvatore Modica, Nicola Zamboni, and Christian Wolfrum. 2023. 'Basal Re-Esterification Finetunes Mitochondrial Fatty Acid Utilization'. *Molecular Metabolism* 71. doi: 10.1016/j.molmet.2023.101701.
- Sherry, Erica B., Phil Lee, and In Young Choi. 2015. 'In Vivo NMR Studies of the Brain with Hereditary or Acquired Metabolic Disorders'. *Neurochemical Research* 40(12):2647–85. doi: 10.1007/s11064-015-1772-1.
- Shiratori, Reika, Kenta Furuichi, Masashi Yamaguchi, Natsumi Miyazaki, Haruna Aoki, Hiroji Chibana, Kousei Ito, and Shigeki Aoki. 2019. 'Glycolytic Suppression Dramatically Changes the Intracellular Metabolic Profile of Multiple Cancer Cell Lines in a Mitochondrial Metabolism-Dependent Manner'. *Scientific Reports* 9(1). doi: 10.1038/s41598-019-55296-3.
- Yang, Chendong, Bookyung Ko, Christopher T. Hensley, Lei Jiang, Ajla T. Wasti, Jiyeon Kim, Jessica Sudderth, Maria Antonietta Calvaruso, Lloyd Lumata, Matthew Mitsche, Jared Rutter, Matthew E. Merritt, and Ralph J. DeBerardinis. 2014. 'Glutamine Oxidation Maintains the TCA Cycle and Cell Survival during Impaired Mitochondrial Pyruvate Transport'. *Molecular Cell* 56(3):414–24. doi: 10.1016/j.molcel.2014.09.025.

Youle, Richard J., and Alexander M. Van Der Bliek. 2012. 'Mitochondrial Fission, Fusion, and Stress'.
Science 337(6098):1062–65.

Dear Dr Gross,

Thank you for submitting your revised manuscript (EMBOJ-2024-117259R) to The EMBO Journal, as well for your patience with our response. Please note that I have taken over this process from my editorial colleague Kelly Anderson since she recently progressed to a different role outside of the office. Your amended study was sent back to the three referees for their scientific re-evaluation, and we have received detailed comments from all of them, which I enclose below. As you will see, the experts state that the work has been substantially improved by the revisions and they are now broadly in favour of publication, pending minor amendments.

Thus, we are pleased to inform you that your manuscript has been accepted in principle for publication in The EMBO Journal.

Please consider the remaining issues of referee #2 carefully and complement the manuscript accordingly by revisiting the discussion of the results where appropriate.

We also now need you to take care of a number of issues related to formatting and data presentation as detailed below, which should be addressed at re-submission.

Please contact me at any time if you have additional questions related to below points.

As you might have seen on our web page, every paper at the EMBO Journal now includes a 'Synopsis', displayed on the html and freely accessible to all readers. The synopsis includes a 'model' figure as well as 2-5 one-short-sentence bullet points that summarize the article. I would appreciate if you could provide this figure and the bullet points.

Thank you for giving us the chance to consider your manuscript for The EMBO Journal. I look forward to your final revision.

Again, please contact me at any time if you need any help or have further questions.

Best regards,

Daniel Klimmeck

>> Please provide maximally five keywords for your study.

>> Authors: please complement the information on the following authors in our online system: C.S., D.A., A.S., B.D. ; and revisit the author email provided for H.W. .

>> Author Contributions: Please remove the author contributions information from the manuscript text. Note that CRediT has replaced the traditional author contributions section as of now because it offers a systematic machine-readable author contributions format that allows for more effective research assessment. and use the free text boxes beneath each contributing

author's name to add specific details on the author's contribution.

More information is available in our guide to authors.

>> Rename the current 'Conflict of Interest' section to 'Disclosure and Competing Interests Statement'.

>> Correct order of manuscript sections: Abstract / Keywords / Introduction / Results / Discussion / Methods / Data Availability / Acknowledgements / Disclosure and competing interests statement // References / Figure legends / Tables and their legends / Expanded View Figure legends

>> Dataset EV legends: Table EV1 should be renamed Dataset EV1 and the legend should be removed from manuscript text and added to the excel file; Table EV2 should be renamed Table EV1 and it also needs its legend added. Table EV3 has a title included in the file as Table 1 => rename and add its legend.

>> References: adjust reference format to EMBO Journal format, 10 authors et al. . DOIs should be removed.

>> Recheck the bioRxiv citations Li et al (2023) and Bartoš et al (2023) for journal publication and update the journal reference in case.

>> Introduce ORCID IDs for all corresponding authors (S. C.) via our online manuscript system. Please see below for additional information.

>> Funding: information on funding is incomplete in our online system. All funders, including project numbers, need to be entered into our system in addition to being mentioned in the Acknowledgements.

>> Author checklist: please enter author name, journal name and manuscript number for your study.

>> Please remove the 'Data not shown' statement on p. 19.

>> Add a Reagents and Tools table to the Methods section, listing key reagents, experimental models, software and relevant equipment.

>> Data availability section: provide database access to the global metabolomics and lipidomics datasets; make sure the dataset privacy is released.

>> Provide a complete set of source data, uploaded as one (zipped) file per figure.

>> Consider additional changes and comments from our production team as indicated below:

- Figure legends:

1. Please define the annotated p values ****/** as well as provide the exact p-values for the same in the legend of figure EV 1a; EV 4e; EV 5a; as appropriate.
2. Please note that the exact p values are not provided in the legends of figures 1b; 2a-f; 3a-d; 4b, d-e; 5a, c-d; EV 2a-c; EV 3a-c; EV 4a-g.
3. Please indicate the statistical test used for data analysis in the legends of figures 4c; EV 1a; EV 4e; EV 5a.
4. Please note that in figure 5c; there is a mismatch between the annotated p values in the figure legend and the annotated p values in the figure file that should be corrected.
5. Please note that information related to n is missing in the legends of figures 3e; 4c, f; EV 1a; EV 5a.
6. Please note that the error bars are not defined in the legends of figures 3e; EV 1a; EV 4e; EV 5a.

Referee #1:

The authors have very thoroughly addressed my concerns. This is a superb paper that deserves to be published

Referee #2:

The authors performed the respirometry experiments requested, which confirm increased respiratory fluxes in the mitochondria that are consistent with increased ATP demand. Authors also provide an answer stating that they provided extra time to NIH3T3 MTCH2 KO cells to grow and reach confluency to differentiate. In addition, authors claim that the differences in cell growth and protein synthesis induced by MTCH2KO were not interfering with the metabolomics data, which is confirmed by the novel measurements of respiratory fluxes. Most of the technical concerns and experimental validations have been successfully completed. However, a thorough discussion of the new data, as well as including some of the data only discussed in the letter, would be necessary to complete the paper:

1) The authors demonstrated that MTCH2 increases mitochondrial ATP demand, while decreasing cell growth and anabolism. In the light of these data, which biological ATP-consuming process do the authors think that is being activated by MTCH2 deletion? This biological process should be stealing large amount of ATP from anabolism and cell growth, as it even requires elevating OXPHOS. Could it be MTCH2 deletion is blocking glycolysis to lactate, as this is a highly relevant metabolic ATP and NADH producing pathway in HeLa cells? The decrease in lactate levels support this possibility.

2) The permeabilized cell respirometry data, as well as the increase in maximal respiration, suggest that either there is an increase in the number of mitochondria in MTCH2 KO cells or that MTCH2KO mitochondria have an increased respiratory capacity per organelle. How do the authors explain this phenotype that goes beyond an increase in cellular ATP demand and nutrient wasting? Could it be that MTCH2 deletion, by limiting mitophagy, induces the accumulation of some dysfunctional mitochondria that consume ATP, while the other mitochondria inside the same cell compensate ATP wasting by increasing ATP production? Or maybe that MTCH2 deletion activates some signaling pathways to stop proliferation and glycolysis, which activate catabolism to secondarily activate OXPHOS?

In the light of these new data, speculative answers to these questions should be included in the discussion, as these answers are at the core of interpreting this exciting and compelling phenotype revealed by the authors. This discussion might be key to inspiring future research on understanding MTCH2 mechanisms of action.

Referee #3:

The Authors did an outstanding job addressing my comments and the comments of the other referees.

Rev_Com_number: RC-2023-02310

New_manu_number: EMBOJ-2024-117259R

Corr_author: Gross

Title: High-energy demand and nutrient exhaustion in MTCH2 knockout cells

The authors addressed the remaining editorial issues.

Dear Dr Gross,

Thank you for submitting the revised version of your manuscript. I have now evaluated your amended manuscript and concluded that the remaining minor concerns have been sufficiently addressed.

I am thus pleased to inform you that your manuscript has been accepted for publication in the EMBO Journal.

Related I would like to hereby ask your consent on keeping the referee figures included in this file.

On a different note, I would like to alert you that EMBO Press offers a format for a video-synopsis of work published with us, which essentially is a short, author-generated film explaining the core findings in hand drawings, and, as we believe, can be very useful to increase visibility of the work. Please see the following link for representative examples and their integration into the article web page:

<https://www.embopress.org/doi/full/10.15252/emj.2019103932>

Best regards,

Daniel Klimmeck

Daniel Klimmeck, PhD
Senior Editor
The EMBO Journal

Rev_Com_number: RC-2023-02310

New_manu_number: EMBOJ-2024-117259R1

Corr_author: Gross

Title: High-energy demand and nutrient exhaustion in MTCH2 knockout cells